# Whole-Genome Sequencing and Machine Learning Analysis of *Staphylococcus aureus* from Multiple Heterogeneous Sources in China Reveals Common Genetic Traits of Antimicrobial Resistance

Wei Wang,[a] Michelle Baker,[b] Yue Hu,[b] Jin Xu,[a] Dajin Yang,[a] Alexandre Maciel-Guerra,[c] Ning Xue,[b] Hui Li,[a] Shaofei Yan,[a] Menghan Li,[a] Yao Bai,[a] Yinping Dong,[a] Zixin Peng,[a] Jinjing Ma,[a,d] Fengqin Li,[a] Tania Dottorini[b]

[a]NHC Key Laboratory of Food Safety Risk Assessment, China National Center for Food Safety Risk Assessment, Beijing, China
[b]School of Veterinary Medicine and Science, University of Nottingham, Sutton Bonington, Leicestershire, United Kingdom
[c]School of Computer Science, University of Nottingham, Nottingham, United Kingdom
[d]School of Chemistry and Chemical Engineering, Anqing Normal University, Anqing, Anhui, China

**ABSTRACT** *Staphylococcus aureus* is a worldwide leading cause of numerous diseases ranging from food-poisoning to lethal infections. Methicillin-resistant *S. aureus* (MRSA) has been found capable of acquiring resistance to most antimicrobials. MRSA is ubiquitous and diverse even in terms of antimicrobial resistance (AMR) profiles, posing a challenge for treatment. Here, we present a comprehensive study of *S. aureus* in China, addressing epidemiology, phylogenetic reconstruction, genomic characterization, and identification of AMR profiles. The study analyzes 673 *S. aureus* isolates from food as well as from hospitalized and healthy individuals. The isolates have been collected over a 9-year period, between 2010 and 2018, from 27 provinces across China. By whole-genome sequencing, Bayesian divergence analysis, and supervised machine learning, we reconstructed the phylogeny of the isolates and compared them to references from other countries. We identified 72 sequence types (STs), of which, 29 were novel. We found 81 MRSA lineages by multilocus sequence type (MLST), *spa*, staphylococcal cassette chromosome *mec* element (SCC*mec*), and Panton-Valentine leukocidin (PVL) typing. In addition, novel variants of SCC*mec* type IV hosting extra metal and antimicrobial resistance genes, as well as a new SCC*mec* type, were found. New Bayesian dating of the split times of major clades showed that ST9, ST59, and ST239 in China and European countries fell in different branches, whereas this pattern was not observed for the ST398 clone. On the contrary, the clonal transmission of ST398 was more intermixed in regard to geographic origin. Finally, we identified genetic determinants of resistance to 10 antimicrobials, discriminating drug-resistant bacteria from susceptible strains in the cohort. Our results reveal the emergence of Chinese MRSA lineages enriched of AMR determinants that share similar genetic traits of antimicrobial resistance across human and food, hinting at a complex scenario of evolving transmission routes.

**IMPORTANCE** Little information is available on the epidemiology and characterization of *Staphylococcus aureus* in China. The role of food is a cause of major concern: staphylococcal foodborne diseases affect thousands every year, and the presence of resistant *Staphylococcus* strains on raw retail meat products is well documented. We studied a large heterogeneous data set of *S. aureus* isolates from many provinces of China, isolated from food as well as from individuals. Our large whole-genome collection represents a unique catalogue that can be easily meta-analyzed and integrated with further studies and adds to the library of *S. aureus* sequences in the public domain in a currently underrepresented geographical region. The new Bayesian dating of the split times of major

Address correspondence to Fengqin Li, lifengqin@cfsa.net.cn, or Tania Dottorini, Tania.Dottorini@nottingham.ac.uk.

drug-resistant enriched clones is relevant in showing that Chinese and European methi-cillin-resistant *S. aureus* (MRSA) have evolved differently. Our machine learning approach, across a large number of antibiotics, shows novel determinants underlying resistance and reveals frequent resistant traits in specific clonal complexes, highlighting the importance of particular clonal complexes in China. Our findings substantially expand what is known of the evolution and genetic determinants of resistance in food-associated *S. aureus* in China and add crucial information for whole-genome sequencing (WGS)-based surveillance of *S. aureus*.

**KEYWORDS** *Staphylococcus aureus*, antimicrobial resistance, methicillin-resistant *Staphylococcus aureus* (MRSA), whole-genome sequencing, Bayesian divergence analysis, supervised machine learning, foodborne pathogen

*S*taphylococcus aureus is a leading cause of numerous illnesses ranging from food poisoning to life-threatening diseases worldwide (1–4). The bacterium has an extraordinary capacity for acquiring new antimicrobial resistance traits. Recently, the emergence and diffusion of multidrug-resistant (MDR) *S. aureus*, particularly methicil-lin-resistant *S. aureus* (MRSA), have attracted considerable public health attention (5–7). The World Health Organization (WHO) has included MRSA on its "global priority list of antimicrobial-resistant bacteria to guide research, discovery, and development of new antimicrobials" (8).

When first reported in 1961 (9), MRSA had been found mostly in hospitals (health care-associated MRSA [HA-MRSA]) (10, 11). By the late 1990s (12, 13), it had been found in individuals with no history of exposure to health care (community-associated MRSA [CA-MRSA]). In 2003, a livestock-associated MRSA (LA-MRSA) sequence type 398 (ST398) clone was first isolated from a human patient in Europe (14). Further evidence showed that LA-MRSA ST398 can cause infection in humans via direct contact with farm animals (15). In recent times, HA-, CA-, and LA-MRSA clones have been increas-ingly found outside their original reservoirs (16–20).

MRSA lineages appear to vary geographically (21). Most of the hospitals in Asia are endemic for MRSA, with two major pandemic HA-MRSA clones, ST239-staphylococcal cassette chromosome *mec* element type III/IIIA (SCC*mec* III/IIIA) and ST5-SCC*mec* II (22). Conversely, CA-MRSA varies significantly, ranging between <5% and >35% (22), and is characterized by clonal heterogeneity, with the major prevalent clones being ST30-SCC*mec* IV, ST59-SCC*mec* IV/V, ST22-SCC*mec* IV, and ST772-SCC*mec* V (20, 22). In China, ST239-SCC*mec* III/II and ST5-SCC*mec* II are also found as the prevalent HA-MRSA (22, 23), while the ST59-SCC*mec* IV/V is the prevalent clone that has been associated with CA-MRSA (22, 23). ST9 is the dominant LA-MRSA clone in China and other Asian coun-tries as opposed to ST398 in Europe (22, 23).

*S. aureus* causes approximately 20% to 25% of foodborne disease outbreaks in China (24) and is included in all food surveillance programs under the Food Safety Law of the People's Republic of China. However, very limited knowledge on transmission routes and the genomic and evolutionary relationships between lineages of this bacte-rium has been acquired so far. Here, we report a comprehensive investigation on *S. aureus* performed on 673 isolates collected from food and human samples in China between 2010 and 2018. Epidemiology, phylogenetic reconstruction, genomic charac-terization, and antibacterial resistance profiling were performed using whole-genome sequencing, Bayesian divergence analysis, and several types of supervised machine learning methods.

## RESULTS

**High genomic diversity, novel sequence types, and MRSA and MSSA monophyletic clusters characterized by specific CCs.** Food-associated *S. aureus* isolates used in this study were selected from a collection of 7,937 isolates collected between 2010 and 2018 from different foods in 27 provinces in China. For this study, we selected 593 food-associated *S. aureus* isolates (343 MRSA and 250 methicillin-susceptible *S. aureus*

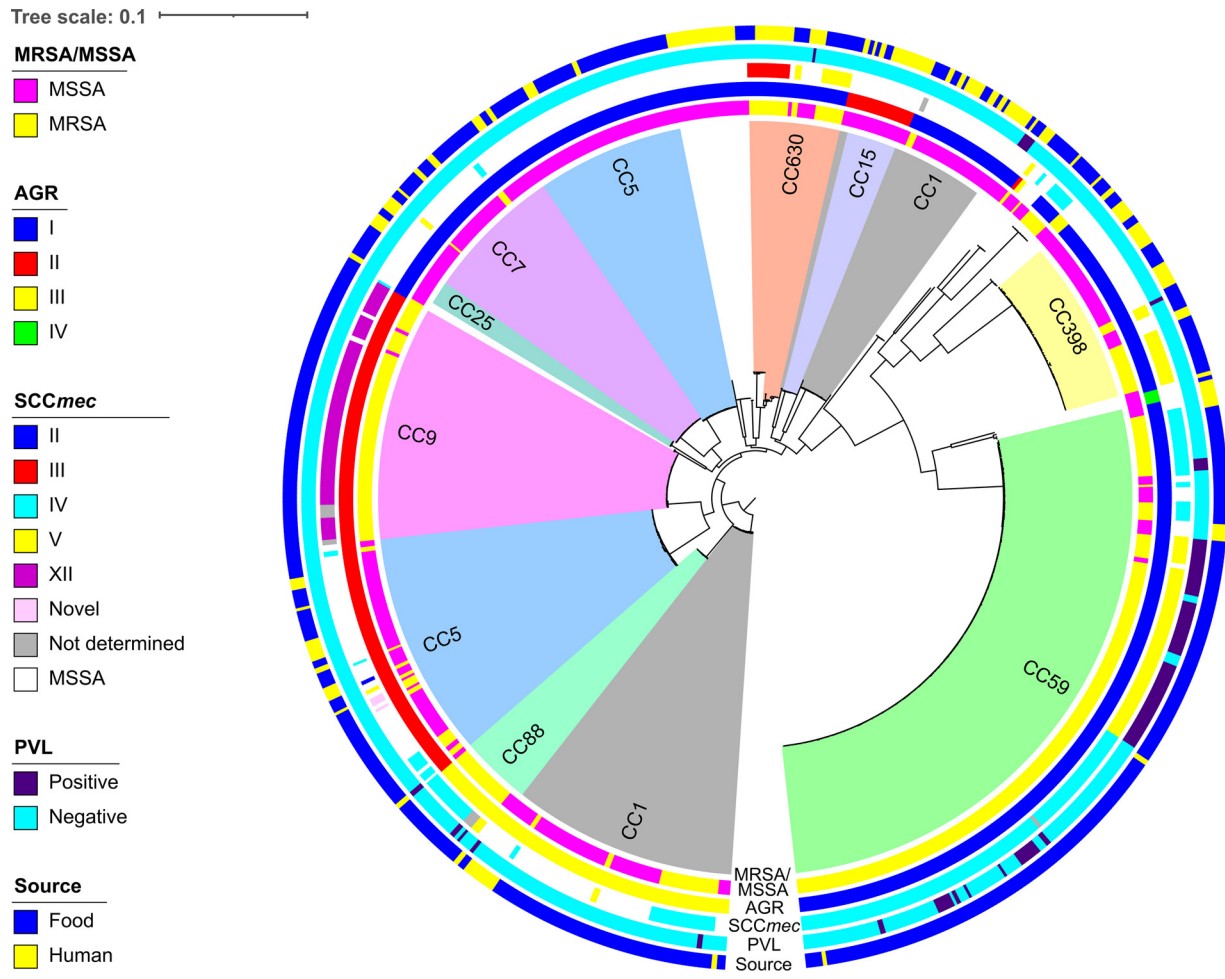

**FIG 1** Sample information and phylogenetic tree of the whole cohort. Maximum likelihood phylogenetic tree based on 1,585 core genes of the 673 *S. aureus* isolates. Clonal complex clusters are distinguished by means of numbers and background colors. Methicillin-resistant and -susceptible phenotypes are indicated by the inner colored ring; *agr* subgroup, SCC*mec* type, PVL type, and the sample sources are color coded in the following rings. One branch consisting of 5 isolates was removed from the tree, as the length of the branch made visualization of the tree difficult. These isolates were all found to be *S. aureus* (subgroup *S. argenteus*). The full tree is shown in Fig. S2 in the supplemental material.

[MSSA]) and sequenced them together with 142 isolates (18 MRSA and 124 MSSA) obtained from 53 healthy and 89 infected people in Shanghai between 2015 and 2017 (see Table S1 in the supplemental material). Of these, 673 resulted in high-quality genomes, 343 being MRSA and 330 being MSSA (see Fig. S1 and Table S1). The 673 isolates were tested for susceptibility against a panel of 13 antimicrobials: 97% showed resistance to at least one, and 72% showed resistance to at least three (Fig. S1d and Table S1). As in previous studies in China (25, 26), high prevalence of resistance to penicillin (93%), erythromycin (67%), cefoxitin (58%), oxacillin (57%), and clindamycin (55%) was found. More than 99% of isolates were found to be susceptible to drugs of last resort, vancomycin and linezolid (Fig. S1d).

The phylogenetic tree was reconstructed with a set of 1,585 concatenated core genes from the 673 *S. aureus* isolates and showed strong clustering by clonal complex (CC) (27), sequence type (ST) (28), and *agr* subgroup (29) (Fig. 1). A total of 72 STs were identified, of these, 29 STs (from 47 isolates) were found to be novel (Tables S1 and 2), showing a single-base mutation in one or more alleles. Two STs (ST3656 and ST5713) of these 29 were identified from human-associated isolates. The other 27 STs were from food-associated isolates, most of which were animal food related (24 STs from 36 isolates) (Table S2). The resistance and virulence profiles of these novel STs were similar

to those of their clonal complexes. Of the 72 identified STs, 57 (representing 92% of the isolates) were related to 10 CCs, CC59 being the most prevalent (187 isolates) followed by CC5 (111 isolates). The *spa* typing (30) led to 114 different *spa* types, t437 being the most prevalent (157 isolates). The *agr* subgroups revealed that all 4 known *agr* subgroups are present in our cohort, *agr* group I being the most prevalent ($n = 418$) (Fig. 1 and Table S1).

MRSA isolates were found to cluster by CC and by type of staphylococcal cassette chromosome *mec* (SCC*mec*) (31). Interestingly, 93% of the isolates in CC59 and 97% of the isolates in CC9 were MRSA (Fig. 1). CC88 isolates were predominantly MRSA (86%) as were those of CC630 (78%) (Fig. 1). Five SCC*mec* types were identified within the 343 MRSA isolates. The predominant SCC*mec* type for food-associated MRSA was IV ($n = 165$) followed by V ($n = 87$) and XII ($n = 63$). In comparison, SCC*mec* III ($n = 11$) was dominant among human-associated MRSA isolates. Notably, SCC*mec* III and SCC*mec* XII were solely found in ST239 (CC630) and CC9, respectively, while more diversity was found within type IV and V cassettes. Analysis of the Panton-Valentine leukocidin (PVL) genes indicated that 76 isolates (11% [76/673]) were PVL positive, and 70 of these (92% [70/76]) were MRSA (Fig. 1 and Table S1). No human-associated MRSA were PVL$^+$.

The presence of different MRSA lineages was investigated by combining multilocus sequence type (MLST), *spa*, SCC*mec*, and PVL typing. We found 81 MRSA lineages across the 343 MRSA isolates (see Table S3), the five most prevalent being ST59-t437-SCC*mec* IV-PVL$^-$ ($n = 75$, 21.9%), ST9-t899-SCC*mec* XII-PVL$^-$ ($n = 53$, 15.5%), ST59-t437-SCC*mec* V-PVL$^+$ ($n = 34$, 9.9%), ST59-t437-SCC*mec* IV-PVL$^+$ ($n = 17$, 5.0%), and ST398-t011-SCC*mec* V-PVL$^-$ ($n = 13$, 3.8%). These results are consistent with those reported elsewhere in Asia (23, 32). ST59-t437-SCC*mec* IV/V isolates were primarily found in food sources, with only two isolates (both ST59-t437-SCC*mec* IV-PVL$^-$) sourced from humans. ST9-t899-SCC*mec* XII-PVL$^-$ was the predominant LA-MRSA, only found in pork, as has been seen in other studies (33, 34). Conversely, ST398-t011-SCC*mec* V-PVL$^-$ was found in different food sources, including cake, chicken, fruit, noodle, pork, sandwich, and vegetable (while in Europe and North America it is commonly found in pigs).

We searched the genomes for the presence of virulence factors and identified a wide range of genes (Table S1), including those for adherence (*clf*, *cna*, *ebp*, *fnb*, *map*, *sdr*, *spa*, *vWbp*, and *ica*), alpha-hemolysin precursor (*hly* and *hla*), aureolysin (*aur*), capsular polysaccharide (*cap8*), capsular polysaccharide (*chp*), glycerol ester hydrolase (*geh* and *scn*), hyaluronate lyase precursor (*hysA*), immune evasion (*adsA*, *esa*, *ess*, *esx*, *hlb*, *hld*, *hlg*, *lukD*, *lukF-PVL*, *lukS-PVL*, and *sbi*), ion transporter (*isd*), serine protease (*ssp*), specific sortase (*srtB*), staphylocoagulase (*coa* and *sak*), toxin (*eta*, *se*, *sel*, and *tsst-1*), and triacylglycerol lipase precursor (*lip*).

**Variants of type IV SCC*mec* carrying extra antibiotic resistance genes and a novel SCC*mec* type.** Among the identified 343 MRSA isolates, nearly half of the isolates (169/343) carried a type IV SCC*mec*. However, in this study, by aligning reconstructed SCC*mec* with the reference cassettes (reference [Ref] IVa, AB063172.2; Ref IVc, B096217.1), we observed several variabilities inside the type IV cassette. Two subtype IVc isolates were found to have insertions of kanamycin and bleomycin resistance genes, as previously found in Italy by Manara et al. (35). Our isolates (11A1151 and 18A25) were almost identical to the cassette (MF062) found by Manara et al. (35), see Fig. 2. Isolate 11A1151 carried a cassette identical to MF062, while isolate 18A25 had a deletion of the gene *uqpQ*. We confirmed our variant to be an integrated plasmid, pUB110 (36), via a BLAST search against N315, which is a confirmed SCC*mec* II isolate with integrated pUB110 (accession number D81934.2; identity, 99.93%). The insertion of plasmid pUB110 carrying these resistance genes is frequently associated with type II SCC*mec* elements (37–40).

An SCC*mec* IVa variant was found, in two isolates from our cohort, to have genes conferring resistance to beta-lactams; *blaI*, *blaR1*, and *blaZ* were found between IS*431* and the *mec* gene complex. When blasting this SCC*mec* variant against the NCBI nonredundant database as the subject sequence, we found that this SCC*mec* variant (99% identity and

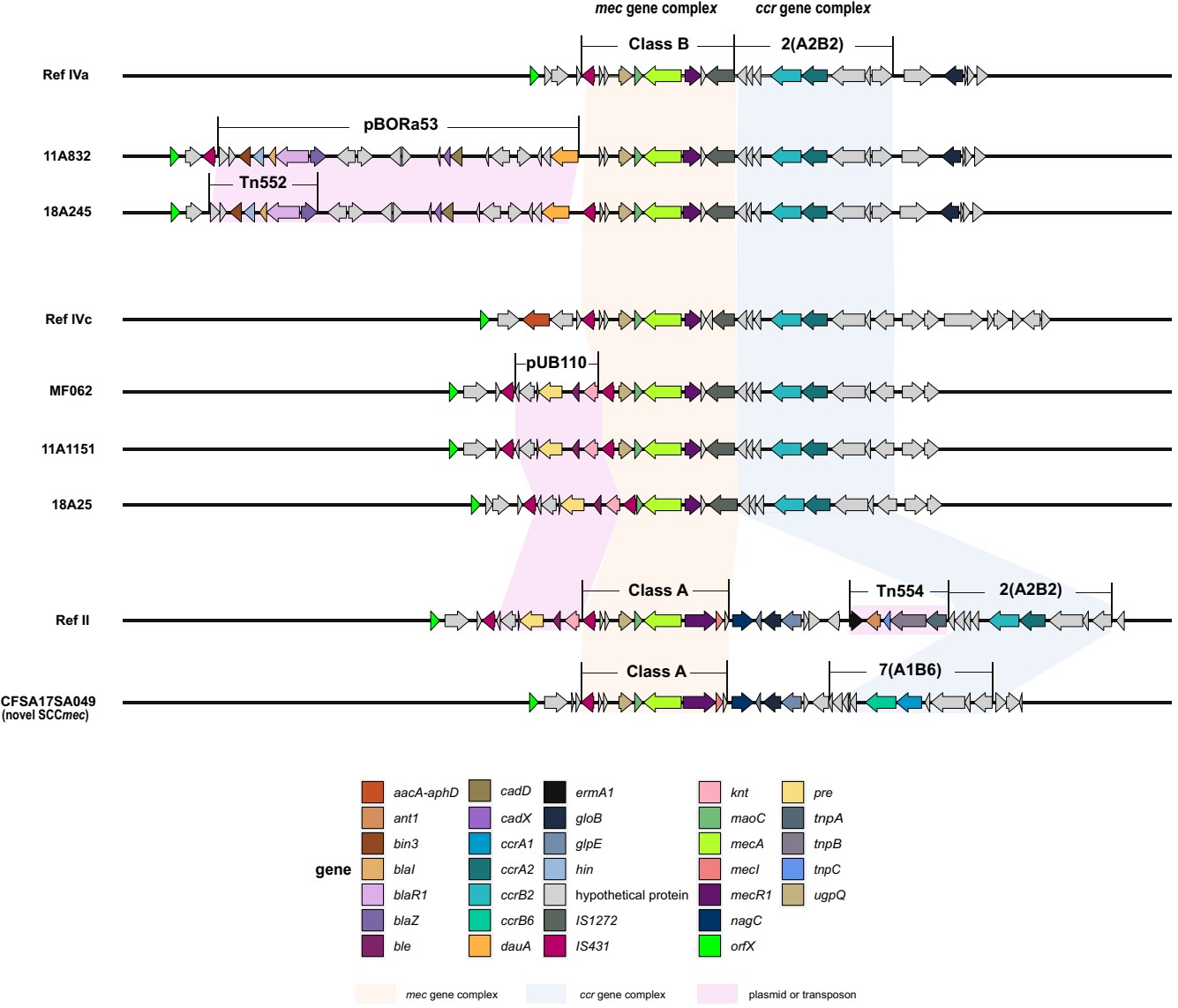

**FIG 2** Schematic representation of SCC*mec* IV variability carrying extra antimicrobial resistance genes and of a novel SCC*mec* type found in our cohort. Genes are shown with the direction of transcription and color coded according to their gene name. Integrated transposons or plasmids (pBORa53, Tn*552*, Tn*554*, and pUB110) upstream or downstream of the *mec* complex and carrying resistance genes are shaded in pink. The *mec* gene complex (class A and B) and the *ccr* gene complexes 2 (A2B2) and 7 (A1B6) are shaded in light orange and blue, respectively. SCC*mec* IV elements are compared with the available references in the GenBank database (Ref IVa, AB063172.2; Ref IVc, B096217.1) for the recovered subtypes IVa and IVc. A recently reported variant of SCC*mec* IVc element (MF062, GenBank GCA_003240235) closely related to that in our study is also shown. A novel SCC*mec* element carrying the *ccr* gene complex 7 (*ccrA1* and *ccrB6*) and the *mec* gene complex class A (*mecI-mecR1-mecA*-IS*431*) is aligned to the closest reference cassette, SCC*mec* type II (Ref II, D86934.1).

query cover) was recently also found in an SCC*mec* IVa clinical isolate from Wuhan, China (GenBank CP033086.1). The genome sequence located in the J3 region (40) of the cassette indicated an integrated plasmid as its structure, where Tn*552-CadX-CadD* is associated with plasmid pBORa53 (41). The plasmid-like region also shows high similarity (100% identity and 94% query cover) to *S. aureus* plasmid pPM1 (GenBank AB699881.1).

According to the International Working Group on the Classification of Staphylococcal Cassette Chromosome Elements and recent reports, there are 14 types of SCC*mec* (31, 42–44). Three of our MRSA isolates were not attributed to any of the 14 known cassette types. These novel SCC*mec* elements carried the *ccr* gene complex 7 (A1B6) and the *mec* gene complex class A. Therefore, our study highlights the presence of further resistances and diversity within the same cassette type and the ongoing rearrangements in MRSA.

**Monophyletic clustering separated STs from reference genomes of other countries.** Sequence types ST59, ST9, ST398, and ST239 are widespread in China and significantly present in our cohort and include the clones most enriched in methicillin resistance. Hence, we investigated the molecular evolution and global distribution of these isolates in our cohort. The STs were analyzed to reconstruct the geographical and temporal evolution. The STs were investigated alongside publicly available isolates (PATRIC database [45]) for contextualization against the global backdrop of *S. aureus* evolution. We first reconstructed the phylogenies of the selected STs using all available good-quality reference genomes for STs, ST59 ($n = 92$), ST9 ($n = 79$), ST398 ($n = 158$), and ST239 ($n = 194$), and a core genome maximum likelihood approach (Fig. 3 and Table S4).

It is noteworthy that the Chinese ST59-SCC*mec* IV/V isolates showed monophyletic subtree profiles compared to those of the reference genomes for the same ST (Fig. 3a), suggesting a single introduction event in China followed by a spread in the Chinese territory. Exceptions were a few ST59 isolates coming from Japan ($n = 3$), Denmark ($n = 5$), and Italy ($n = 2$) that clustered with the Chinese ones, indicating potential effects of travelling and commerce. ST9 isolates from our cohort were found mainly in pork samples. They clustered with reference isolates recovered from China and a single isolate from the Netherlands (Fig. 3b and Table S4). The only exception was an isolate from pork in Beijing, which clustered with pig isolates from the United States. One possible explanation for this U.S./China clustering is that China is one of the top three markets for U.S. pork production (46), and the MRSA isolates might be transported during pork trade. As previously observed, MRSA clones can transmit cross-regionally via food trading (47–49), although evidence of a risk to humans from this is lacking. ST398 (Fig. 3c) is commonly found in livestock-associated isolates from pig farming in Europe (50, 51) and also in China (52), though at very low prevalence. It is possible that this isolate derives from ST398 sourced from swine in China. However, here, ST398 was found in higher numbers (5%, $n = 34$) from human and more diverse food sources, including pork and nonmeat products. Several studies have shown that ST398 is becoming commonly present in humans (35, 53, 54), suggesting that humans could be the most obvious source of contamination for food preparations. A total of 11 isolates (1.6% of all isolates, 78.6% of human MRSA isolates) were found to be ST239 within our cohort (Fig. 3d). Nine (with *spa* type t037) were human health care associated and clustered together, but the other two (t030) were food associated and clustered with reference Chinese human isolates (t030). The nine human isolates all carried the virulence gene *sasX*, widely associated with human-associated ST239 in China (53, 55). None of the other isolates in this study carried this gene, including the two food-associated ST239 isolates. Globally, ST239 was previously found to be associated with food animals (56, 57), but to our knowledge, this has not been found to be the case in China (58) and suggests that ST239 in China is not exclusively transmitted in clinical settings but has the potential to spread between food and humans.

To better understand the evolution of the major MRSA lineages, Bayesian divergence analysis for MRSA isolates belonging to ST59-SCC*mec* IV/V, ST9-t899-SCC*mec* XII, ST398-t011-SCC*mec* V, and ST239-t037-SCC*mec* III was performed (Fig. 4 and Table S4). Our isolates clustered away from those from other countries in the maximum likelihood trees and were characterized by different divergence times. The mean substitution rate was $2.118 \times 10^{-6}$ (highest posterior density [HPD] 95%, $1.958 \times 10^{-6}$ to $2.523 \times 10^{-6}$) substitutions per site per year (59, 60). For ST59-SCC*mec* IV/V, the non-Chinese isolates showed a pattern of divergence starting in the 1960s, with divergence from Chinese clones occurring in the 1940s, earlier than another study of East Asian clones (61); however, small sample sizes in both studies may explain the differences. On the contrary, Chinese isolates diversified from ~1980 onwards, with the majority of divergence occurring after ~1990, in agreement with another study of food-related *S. aureus* isolates in China (62). Our data included a small number of human samples from China within this clone (two from our cohort and three reference genomes). These isolates showed no differential evolution to the food-related *S. aureus* isolates and were most closely related to

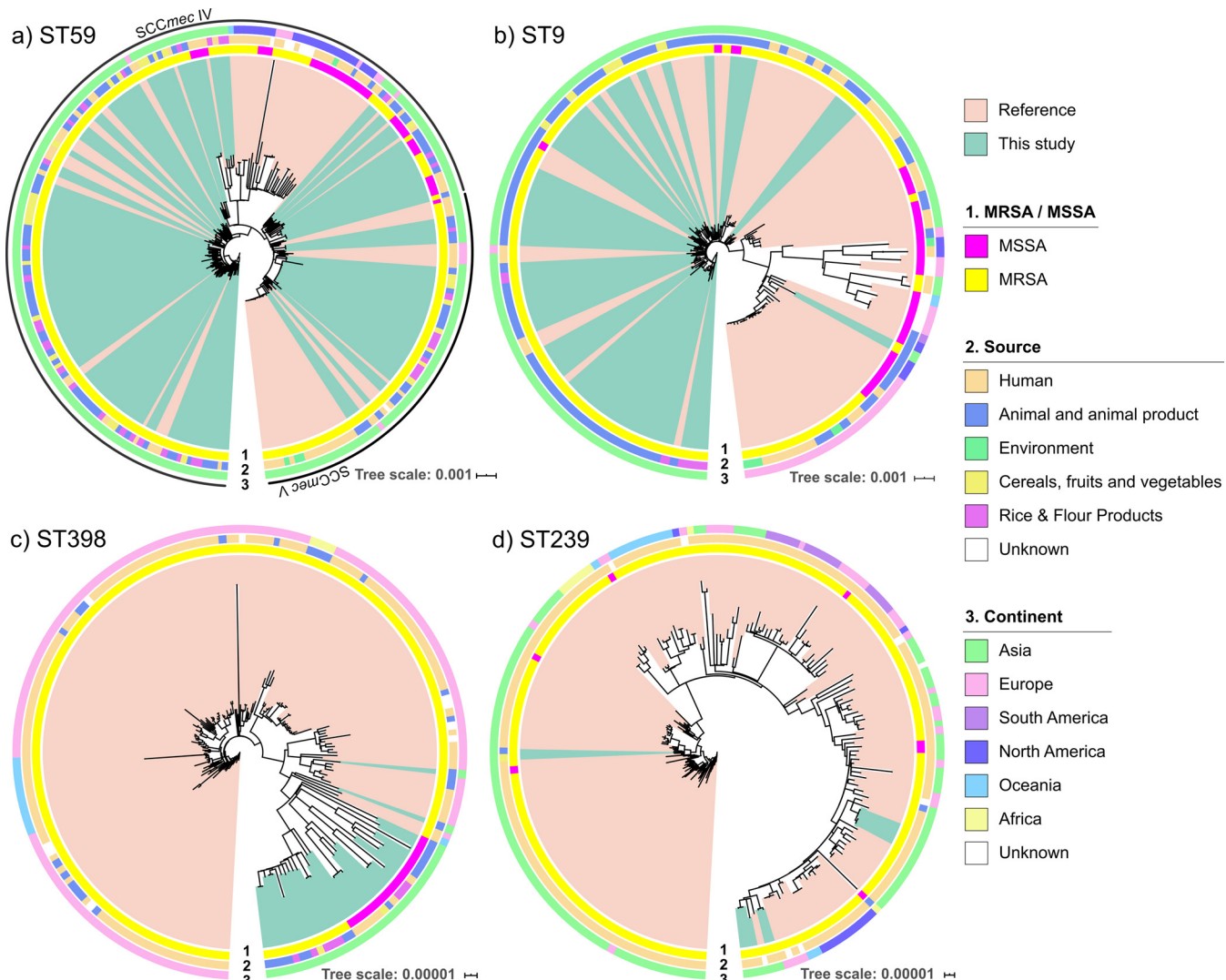

**FIG 3** Whole-genome maximum likelihood phylogenetic trees of the four major ST subsets. Plots show all the isolates from this study together with all publicly available reference genomes for ST59, ST9, ST398, and ST239. Methicillin-resistant and -susceptible genotypes are indicated by the inner colored ring. Origin (sample sources) and region are color coded in the following rings. Isolates from this study and publicly available reference genomes are color coded in salmon and cyan, respectively. (a) ST59 isolates. The Chinese ST59 SCC*mec* V and ST59 SCC*mec* IV isolates clustered separately from the reference genomes collected elsewhere. ST59 isolates from our Chinese cohort were almost exclusively MRSA and *spa* type t437 and t441, in contrast to what was found elsewhere. (b) ST9 isolates. The ST9 isolates from our cohort clustered together with isolates from China and away from those collected elsewhere. ST9 isolates from our Chinese cohort were almost exclusively MRSA and *spa* type t899, in contrast to what was found elsewhere. (c) ST398 isolates. The Chinese ST398 clustered separately from the reference genomes collected elsewhere. The Chinese isolates are mainly MSSA and show a wide diversity of origin, unlike in Europe, where they are linked to pig farming. (d) ST239 isolates. The clustering indicates a close relationship between U.K. samples and samples from this study, possibly indicating international routes of transmission. Two food-associated MRSA isolates (t030) were found in this study and clustered far away from the other nine HA-MRSA isolates in this study (t037).

isolates from both meat and vegetable products, suggestive of CA (rather than HA) transmission. ST9-t899-SCC*mec* XII isolates showed much more recent diversification, with our isolates diversifying between 2004 and 2010 and the most recent common ancestor dating to the year 2000. Diversification of the Chinese ST398-t011-SCC*mec* V isolates dated back to ~2008, later than human ST398 isolates in a recent study in Taiwan (63), with their most recent common ancestor to the European strains dating back to approximately 1996. ST239-t037-SCC*mec* III showed a diverse pattern of evolution, with diversification starting in the 1950s. Our Chinese isolates diversified from 1995 onwards and were found most closely related to other samples from China. Two of our samples were more closely related to samples from Algeria with earlier diversification. The evolution of our Chinese cohort appears to differ from geographically local countries, such as Singapore (64), possibly due to different political and economic histories.

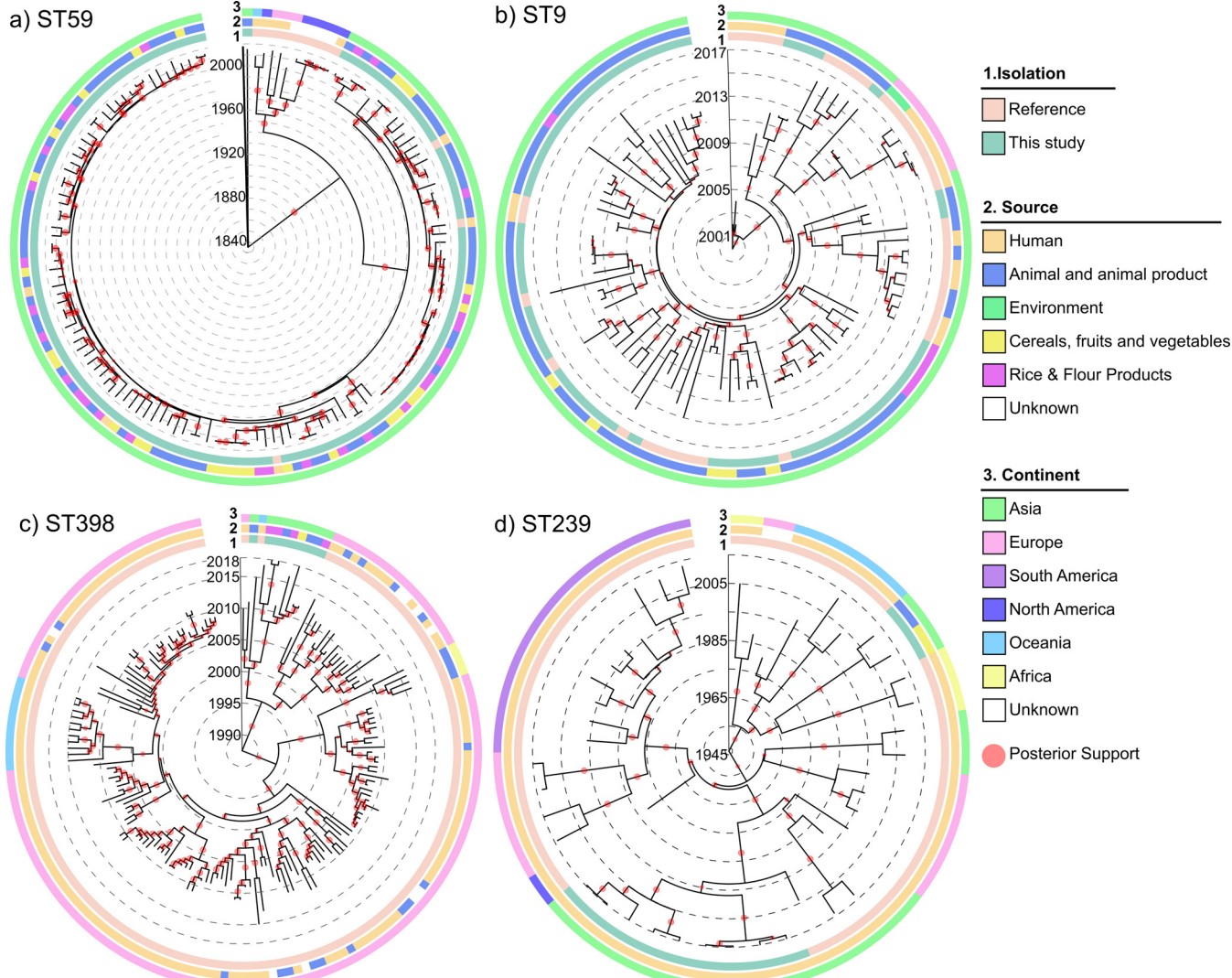

**FIG 4** Evolution of four major MRSA lineages. Bayesian evolutionary analysis of ST59-SCC*mec* IV/V (a), ST9-t899-SCC*mec* XII (b), ST398-t011-SCC*mec* V (c), and ST239-t037-SCC*mec* III (d). The lineages are evolutionarily distinct. In all CC subsets, East Asian isolates cluster separately and have generally evolved later than European and American branches. Isolates from this study and publicly available reference genomes are indicated by the inner ring. Origin (sample sources) and region are color coded in the following rings. The size of the red circles on trees represents the posterior probability of each node.

Overall, the molecular evolution of MRSA in China appears to be more recent than in other countries.

**Resistance to oxacillin and cefoxitin appears strongly associated with the clonal complexes CC59 and CC9.** A search of features in the genome sequence of each isolate which could strongly correlate to resistance to each one of the 10 selected antimicrobials was implemented using multiple supervised machine learning methods. Support vector machines (SVMs) are powerful yet flexible supervised machine learning algorithms which aim to classify the data by finding an N-dimensional hyperplane to separate the data points. In this study, the best overall performance considering all antibiotic models was obtained using a linear radial basis function (RBF) SVM classifier: accuracy, 76.6% to 95.5% (range of means across all antibiotic models); area under the curve (AUC), 66.5% to 96.2% (range of means across all antibiotic models); sensitivity, 5.8% to 99.9% (range of means across all antibiotic models); specificity, 32.9% to 99.8% (range of means across all antibiotic models) (Fig. 5a and Table S5). The best predictions of resistance/susceptibility were obtained for oxacillin (accuracy, 93.85%; sensitivity, 89.42%; specificity, 99.71%; and AUC, 96.24%) and cefoxitin (accuracy, 92.83%; sensitivity, 87.70%;

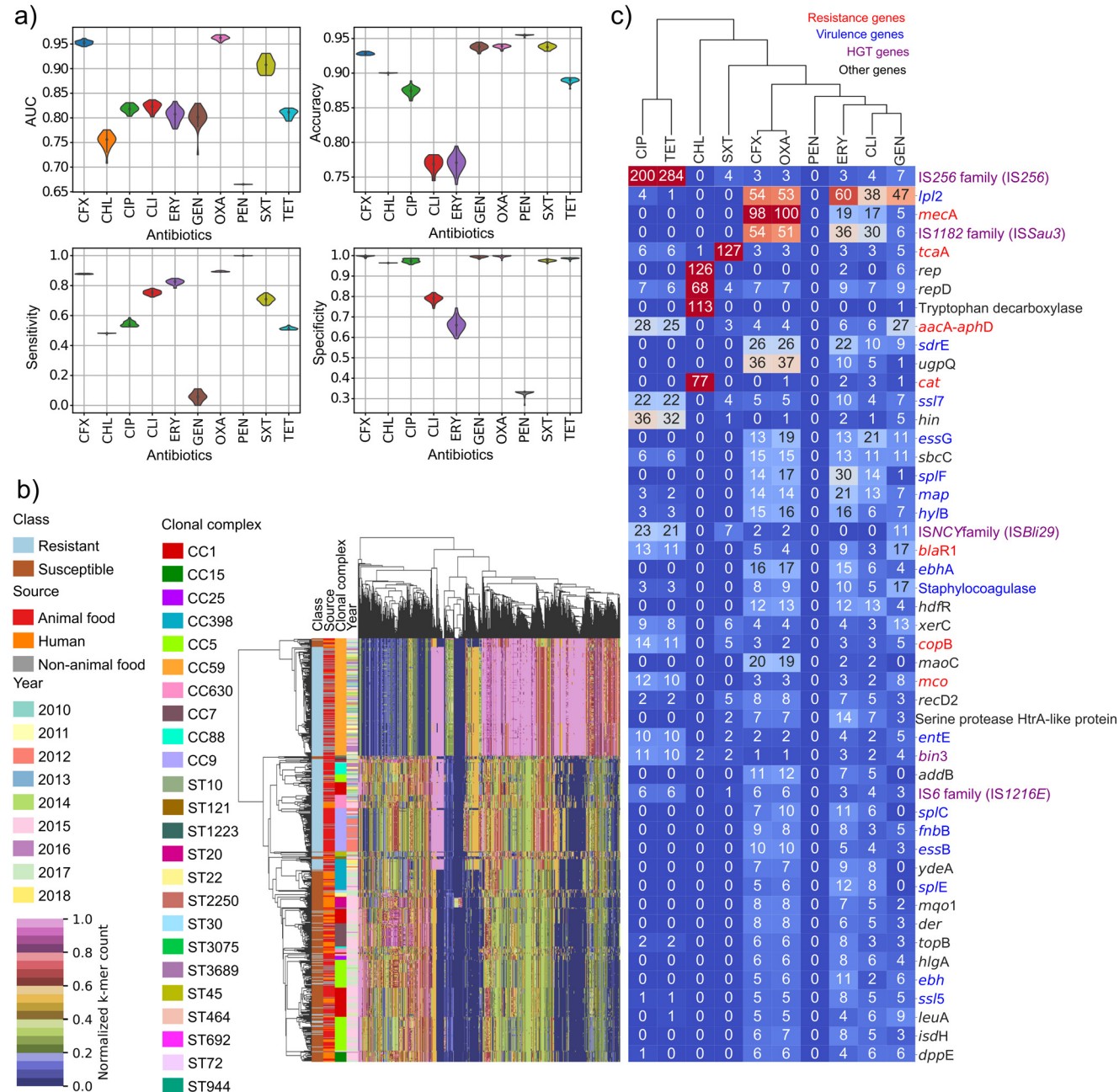

**FIG 5** Supervised machine learning prediction of antimicrobial resistance signature profiles to 10 antimicrobials in the *S. aureus* Chinese cohort. (a) Prediction performance results of the RBF SVM classifier that achieved the best performance among the three investigated machine learning classifiers. The scores for each performance metric (*y* axes) are mean AUC, accuracy, sensitivity, and specificity from 30 training runs for each antimicrobial. Predictive models were generated to classify the resistance versus susceptibility profiles of 10 different antimicrobials (*x* axes). CFX, cefoxitin; CHL, chloramphenicol; CIP, ciprofloxacin; CLI, clindamycin; ERY, erythromycin; GEN, gentamicin; OXA, oxacillin; PEN, penicillin; SXT, trimethoprim-sulfamethoxazole; TET, tetracycline. (b) Hierarchical clustering of 673 isolates based on the 2,000 oxacillin-resistant genomic signatures (k-mers) recognized as the most significant by the trained classifier. Results have been data mined with respect to year of collection, ST, CC, type of sample, and source. (c) Hierarchically clustered heat map of the 10 antimicrobials based on the top 50 genes corresponding to the genomic features recognized as the most significant by the trained classifiers. Genes are color coded according to their function: resistance gene (red), virulence gene (blue), genes with function in horizontal gene transfer (HGT) (purple), genes with other functions (black). For each gene, the number of different k-mers present per gene per antibiotic model is shown on the plot, from a total of 2,000 recognized as the most significant by the trained classifier.

specificity, 99.79%; and AUC, 95.26%). Overall, 8 antibiotics achieved an AUC of >80%; chloramphenicol and penicillin did not. For oxacillin, the isolates clustered strongly by resistant/susceptible phenotype and showed strong correlation with clonal complexes despite population structure correction (Fig. 5b). Specifically, resistant isolates came from two

major clonal complexes, CC59 and CC9. CC59 was found associated with a diverse range of food sources collected over all years. Given the correlation of both oxacillin and cefoxitin to the *mecA* gene, we would expect a high degree of similarity between the features of these two models. Of the 2,000 features (k-mers) considered for each model, 1,927 (96.35%) were the same for oxacillin and cefoxitin models, and patterns of resistance seen for oxacillin were also observed in the cluster map for cefoxitin (see Data Set S1). Moreover, the cluster maps for oxacillin and cefoxitin showed a higher normalized k-mer frequency for samples belonging to CC59 than for other clonal complexes in the resistant samples. To the best of our knowledge, this difference in terms of k-mer frequency of CC59 samples was not previously observed in any other work, and it further indicates the importance of analyzing CC59 samples, especially in China. The cluster maps of the other eight antimicrobials (Data Set S1) did not show such strong clustering by susceptible/resistant phenotype. Notably, trimethoprim-sulfamethoxazole, tetracycline, and ciprofloxacin showed very similar patterns of resistance, with most samples being susceptible to these antimicrobials but with a small cluster of resistant samples present in all three models. The resistant isolates in this cluster were primarily associated with two clones and sample types, CC9 pork isolates and CC630 infectious human isolates.

Isolates exhibiting both trimethoprim-sulfamethoxazole and tetracycline resistance were also strongly correlated with MRSA, whereas ciprofloxacin resistance showed no association with MRSA. CC59 isolates, primarily susceptible to all three antimicrobials (trimethoprim-sulfamethoxazole, tetracycline, and ciprofloxacin) formed a large cluster on the trimethoprim-sulfamethoxazole and ciprofloxacin trees, while on the tetracycline tree, CC59 was more fragmented, showing that while there is clear correlation, there are important differences between these three models. Comparison of the important features of the three models showed an overlap of 79.9%. Both the erythromycin and clindamycin models showed two separate resistant clusters, one predominantly CC59 and one predominantly CC9. Other samples were also clustered by CCs but were mostly fragmented. Two antimicrobial models, gentamicin and chloramphenicol, had very imbalanced samples, with most isolates susceptible to these antimicrobials. In all these models, there was fragmented clustering by CC.

**Machine learning reveals robust prediction of known resistance genes correlated with phenotype and novel genetic determinants of AMR phenotype.** To better understand the relationship between antimicrobial resistance phenotype and genotype, we cross-referenced the 2,000 significant k-mers for each antibiotic model (10 of the 13 antibiotics) to the pan-genome of the 673 isolates (see Table S6) and summarized the genes found in Fig. 5c. For each antibiotic except penicillin, we obtained a list of potential genetic determinants of antibiotic resistance, with importance measured as the number of unique k-mers mapped back to the gene. For penicillin, the small number of susceptible samples impeded machine learning. For oxacillin and cefoxitin resistance, as expected, the *mecA* gene previously recognized as conferring resistance (65–67) was the primary gene found by machine learning, with 100 and 98 different k-mers, respectively, mapping back to this gene. The genes *maoC* and *ugpQ*, previously shown to be SCC*mec*-associated elements (68), were also found to be highly discriminant between resistant and susceptible cefoxitin and oxacillin phenotypes. Interestingly, *ugpQ* was also found to be in the SCC*mec*s of 342 of the 343 MRSA strains, and *maoC* was located in the SCC*mec*s of 332 isolates (see Table S7). The identification of genes known and expected to be correlated with the selected resistance phenotypes indicates the robustness of the methods employed, as stressed by Jaillard et al. (69). The insertion sequence IS*Sau3* (IS*1182* family) was found to be highly predictive for both antibiotics. This insertion sequence has been reported to be close to the SCC*mec* complex but has also been reported to inactivate the gene *lytH*, increasing resistance (70). Chloramphenicol resistance was highly associated with the known resistance gene *cat* (77 k-mers). The chloramphenicol resistance gene *cat* and the plasmid replication gene *rep* (also correlated with chloramphenicol resistance) were found to cooccur in 63 isolates from both animal and nonanimal food sources but were not found in any human isolates. *rep* is an initiator protein in pT181 family plasmids, including pC221, known to typically carry chloramphenicol resistance (71), indicating the likely presence of this plasmid in these food isolates.

Another gene, encoding tryptophan decarboxylase, was also significant for chloramphenicol resistance. This gene is a promising gene candidate as despite no previous known links to chloramphenicol resistance, the tryptophan biosynthesis pathway was previously associated with vancomycin (72). Interestingly, resistance to trimethoprim-sulfamethoxazole was highly associated with the gene *tcaA*. Inactivation of this gene was previously shown to increase resistance to teicoplanin and vancomycin (73), but no link has been found to trimethoprim-sulfamethoxazole. This would benefit from further experimental validation. Ciprofloxacin and tetracycline were primarily related to genes involved in horizontal gene transfer (IS*256* and IS*Bli29* [IS*NCY* family]) (Fig. 5c), suggesting these resistances may be plasmid mediated. Resistance to ciprofloxacin is typically caused by point mutation in the chromosomal *gyrA* and *parC* genes; however, there is growing evidence of plasmid-mediated resistance (74). This likely indicates a prevalence of plasmid-mediated quinolone resistance in food-related *S. aureus* isolates in China, as was previously reported in a smaller-scale study in *Escherichia coli* from farmed fish (75). Transposases, IS*Sau3* (IS*1182* family), IS*Bli29* (IS*NCY* family), and IS*256*, and the antimicrobial resistance genes *aacA-aphD* are present in many but not all isolates in our cohort from both human and animal food sources, suggesting the widespread prevalence of these genes correlated with multiple resistances in China.

The machine learning approach also revealed the presence of significant associations between virulence genes and the antimicrobial resistance profiles. Several virulence genes (e.g., *lpl2*, *essG*, *splF*, *sdrE*, *map*, and *ssl*, and others) (Table S6) were found to be associated with resistant phenotypes. Specifically, *lpl2*, a host invasion gene, was found to be a discriminant genetic feature of multiple resistant phenotypes (cefoxitin, oxacillin, ciprofloxacin, gentamicin, clindamycin, tetracycline, and erythromycin). The genes for type VII secretion (*essG*), serine protease (*splF*), serine-aspartate repeat protein (*sdrE*), and adhesin (*map*) were correlated with cefoxitin, clindamycin, erythromycin, gentamicin, and oxacillin resistance. Finally, toxin gene *ssl7* was correlated with cefoxitin, ciprofloxacin, clindamycin, erythromycin, gentamicin, oxacillin, trimethoprim-sulfamethoxazole, and tetracycline. Other virulence genes also found to be associated with antimicrobial resistance (AMR) phenotypes can be seen in Fig. 5c and Table S6.

## DISCUSSION

We have analyzed a large variety of *S. aureus* samples over a 9-year temporal window collected across China (27 provinces), considering both food and human samples. In our study, we identified 29 novel sequence types (27 in food, 2 in human) with no genomic sequence available in public databases. Most *S. aureus* sequences available so far are of clinically relevant strains, and a big gap exists for less-pathogenic ones; thus, the new STs can be of relevance for further epidemiological studies leading to a better understanding of emergence, reemergence, and spread of *S. aureus* diseases. Our refined analysis of the type IV SCC*mec* highlights the presence of further resistances and plasmid insertions within this short cassette. Variants of SCC*mec* IV hosting further metal (cadmium) and antimicrobial resistance genes (kanamycin and bleomycin), and of a novel SCC*mec* type identified in this study, provide further avenues of investigation into the epidemiology of MRSA. These findings suggested that the SCC*mec* and contained resistance determinants might be transmitted via horizontal gene transfer and thus have a separate epidemiology with respect to the rest of the genomes (76, 77). Altogether, our observations shed more light on the complexity of *S. aureus* epidemiology and on the need for surveillance of the MDR and cross-host *S. aureus* to clarify the dissemination route and avoid the spread of specific genomic traits. This data set can be easily meta-analyzed and integrated with further studies, and this clearly could lead to a deeper and further understanding of the epidemiology of this bacterium and how to prevent and treat resistant infections, especially from an important area such as China.

Using a comparative genomics pipeline inspired by Manara et al. (35), we considered our isolates in the context of other similar studies. Across all isolates in our study, 97% of the *S. aureus* isolates showed resistance to at least one antimicrobial, and 72%

showed MDR, which was consistent with previous reports in China (>94% showed resistance to at least one antibiotic and >58% were MDR in food-based isolates) (78). The number of isolates carrying resistance to at least one antimicrobial found in this study was higher than in other food-based reports from Brazil (83%) (79), South Africa (71%) (80), and South Korea (51%) (81). Analogously, the rate of MDR in our study was higher than in India (53%) (82), South Korea (35%) (81), the United States (10%) (83), and Brazil (8%) (79). The recent surveillance study in Europe (53) revealed a high level of resistance among human-associated *S. aureus*, with 90% of isolates showing resistance to at least one antibiotic and 45% showing MDR, which is in accordance with our human data (84% showed resistance to at least one antibiotic and 31% were MDR).

CC59 and CC5 were the most predominant clones in this study, which agrees with those previously reported in Asia (22, 23). CC5 was also the most abundant CC type in a pan-European study of invasive *S. aureus* infections (53). Data from foodborne disease outbreaks and investigations showed that both CC59 and CC5 are common epidemic clones that cause staphylococcal food poisoning and infectious diseases (84–86). Our food isolates also showed a prevalence of SCC*mec* type IV (49.1%) and type V (25.4%), similar to previous findings in both China (78) and Germany (87). Additionally, SCC*mec* XII was also found in 63 of our food isolates. SCC*mec* XII mainly spreads as LA-MRSA in China (44) and several Asian countries, such as Japan, Malaysia, and Thailand (88–90). Recently, MRSA with SCC*mec* XII has been reported to lead to clinical infections, suggesting a potential pathogenic risk for humans (91). In our human isolates, SCC*mec* III was prevalent (9/14), which is similar to other hospital-based studies in China (92, 93), whereas SCC*mec* IV was prevalent in studies from Europe (35, 94). The *spa* types in our cohort showed a different distribution to those in another Chinese food-based study (78), with t437 most prevalent in ours compared to t071 and t091 in the study by Liao et al. (78), although all of the most prevalent isolates in that study were also present in our cohort. In an Italian study (35), *spa* types t001 (4.9%), t002 (4.3%), t008 (3.3%), and t127 (2.7%) were prevalent. While t002 (5.8%) and t127 (5.8%) were present in our human isolates, t001 and t008 were absent and *spa* types t164 (11.6%), t189 (7.2%), t037 (6.5%), and t085 (6.5%) were the most prevalent. Our cohort has a low level of PVL+ human isolates (2.17%) compared to that in other studies in China (95) (12.8%), Europe (35) (27.4%), and Africa (96) (17% to 80%).

A comparison of the virulence factors present in our Chinese cohort with those of the Italian study by Manara et al. (35) shows many similarities, despite the differences in isolate hosts. Iron uptake, conserved antigen, and arginine catabolic mobile element (ACME) virulence factors found in the study by Manara et al. (35) were not present in our cohort. However, genes for immune evasion, toxins, ion transporter, adhesins, capsular polysaccharides, and serine proteases were found in similar proportions of isolates in both cohorts, though specific gene presence varied. Several immune evasion genes in our cohort are present in almost all isolates, including *esxA/B/C/D*, *hld*, and *sbi*, as found by Manara et al. (35). Additionally, the toxic shock syndrome toxin gene *tsst-1* associated with ST22 in the study by Manara et al. (35) was present in 28 of our isolates; however, in our cohort, this toxin was associated with ST1 ($n = 17$, $P < 0.0001$) and ST30 ($n = 4$, $P > 0.0001$). The presence of these virulence factors increases the risk of isolates carrying these factors harming human health, and so the widespread existence of these in food-related isolates needs to be monitored.

The CC59-t437-SCC*mec* IV/V, CC9-t899-SCC*mec* XII, and CC398-t011-SCC*mec* V clones were the most frequent food-associated MRSA clones in our study. It has been well documented that CC59-t437-SCC*mec* IV/V clones are major CA-MRSA clones in China and other Asian countries, threatening a vast population due to their epidemiological potential (22). However, they remain geographically confined and are seen in only low numbers in Europe (35, 53), suggesting importation to Europe from Asia, as was noted previously (97). The considerable numbers of this CA-MRSA lineage from food in this study could indicate some human contamination, possibly a result of inadequate hygiene measures and improper handling of food (98). Additionally, most of

mSystems®

the PVL-positive isolates (86.8%) belonged to CC59, demonstrating the pathogenic potential of these MRSA strains. CC9, the predominant LA-MRSA clone in Asia, was mostly identified from retail meat samples in this study. Elsewhere, CC398 is the prevalent LA-MRSA strain found in animals and humans across European countries and North America (23). Furthermore, in addition to meat, CC398-t011-SCC*mec* V MRSA isolates were detected from a greater variety of food samples, including nonanimal food such as cakes, noodles, fruits, and vegetables, suggesting a different epidemiological characteristic of these CC398 MRSA strains in China. Recently, this CC398-t011-SCC*mec* V MRSA clone was detected from humans with nonanimal contact in China and denominated as CA-MRSA (99). Thus, continued monitoring of this strain's epidemiology and preventing its widespread transmission are essential.

MRSA belonging to CC5, CC22, and CC30 clones are major HA-MRSA clones in Europe, according to a previous pan-European study (53). In our study, while all three clones are present in human samples, they are in low numbers (CC5, $n = 19$; CC22, $n = 4$; CC30, $n = 1$), and only one of these isolates was MRSA (CC5), highlighting the importance of geographical isolation in MRSA dissemination. Moreover, an Italian study on human-associated *S. aureus* in a pediatric hospital showed that CC5, CC22, CC8, CC1, and CC121 were the prevalent clones and that all CC1 and most CC5 (>96%) isolates were MRSA (35). In contrast, in our study, more than 95% of the CC1 and CC5 isolates were MSSA. In addition, for our human MRSA isolates, the major SCC*mec* type was SCC*mec* III (9/14 [64%]), while SCC*mec* IV (54/83 [67%]) was frequently detected by an Italian study and SCC*mec* III was not detected in any samples (35). Similarly, Aanensen et al. (53) only detected SCC*mec* III in clones from ST239, and the authors suggested these were likely to have been largely imported from Asia, with SCC*mec* IV again the most prevalent type in the pan-European study. However, the presence of PVL-positive isolates was strongly associated with MSSA in our study, which agreed with the Italian results (35).

CC630 has been circulating in Asia since the 1970s and is the prevalent HA-MRSA clone (22). Correspondingly, in our study, the CC630 (ST239-t037-SCC*mec* III) clone was the predominant MRSA clone among human samples (9/14), with the other isolates being ST59 (3), ST1 (1), and ST5 (1). While these are not prevalent clones in Europe, both have been found in clinical isolates. ST239 clones were present in 8 isolates in the pan-European collection (53), and ST59 was present in 3 of 86 MRSA isolates in the study by Manara et al. (35). As similarly reported by Aanensen et al. (53), the human ST239 isolates in our study carried the virulence gene *sasX*, reflecting the widespread distribution of this gene in ST239 clones in China. However, the two food-associated ST239 isolates did not carry this gene and clustered away from the other ST239 isolates in our study in the phylogenetic tree, possibly indicating that lineages not carrying this gene may be circulating in food. Additionally, other MRSA lineages, for instance, ST88-SCC*mec* IV/V, which is already known as the "African clone" (96), were also frequently detected in this study. This finding indicates that the MRSA clones have spread cross-regionally.

As in a limited number of previous phylogenetic studies, the ST9, ST59, ST239, and ST398 (58, 78) clones from our cohort tended to form separate clusters from the reference group, hinting at geographical differentiation. Meanwhile, Bayesian divergence analysis pointed to a more recent clonal evolution of MRSA in China compared to that in other countries (62, 63), potentially driven by increased economic growth and antimicrobial usage in China (100, 101). Bayesian divergence analysis also showed that while the ST9 and ST398 MRSA isolates in China exhibited relatively independent phylogenetic evolutionary relationships compared to those of isolates from other countries, there were signs of mixing of strains, possibly linked to the importation of meat products. ST239 MRSA in China showed a more recent phylogenetic relationship with isolates from other Asian countries, suggesting an introduction of ST239 MRSA to China from neighbors (64).

Several approaches to analyzing whole-genome sequences (WGS) against resistance phenotypes were previously published in the literature. These fall into two main

branches, machine learning approaches, as we have implemented here, and genome-wide association studies, a statistical association-based approach. For *S. aureus*, several groups have attempted to link resistance phenotypes to genes (53, 69, 102, 103). For example, Aanensen et al. (53) used published resistance genes, with manual curation, to predict antibiotic resistance phenotypes from genotypes with high accuracy. Although their findings provided an *in silico* typing method comparable with phenotypic accuracy, our machine-learning based approach allows impartial identification of genetic determinants of resistance, and is not limited to those already known. This is particularly important for China, where the differential evolution may have allowed resistance determinants to arise that are not as well represented in the public databases. Machine learning (ML) offers a powerful opportunity to analyze entire genomes quickly and efficiently against selected phenotypes, allowing for the identification of arbitrary numbers of sequences and other genomic features ranked on strength of correlation with the phenotype. Sequences identified by ML may contain genes with a known functional relationship with the phenotype as well as genes with no previously known association with that specific phenotype, thus providing a significant advantage to conventional bioinformatics methods based on checking for presence/absence of known manually chosen genes. Here, we have shown, for a large number of antibiotics, genes that are significantly and predictively associated with antibiotic resistance phenotypes in isolates evolved in China. Moreover, to the best of our knowledge, for the first time, we have shown that CC59 isolates have a higher k-mer frequency (for most of the studied k-mers) than other resistant clonal complexes for oxacillin and cefoxitin. This result further indicates the importance of CC59 samples, especially in China, and demonstrates the difference between this clonal complex and the other resistant clonal complexes available in our data set.

In agreement with previous studies in China (104) all our cohort isolates were susceptible to vancomycin and linezolid, while large numbers of isolates were resistant to widely used antibiotics such as penicillin, erythromycin, and cefoxitin. Thanks to ML, we were able to identify genes which, individually or in patterns, featured a strong correlation with resistance to multiple antimicrobials, regardless of the source of the isolate. As an example of the robustness of the methodology, the *maoC* and *ugpQ* genes previously found to be SCC*mec*-associated elements, in addition to *mecA*, were found strongly correlated with cefoxitin and oxacillin resistance. The identification of genes known and expected to be correlated with the selected resistant phenotypes indicates the robustness of the methods employed, as stressed by Jaillard et al. (69). ML also revealed a correlation between IS*Sau3* (IS*1182* family) and cefoxitin and oxacillin. This insertion sequence has been reported to be close to the SCC*mec* element and has also been reported to inactivate the gene *lytH*, increasing resistance (70). The *rep* gene is an initiator protein in pT181 family plasmids, including pC221, known to typically carry chloramphenicol resistance (71), indicating the likely presence of this plasmid in these food isolates. Another gene encoding tryptophan decarboxylase was also found to be significant for chloramphenicol resistance (71). Although not previously linked to chloramphenicol resistance, this gene is a promising gene candidate, as the tryptophan biosynthesis pathway has been previously associated with vancomycin resistance (72). The *tcaA* gene was found to be associated with resistance to trimethoprim-sulfamethoxazole. Inactivation of this gene was previously shown to increase resistance to teicoplanin and vancomycin, but no link has been found to trimethoprim-sulfamethoxazole. This would benefit from further experimental validation. Additionally, resistance to ciprofloxacin is typically caused by point mutations in the chromosomal *gyrA* and *parC* genes; however, there is growing evidence of plasmid-mediated resistance. In this study, the ML result showed that ciprofloxacin resistance was linked to several insertion sequences, suggesting a potential rapid spread of resistance among isolates in food and humans. Notably, using ML, significant correlations between virulence genes (*lpl2*, *essG*, *splF*, *sdrE*, *map*, and *ssl7*, plus others less strongly associated) and antimicrobial resistance phenotypes were found. The correlation between the presence of

antibiotic resistance genes (ARGs) and virulence factors was observed previously (105), and it has been proposed that an increase in virulence allows the bacteria to overcome the fitness costs associated with the carriage of AMR genes (106, 107).

Other published papers have applied machine learning to identify AMR genes associated with resistance phenotypes; however, there are many differences that make this work unique. Hyun et al. (102) employed an SVM approach to identify genes from single nucleotide polymorphisms (SNPs) based on the pan-genome of 288 *S. aureus* isolates. These isolates, all taken from human sources, were isolated in Singapore, the United States, and Russia and were primarily composed of ST239, ST22 and ST5. Hence, our study is very different; this is also highlighted by the fact that only 5 of the 13 gene candidates identified by Hyun et al. (102) (*hylX* linked to erythromycin, *oppD* and the gene for acyl coenzyme A [Acyl-CoA] linked to gentamicin, and the gene for P-type ATPase and *rep* linked to tetracycline) were also found to be important in our primarily food-based work. In addition, in our study, as described above, novel determinants underlying resistance were found strongly associated with the AMR phenotypes, such as *tcaA*, the gene for tryptophan decarboxylase, and IS*Sau3* (IS*1182* family), an original finding with respect to the literature. Both Jaillard et al. (69) and Wheeler et al. (103) used genome-wide association studies to make predictions about *S. aureus* phenotypes from the whole-genome sequences. Both studies used isolates from U.K.-based humans, giving a very different population structure to that in this study. Jaillard et al. (69) made predictions against four of the antibiotics used in our study. They also tested methicillin, which we did not; however, we tested oxacillin, which is chemically very similar and has replaced methicillin in clinical use. For these antibiotics, the predicted genes associated with the resistance phenotype for methicillin (*mecA*), erythromycin (*ermC*), and one of the 2 gentamicin genes (*aac*) were in common with our work. While two genes predicted to correlate to trimethoprim resistance in the work by Jaillard et al. (69) (*ybaK* and *mqo*1) were associated with different antibiotics in our study (*ybaK* with cefoxitin, erythromycin, gentamicin, and oxacillin and *mqo1* with cefoxitin, clindamycin, erythromycin, gentamicin, and oxacillin). This may be because, in our study, we tested trimethoprim in conjunction with sulfamethoxazole, as it is generally used clinically, rather than trimethoprim alone, as tested by Jaillard et al. (69). In the case of the study by Wheeler et al. (103), six antibiotics overlapped with our study (gentamicin, oxacillin/methicillin, erythromycin, tetracycline, ciprofloxacin, and clindamycin), and the genes found to be associated with resistance for five of these were also found by us (*aacA-aphD* linked to gentamicin, *mecA* linked to oxacillin/methicillin, *ermA* and *ermC* linked to erythromycin, *ermA* linked to clindamycin, and *tetK* and *tetM* linked to tetracycline). However, differences were found in the case of ciprofloxacin, where distinct genes were identified in the two studies (chromosomal genes *gyrA* and *parC* were significant in the study by Wheeler et al. [103], while insertion element IS*256* was found in this study). Discrepancies may reflect different resistance mechanisms, with this study indicating a prevalence of plasmid-mediated ciprofloxacin resistance in China, and the study by Wheeler et al. [103] suggesting a prevalence of chromosomal resistance mechanisms in their UK-based study.

The strong overlap between our results and previous works in identifying genes known to be correlated with the selected resistance phenotypes indicates the robustness of the methods we have employed and gives confidence to the novel predicted genes that have arisen from our analysis. Our more heterogenous data set results in a slightly lower machine learning accuracy than that in other published works, 76.6% to 95.5% compared to 91.9% to 98.6% (Hyun et al. [102]) and 94.7% to 100% (Wheeler et al. [103]). Furthermore, we have also, for the first time, found a genetic determinant of AMR for chloramphenicol and cefoxitin in *S. aureus* by using machine learning. Our study has been able to assess the differentially evolved Chinese clones of *S. aureus* and show that while many resistance mechanisms align with those seen elsewhere globally, differences may have evolved. However, we would like to point out that approaches for AMR prediction models, which have already been developed from genome sequence collections

of *S. aureus* and also for many other species, are often designed to maximize accuracy in predicting AMR phenotypes, emphasizing their diagnostic capabilities over their capacity to uncover genetic mechanisms for resistance. Many such models are also based on the detection of genes from a curated set of known AMR determinants, rendering them difficult to generalize to different treatments or organisms and unsuitable for discovering novel genes or interactions that drive resistance. In our case, we did not use known AMR determinants to search but rather the whole-genome information obtained by mapping back the k-mers on the pan-genome. Based on the concept of One Health, our study emphasizes the importance of a holistic working approach for food, human, food animals, and related sectors. The strong set of potential gene candidates identified in this work could provide new avenues of research to tackle the significant threat posed by antibiotic resistance in this part of the world.

## MATERIALS AND METHODS

**Sample collection and bacterial isolation.** A total of 7,937 food-associated *S. aureus* isolates were cultured from various foods from 27 provinces during the years 2010 to 2018 in China. In addition, 142 *S. aureus* human-associated isolates, including 53 healthy and 89 infected human-associated isolates collected in Shanghai between 2015 and 2017, were employed in this study. All *S. aureus* isolates were confirmed using Vitek 2 Compact (bioMérieux, Craponne, France) and then were screened for MRSA by amplifying the *mecA* gene. In total, 343 food-associated isolates and 18 human-associated isolates were identified as MRSA. These MRSA isolates together with all human-associated MSSA isolates and 250 food-associated MSSA isolates (randomly selected from the full collection), giving 735 isolates in total, were used for the study of the population structure and the molecular epidemiology (see Table S1 in the supplemental material) and sent for genome sequencing. The identified isolates were stored in brain heart infusion broth with 40% (vol/vol) glycerol (HopeBio, Qingdao, China) at −80°C for the following analysis.

**Antimicrobial susceptibility testing.** The MICs of 13 drugs were determined using the broth dilution method by the Biofosun Gram-positive panels (Shanghai Biofosun Biotech, China) according to the CLSI guidelines (108). The panel of antimicrobial compounds tested included penicillin (PEN) (0.06 μg/ml to 8 μg/ml), oxacillin (OXA) (0.25 μg/ml to 16 μg/ml), cefoxitin (CFX) (0.25 μg/ml to 16 μg/ml), vancomycin (VAN) (0.5 μg/ml to 32 μg/ml), daptomycin (DAP) (0.125 μg/ml to 8 μg/ml), erythromycin (ERY) (0.125 μg/ml to 8 μg/ml), gentamicin (GEN) (0.5 μg/ml to 64 μg/ml), tetracycline (TET) (0.5 μg/ml-32 μg/ml), ciprofloxacin (CIP) (0.125 μg/ml to 8 μg/ml), clindamycin (CLI) (0.125 μg/ml to 8 μg/ml), trimethoprim-sulfamethoxazole (SXT) (0.125/2.3 μg/ml to 8/152 μg/ml), chloramphenicol (CHL) (1 μg/ml to 64 μg/ml), and linezolid (LZD) (0.25 μg/ml to 16 μg/ml). *S. aureus* ATCC 29213 was used as the control for the antimicrobial susceptibility testing. As no resistant breakpoint has been officially established for DAP, the resistance term (DAP-R) was assigned to strains with an MIC of >1 μg/ml in the present study to simplify the understanding (109).

**DNA purification and extraction.** Each isolate was grown in brain heart infusion (BHI) broth (HopeBio, Qingdao, China) at 37°C, and genomic DNA (gDNA) was purified using an Omega EZNA Bacterial DNA kit (Omega Bio-Tek, GA, USA). Genomic DNA was extracted with the sodium chloride-Tris-EDTA (STE) methods. The harvested DNA was detected by agarose gel electrophoresis and quantified by a Qubit 2.0 fluorometer (Thermo Fisher Scientific, USA).

**Library construction and whole-genome sequencing.** A total amount of 1 μg DNA per sample was used as input material for the DNA sample preparations. Sequencing libraries were generated using NEBNext Ultra DNA library prep kit for Illumina (NEB, USA) according to the manufacturer's recommendations, and index codes were added to attribute sequences to each sample. Briefly, the DNA sample was fragmented by sonication to a size of 350 bp, and then DNA fragments were end polished, A tailed, and ligated with the full-length adaptor for Illumina sequencing with further PCR amplification. Finally, PCR products were purified (AMPure XP system), and libraries were analyzed for size distribution by an Agilent 2100 Bioanalyzer and quantified using real-time PCR. The resultant DNA preps were sequenced using Illumina NovaSeq PE150 at Beijing Novogene Bioinformatics Technology Co., Ltd.

**Genome assembly and annotation.** All sequences were preprocessed through readfq v10 (110). To clean the data, reads containing low-quality bases (mass value ≤ 20) over 40% were removed. Those with an N in reads beyond 10% were removed. The reads which overlapped with the adapter, which exceeded 15 bp and had less than 3 mismatches between them, were removed. Clean data were processed for genome assembly with SPAdes v3.13 (111), and QUAST v4.5 (112) was used for assessing the contigs through assembly. The contigs with a length shorter than 1,000 nucleotides were filtered out. The completeness and contamination of genomes were assessed through checkM (113) with the lineage_wf pipeline. We then obtained 673 high-quality *S. aureus* genomes ($N_{50} > 50,000$) which were used for further analysis. Genomes were annotated with Prokka v1.14.5 (114) using default parameters with -addgenes -usegenus.

**In silico subtyping identification.** Sequence types were identified through MLST, which mapped the sequences to the PubMLST *Staphylococcus aureus* MLST database (115). Novel STs all showed one or more single base mutations from known STs and were further confirmed by PCR, according to the same protocol given on the PubMLST website (https://pubmlst.org/organisms/staphylococcus-aureus/primers),

and by DNA sequencing before being submitted to PubMLST for verification (115). Clonal complexes (CC) were annotated through the geoBURST Full MST algorithm using phyloviz software (116) with a primary founder surrounded by single-locus variants (SLVs) and known CC type in the MLST database. Spa protein A repeat region was identified through spaTyper (117) for *S. aureus* sequences. SCC*mec*Finder (118) was used for typing SCC*mec*, and the contigs that aligned to the SCC*mec* region were further annotated with Prokkav1.14.5 (114). The names of predicted proteins from Prokka were further confirmed with BLASTp (119). The variant SCC*mec* regions were further aligned using BLAST to identify SCC*mec* variants and plasmid integration into the SCC*mec* element (120, 121). SCC*mec* variants in isolates 11A1151 and 18A25 (region encompassing *OrfX* [*rlmH*] to the direct repeat [DR] sequence GAAGCTTATCATAAGTAA) were aligned to MF062 (GenBank GCA_003240235) and N315 (GenBank D81934.2) using default BLASTn settings (119). Pairwise comparison using BLASTn with default settings was performed for the SCC*mec* IVa variant from isolates 11A832 and 18A245 (region encompassing *OrfX* [*rlmH*] to the DR sequence GAAGCGTATCATAAATGA) against the NCBI nonredundant database using the SCC*mec* region and/or only the J3 region of the cassette, using default BLASTn settings.

SCC*mec* genes were visualized using the gggenes and ggplot packages in R (122). The accessory gene regulator (*agr*) subgroup was searched through BLAST using MyDbFinder 1.0 (https://cge.cbs.dtu.dk/services/MyDbFinder/) with setting parameters of -identity 90 -coverage 90. GenBank accession numbers AFS50129.1, AFS50128.1, AFS50130.1, and AFS50131.1 were used as reference sequences for *agr* type I to IV, respectively, in this study.

**Virulence factors and ARG analysis.** Virulence factors and ARGs were searched through Abricate software (https://github.com/tseemann/abricate) using the VFDB (123) database set B and the CARD database (124). The BLAST search within Abricate was conducted with parameters of >90% identity and >75% coverage (proportion of gene covered), in agreement with previous studies (35). Virulence gene functional categories were assigned manually based on the gene entry in the VFDB database (123). All results were manually curated by careful search of the literature. In addition, to specifically target the *fnbA/B* genes known to generate false negatives due to isoforms not present in the public databases (125), we performed an additional BLAST search, using as queries the different isoforms as described by Loughman et al. (125) and selecting the best hit entries. To identify the presence of the virulence factor *sasX* within each of our isolates, a BLASTn search was used with *sasX* (GenBank MH143577.1) as the query. Thresholds of 90% identity and 90% coverage were used.

**Core gene alignment and phylogenetic analysis.** All annotated files were taken as input for pangenome analysis with core gene alignments through Roary v3.13 (126). IQTree v2.0.3 (127) was then used to construct the phylogenetic trees from the core genome alignment with the general time reversible (GTR) (+F+R10) replacement model. In addition, core genome maximum likelihood phylogenetic trees clustered by different STs were also constructed with all available reference genomes downloaded from the PATRIC database (45). For ST398, only a subset of reference genomes, with collection dates, were used due to the large number of isolates available. The phylogenetic trees were subsequently visualized through iTOLv4 (128).

**Bayesian divergence estimates.** A subset of sequences from this study and PATRIC from ST9-t899-SCC*mec* XII, ST59-t437-SCC*mec* IV/V, ST239-t037-SCC*mec* III, and ST398-t011-SCC *mec*V in our cohort, alongside reference genomes publicly available (45) for the same lineages for which collection dates were available, were selected for Bayesian evolutionary analysis using BEAST v 1.10.4 (129). As only two reference genomes belonging to ST59-t437-SCC*mec* IV/V were available, the other 11 ST59-SCC*mec* IV/V reference genomes were also recruited to the analysis data set. Analysis was conducted on a core genome alignment of each lineage by using Roary v3.11.2 (126). All combinations of three clock models (strict, uncorrelated log normal, and uncorrelated exponential) and four tree priors (constant coalescent, logistic growth, Bayesian skyline, and birth-death model) were tested using stepping stone sampling on a subset of the isolates (ST239 isolates) to identify the best model. Log marginal likelihood values were in the range of −2,459,423 to −2,459,156. The best model was a random uncorrelated exponential clock model, with a Bayesian skyline growth model. The GTR-gamma nucleotide substitution model was used, as selected for the maximum likelihood tree. For ST59, the model did not converge after 250 million steps, and so instead, a simpler model was used (strict clock and constant coalescent growth). The analysis was run for 2 independent chains until the effective sample size (ESS), that is, the effective number of independent draws from the posterior distribution, for all parameters was greater than 200 per chain. This entailed each chain running for approximately 150 million steps. Convergence was assessed in Tracer v1.7.1 (130), and chains were subsequently combined using LogCombiner v1.10.4 (131). The maximum clade credibility tree was selected using TreeAnnotator v1.10.4 (131) and then visualized in iTOL v5 (128).

**Machine learning.** Machine learning was used to find features of the sample genomes that could be used to predict resistance to the panel of 13 antimicrobials. Sample genomes were first split into overlapping k-mers of 13 bp in length using GenomeTester (132) to produce a feature table for all samples. Taking each antimicrobial individually, the AMR phenotype of susceptible or resistant (Table S1) was used as the class label, with intermediate phenotypes neglected. As the classes were unbalanced, a synthetic minority oversampling technique (SMOTE) was applied to oversample data of minority class, compensating for unbalanced classes (133). The number of splits in the nested cross-validation and the number of k-nearest neighbors (default values of 5 neighbors) for SMOTE necessitated a minimum number of 12 samples in the minority class. From the panel of 13 antimicrobials, three (daptomycin, linezolid, and vancomycin) had an insufficient number of resistant isolates (minority class) to train and cross-validate the machine learning model, leaving 10 trained models. The Python package Scikit-learn (134) was used to reduce the number of features used. To correct for bias, the clonal the population structure was filtered according to k-mers based on weighted pairwise chi-squared tests between each feature and

the phenotype class, as suggested by Aun et al. 2018 (135). The weights of each genome were calculated using the method of Gerstein, Sonnhammer, and Chothia (136). Subsequently, another k-mer filtering was performed, and the top 2,000 features (k-mers) were selected with the highest chi-squared statistic ($P$ value lower than 0.00000000000001, confidence value of more than 99.9999999%). The weighting was based on a distance matrix based on mash distances (135). A panel of machine learning algorithms was then run in the Scikit-learn package (134): logistic regression (LR), linear support vector machine (SVM), and radial basis function (RBF) SVM.

Nested cross-validation (NCV) was employed to assess the performance and select the hyperparameters of the proposed classifiers. The inner loop of the NCV found the best hyperparameters of each classifier using stratified 3-fold cross-validation; the outer loop measured the performance metrics using 5-fold stratified cross-validation. Each algorithm was run 30 times and metrics were collected for each run. The mean and standard deviation (SD) from the 30 iterations was then used as the final result statistic. The following prediction metrics were plotted using Seaborn: accuracy (true positive [TP] + true negative [TN]/[P + N]), sensitivity (true positive rate, TP/P), specificity (true negative rate: TN/N), and area under the receiver operating characteristic curve (AUC).

**Identification of AMR virulence and HGT genes.** Where the machine learning was able to predict the antimicrobial class based on k-mers, these were then used to search the genome for genes that contained the k-mers. Using the pan-genome of the study isolates, annotated using Prokka, the k-mers were mapped to genomes using a BLASTn query with the following parameters: evalue, 1,000; word_size, 13; gapopen, 5; gapextend, 2; outfmt, 5; strand, "plus." Genes with an identity of >70% and coverage of >70% were considered to be variants of the same gene and hence were discounted as duplicates, as done in previous literature (137); however, a more stringent threshold was used to ensure all gene variants were accounted for. The k-mer hit count (how many k-mers mapped to each identified gene) of the genes identified was then assessed for statistical significance at a significance level of 0.05 using a binomial exact test, with the probability of a gene hit based on the length of the gene and number of k-mer combinations possible per gene. All genes found were checked in the published literature to find previous associations with AMR, virulence, or horizontal gene transfer (HGT). A clustered heat map was produced using the Seaborn package in Python showing the number of k-mers mapped to each gene per antibiotic.

**Data availability.** Short-read sequence data for all 673 isolates used in this study are deposited in the NCBI SRA and can be found associated with BioProject PRJNA633996. The code used in this study is available in the following GitHub repository: https://github.com/tan0101/Saureus-mSytems-2021.

## SUPPLEMENTAL MATERIAL

Supplemental material is available online only.
**DATA SET S1**, XLSX file, 18.1 MB.
**FIG S1**, TIF file, 0.2 MB.
**FIG S2**, TIF file, 2.6 MB.
**TABLE S1**, XLSX file, 0.5 MB.
**TABLE S2**, XLSX file, 0.1 MB.
**TABLE S3**, XLSX file, 0.1 MB.
**TABLE S4**, XLSX file, 0.1 MB.
**TABLE S5**, XLSX file, 0.1 MB.
**TABLE S6**, XLSX file, 0.1 MB.
**TABLE S7**, XLSX file, 0.1 MB.

## ACKNOWLEDGMENTS

This work was supported by the Ministry of Science and Technology of People's Republic of China under Grant Key Project of International Scientific and Technological Innovation Cooperation Between Governments (number 2018YFE0101500) and the InnovateUK grant (104986), FARMWATCH: Fight AbR with Machine learning and a Wide Array of sensing TeCHnologies.

We thank the University of Nottingham Research Beacon of Excellence: Future Food and Green Chemicals for support.

Conceptualization, Wei Wang, Jin Xu, Dajin Yang, Fengqin Li, and Tania Dottorini; Methodology, Wei Wang, Jin Xu, Dajin Yang, Fengqin Li, and Tania Dottorini; Supervision, Jin Xu, Fengqin Li, and Tania Dottorini; Writing, Editing, & Reviewing the Draft, Wei Wang, Jin Xu, Michelle Baker, Yue Hu, Fengqin Li, and Tania Dottorini; Investigation, Hui Li, Shaofei Yan, Menghan Li, Yao Bai, Yinping Dong, Zixin Peng, and Jinjing Ma; Formal Analysis and Visualization, Wei Wang, Michelle Baker, Yue Hu, Alexandre Maciel-Guerra, Ning Xue, and Menghan Li; Funding Acquisition, Fengqin Li, Zixin Peng, and Tania Dottorini.

We declare no competing interests.

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
