## [Reviewer comments · mSystems]

Whole-genome sequencing and machine learning analysis of *Staphylococcus aureus* from multiple heterogeneous sources in China reveals common genetic traits of antimicrobial resistance

WEI WANG, Michelle Baker, Yue Hu, Jin Xu, Dajin Yang, Alexandre Guerra, Ning Xue, Hui Li, Shaofei Yan, Menghan Li, Yao Bai, Yinping Dong, Zixin Peng, Jinjing Ma, Fengqin Li, and Tania Dottorini

Corresponding Author(s): Fengqin Li, China National Center for Food Safety Risk Assessment

Review Timeline:

Submission Date:	November 12, 2020
Editorial Decision:	December 21, 2020
Revision Received:	April 1, 2021
Accepted:	May 10, 2021

Editor: Christopher Marshall

Reviewer(s): The reviewers have opted to remain anonymous.

Transaction Report:

DOI: <https://doi.org/10.1128/mSystems.01185-20>

Dear Dr. Christopher Marshall,

We thank both the Editor and the Reviewers for their useful comments, which we have addressed below and in the revised manuscript. In the following, a point-by-point response to all the questions and comments is provided. The original questions are in black, our replies in blue.

Editors' comments

I have received the reviews of your manuscript. While your paper adds valuable data to the field, the reviewers stated several concerns about your study and did not recommend publication in mSystems in its current form. In particular, please note the lack of scholarship pertaining to genome-based predictions of antibiotic resistance and the need for a more robust machine learning approach.

As you know, at mSystems we are committed to making rapid final decisions. Because it appears that addressing the reviewers' concerns will require a significant amount of additional work that would delay the ultimate outcome, my decision at this time is to reject the manuscript.

If you feel that you wish to address the criticisms of the reviewers, you may submit a revised manuscript to mSystems as a new submission, which will be assigned a new manuscript number and receipt date. Please note the previous manuscript number and my name in the cover letter. Provide point-by-point responses to the issues raised by the reviewers in a file named "Response to Reviewers," not in your cover letter. Upload a compare copy of the manuscript (without figures) as a "Marked-Up Manuscript" file. In the response file, specify with page and line numbers where the revisions have been made in the marked-up manuscript.

Firstly, we wish to thank the Editor for giving the opportunity to reply to the reviewers. We thank reviewers for their useful comments which we have addressed below and in the revised manuscript. Undoubtedly, the reviewers provided useful insight and comments, which no doubt contributed to make this a better paper.

We feel it is important to say that regarding the comments on the genome-based predictions of antibiotic resistance and the need for a more robust machine learning approach. There are a wide variety of published articles that achieved important results in AMR prediction using machine learning k-mers-based approaches without considering the population structure and also applied to *S. aureus* (Davis JJ et al., Sci Rep, 2016, 6:27930; Kavvas ES et al., Nat Commun. 2018, 9: 4306; Nguyen, Long et al., J. Clin, 2019, 57:2; Hyun, Kavvas et al., PLoS Comput. Biol, 2020, 16:3). However, we have fully acknowledged the comment posed by Reviewer 1 and we have made considerable changes by implementing a more robust approach that accounts for potentially biased subpopulations in the data (see details below). Regarding the virulence factor prediction, in the original manuscript, we have used protocols extensively published in the field and specifically used for *S. aureus* (Manara et al., Genome Med. 2018, 10:82, Bosi et al., PNAS, 2016, 113:26; Rocha et al., PLoS ONE, 2019, 14:8; Murphy et al., Int. J. Antimicrob. Agents, 2019, 54:6; Goyal et al., Front. Microbiol., 2019, 10:1525; Randad et al., Sci Rep, 2019, 9:6774; Klein et al., Sci Rep, 2020, 10:13243). However, we recognise, as pointed out by the reviewer the possibility of false negatives. To this aim we have made changes to the methodological approach to account for this issue and posed attention if the results obtained would make the any novelty of this paper to other published works. A detailed explanation of the changes in the revised manuscript are provided below.

Reviewer comments:

Reviewer #1 (Comments for the Author):

The paper by Wang et al presents the sequencing and analysis of a convenience collection of 673 *S. aureus* isolates from China. The data appear to be good quality and have been deposited in a public database. All the strains have MICs determined to a range of antibiotics using CLSI guidelines, which is excellent. Most of the paper is a descriptive analysis of the data using public analysis tools. There are few surprises in the strains that dominate the region, as this ground has been covered extensively in prior surveys based on MLST, *spa*-typing etc. Nevertheless, there is value in confirmation and the genome sequence data is a welcome addition to an important geographical area that has not been well covered to date. There are some interesting results as well. These include new SCCmec subtypes, new Bayesian dating of the split times of major clades and also showing that Chinese and European groups of ST59 and ST239 fell in different branches. It was also interesting to see that the St398 were more intermixed in regard to geographic origin and this likely reflected importation of meat products into China.

We were delighted that the reviewer recognised the added value in the genome sequence data obtained from an important geographical area that has not been well surveyed to date. We believe that the collection and analysis of 673 *S. aureus* isolates from food, as well as from hospitalised and healthy individuals collected over a 9-year period, across China represents a unique catalogue that not only provide relevant results *per se* (see for more details below) but studies like this one are crucial to survey the global epidemiology of infectious agents. This dataset can be easily meta-analysed and integrated with further studies and this clearly could lead to a deeper and further understanding of the epidemiology of this bacterium and how to prevent and treat resistant infections, especially from an important area such China.

In this respect, the new Bayesian dating of the split times of major clades is relevant, as also pointed out by the reviewer. We believe that altogether, these observations shed more light on the complexity of *S. aureus* epidemiology in a geographically important region and on the need for a more unbiased survey of the commensal and pathogenic *S. aureus* community, to avoid the misrepresentation of specific genomic traits. This is further supported by the resistance patterns and phenotypes/genotype associations identified by the machine learning approach that has been improved thanks to the suggestions of the reviewer (see below for details). For example, in the updated version of the manuscript we show that CC59 samples had a higher k-mer frequency for most of the k-mers when compared to other resistant clonal complexes for oxacillin and cefoxitin. This result further indicates the importance of CC59 samples, especially in China, and demonstrate the difference between this clonal complex and the other resistant clonal complexes available in our data set. In addition, we found that resistance to trimethoprim/sulfamethoxazole was highly associated with the gene *tcaA*, no previous link has been found to trimethoprim/sulfamethoxazole in literature. More details are given in the results section.

Our analyses also highlighted a high diversity of STs, SCCmec, and *spa*-types, resulting into a wide number of clones, including novel ST and SCCmec types. Also, our refined analysis of the SCCmec cassettes highlighted the presence of resistances and diversity within cassette type IV, hosting further metal (cadmium) and antimicrobial resistance genes (kanamycin and bleomycin), as well as completely new SCCmec subtypes.

We recognise, as correctly pointed out by the reviewer, our depiction of the results of our work was too simplistic and too much oriented on emphasizing the machine learning approach, which led to downplaying of the other results. We have now edited the Abstract, Results and Discussion sections of the manuscript to better emphasizes the results obtained with this work and any novelty of this paper to other published works.

Please see the text added in the Abstract, Results and Discussion and reported below:

Abstract – Line 38-43

‘In addition, novel variants of SCC*meclV* cassette hosting extra metal and antimicrobial resistance genes, as well as a new SCC*mec* type were found. New Bayesian dating of the split times of major clades, showed Chinese and European that ST59 and ST239 fell in different branches. On the contrary, the clonal transmission of ST398 was more intermixed in regard to geographic origin’

Abstract – Line 45-46

‘..MRSA lineages enriched of AMR determinants that share similar genetic traits of antimicrobial.’

Results – Line 312-317

‘Moreover, the cluster maps for oxacillin and cefoxitin shown a higher normalized k-mer frequency for samples belonging to CC59 when compared to other clonal complexes in the resistant samples. To the best of our knowledge, this difference in terms of k-mer frequency of CC59 samples was not previously shown in any other work and it further indicated the importance of analysing CC59 samples, especially in China.’

Results – Line 358-380

‘The insertion sequence IS1182 was found to be highly predictive for both antibiotics. This insertion sequence has been reported to be close to the SCC*mec* complex but has also been reported to inactivate the gene *lytH*, increasing resistance (Fujimura and Murakami, Antimicrob Agents Chemother., 2008, 52:2). Chloramphenicol resistance was highly associated with the known resistance gene *cat* (77 k-mers). The chloramphenicol resistance gene *cat* and the plasmid replication gene *rep* (also correlated with chloramphenicol resistance), were found to co-occur in 63 isolates from both animal and non-animal food sources but were not found in any human isolates. *rep* is an initiator protein in pT181 family plasmids, including pC221 known to typically carry chloramphenicol resistance (Kwong *et al.*, Front Microbiol, 2017, 8:2279), indicating the likely presence of this plasmid in these food isolates. A further gene encoding tryptophan decarboxylase was also significant for chloramphenicol resistance. This gene is a promising gene candidate as despite no previous known links to chloramphenicol resistance, the tryptophan biosynthesis pathway has been previously associated with vancomycin (Matsuo *et al.*, Antimicrob Agents Chemother., 2013, 57:12). Interestingly, resistance to trimethoprim/sulfamethoxazole was highly associated with the gene *tcaA*. Inactivation of this gene has been previously shown to increase resistance to teicoplanin and vancomycin (Maki *et al.*, Antimicrob Agents Chemother., 2004, 48:6), but no previous link has been found to trimethoprim/sulfamethoxazole. This would benefit from further experimental validation. Ciprofloxacin and tetracycline were primarily related to genes involved in horizontal gene transfer (IS256 and ISNCY, Figure 5c), suggesting these resistances may be plasmid-mediated. Resistance to ciprofloxacin is typically caused by point mutation in the chromosomal *gyrA* and *parC* genes, however there is growing evidence of plasmid mediated resistance (Kim, Antimicrob Agents Chemother., 2009, 53:2). This likely indicates a prevalence of plasmid mediated quinolone resistance in food-related *S. aureus* isolates in China, as has previously been reported on a smaller scale study in *Escherichia coli* farmed fish (Jiang, J Antimicrob Chemother., 2012, 67:10).’

Discussion - Lines 449-452

‘This dataset can be easily meta-analysed and integrated with further studies and this clearly could lead to a deeper and further understanding of the epidemiology of this bacterium and how to prevent and treat resistant infections, especially from an important area such as China.’

Discussion– Line 458-461

'Bayesian divergence analysis also showed that whilst the ST9 and ST398 MRSA in China exhibited relatively independent phylogenetic evolutionary relationships compared with the isolates from other countries, there were signs of mixing of strains, possibly linked to the importation of meat products.'

Discussion – Lines 476-482

'Here we have shown for a large number of antibiotics, genes that are significantly and predictively associated with antibiotic resistance phenotypes in isolates evolved in China. Moreover, to the best of our knowledge for the first time, we have shown that CC59 samples had a higher k-mer frequency for most of the k-mers when compared to other resistant clonal complexes for oxacillin and ceftazidime. This result further indicates the importance of CC59 samples, especially in China, and demonstrates the difference between this clonal complex and the other resistant clonal complexes available in our data set.'

Discussion – Lines 491-496

'ML also revealed a correlation between IS1182 and ceftazidime and oxacillin; *rep* and tryptophan decarboxylase and chloramphenicol; and *tcaA* and trimethoprim-sulfamethoxazole, which could be investigated further. Additionally, ciprofloxacin resistance was found to be predominantly plasmid mediated, in contrast to the global prevalence of chromosomal point mutations.'

The section of virulence factors was disappointing, as the analysis was based on BLAST search of public virulence factor databases against the genomes using a single cutoff threshold (the choice of which was not justified). The results were quite unsurprising - it is well-known that virulence factors are highly linked to clonal complex but in the case of highly variable proteins such as FnbA/B it is likely that there are false negatives as only one BLAST threshold was used.

We thank the reviewer for pointing out the likelihood of false negatives in our results. We have acknowledged the comment posed by the reviewer by implementing a modification to our original approach to specifically address the analysis of variable proteins such as *FnbA/B* and likelihood of false negatives (see below). However, regarding the methodological approach (method, setting and single cut-off threshold) that we have originally used to identify the virulence factors, we feel that it is important to say that the original protocol is routinely and widely used in the scientific community working in this field, and in our opinion we considered it as a well-accepted way to approach the investigation of virulence factors. In support of our opinion, we report below a list of articles published in major journals, where an identical or similar approach (single cut off threshold) to the one we have originally used to predict the same virulence factors, including *FnbA/B*, in *S. aureus* is reported.

Manara *et al.*, Genome Med. 2018, 10:82

Bosi *et al.*, PNAS, 2016, 113:26

Rocha *et al.*, PLoS ONE, 2019, 14:8

Murphy *et al.*, Int. J. Antimicrob. Agents, 2019, 54:6

Goyal *et al.*, Front. Microbiol., 2019, 10:1525

Randad *et al.*, Sci Rep, 2019, 9:6774

Klein *et al.*, Sci Rep, 2020, 10:13243

In all these published works the authors used a single cut off threshold, that is identical or less stringent to the one we have used. In our work we have used threshold (detailed in the Method section), identity > 90% and coverage > 75%, that is more stringent than the one suggested by the developers of ABRicate (-minid [n.n] Minimum DNA %identity [75]. -- mincov [n.n] Minimum DNA %coverage [0]). Our choice was mandated by other works already published and in particular Manara *et al* 2018 (referenced above). None of the

published works reported above and that used our same approach have used multiple BLAST cut off thresholds.

However, we fully acknowledge the reviewer' concern and to this aim we have revised our methodological approach. Specifically, we undertook further analysis of the virulence factors we had discussed and found in some cases isoforms of these virulence factors were not present in the VFDB database. To account for this lack of information possibly leading to false negatives comparative results we have applied several changes. Specifically, in this new version the virulence factors were searched against the full dataset (set B) of VFDB that includes >30 000 sequences of all genes related to known and predicted VFs, and not just against the core dataset (set A) which covers only experimentally verified as done previously. The search was done using Abriicate software (<https://github.com/tseemann/abriicate>) based on BLAST with default parameter setting and only the best hits were recorded as entries. Our approach (searching parameters, virulence genes to be searched for - selected based on a careful literature review for their clinical relevance was done as detailed in Manara *et al.*, Genome Med. 2018, 10:82; Rocha *et al.*, PLoS ONE, 2019, 14:8 and Randad *et al.*, Sci Rep, 2019, 9:6774. In addition, we have introduced a further step to specifically target those genes, such as FnbA/B, encoding several isotypes (Loughman *et al.*, BMC Microbiol, 2008, 8:74), and hence prone to generate false negatives, as mentioned by the reviewer. To this aim we used as queries the different isotypes as described in Loughman *et al.*, BMC Microbiol, 2008, 8:74, and selected the best hits as entries.

However, although we believe that the updated procedure is dramatically improved, no major or novel results have been found (see figure below). Because of this, we have omitted these results and only the virulence factors obtained with the machine learning approach have been kept in the updated manuscript. Now that we have incorporated the population structure in our machine learning approach, we have also adjusted for the effect of clonality when searching for the virulence factors as previously done in Peacock *et al.*, Infection and Immunity, 2002, 70:9, we believe that our approach is more robust now.

The manuscript has been edited as detailed below:

Methods: Lines 688-703

‘Identification of AMR virulence and HGT genes

Where the machine learning was able to predict the antimicrobial class based on k-mers, these were then used to search the genome for genes that contained the k-mers. Using the pan genome of the study isolates, annotated using Prokka, the k-mers were mapped to genomes using a BLASTN query with the parameters: *eval=1000, word_size=13, gapopen=5, gapextend=2, outfmt=5, strand='plus'*. Genes with an identity of >70% and coverage of >70% were considered to be variants of the same gene and hence were discounted as duplicates, as done in previous literature (Meric *et al.*, Nat Comms. 2018, 9:5034), however a more stringent threshold was used to ensure all gene variants were accounted for. The k-mer hit count (how many k-mers mapped to each identified gene) of the genes identified was then assessed for statistical significance at a significance level of 0.05 using a binomial exact test, with the probability of a gene hit based on the length of the gene and number of k-mer combinations possible per gene. All genes found were checked in published literature to find previous associations with AMR, virulence, or HGT. A clustered heatmap was produced in using the Seaborn package in python showing the number of k-mers mapped to each gene per antibiotic. Gene names were colour coded according to their association with AMR, virulence, or HGT.'

Results - Lines 412-428

'The machine learning approach also revealed the presence of significant associations between virulence genes and the antimicrobial resistance profiles. Several virulence genes (e.g. *lp12*, *essG*, *sp1F*, *sdrE*, *map*, *ss1* plus others, see Supplementary Table 6) were found to be associated with resistant phenotypes. Specifically, *lp12*, a host invasion gene was found to be a discriminant genetic feature of multiple resistant phenotypes (cefoxitin, oxacillin, ciprofloxacin, gentamicin, clindamycin, tetracycline and erythromycin). Type VII secretion gene (*essG*), serine protease (*sp1F*), serine-aspartate repeat protein (*sdrE*) and adhesin gene *map* were correlated to cefoxitin, clindamycin, erythromycin, gentamicin, and oxacillin resistance. Finally, toxin *ss1* was correlated to cefoxitin, ciprofloxacin, clindamycin, erythromycin, gentamicin, oxacillin, trimethoprim/sulfamethoxazole and tetracycline. Other virulence genes also found to be associated with AMR phenotypes can be seen in Figure 5c and Supplementary Table 6.'

Discussion – Lines 497-499

'Notably, using machine learning significant correlations between virulence genes (*lp12*, *essG*, *sp1F*, *sdrE*, *map*, and *ss1*, plus others less strongly associated) and antimicrobial resistance phenotypes were found.'

Results Lines 251-280 – the following text has been deleted:

Diversity of virulence and antimicrobial resistance traits among different clonal complexes. We searched the genomes for the presence of virulence genes (Figure 5 and Supplementary Table 1) and identified virulence genes including genes with adherence (*clf*, *cna*, *ebp*, *fnb*, *map*, *sdr*, *spa*, *vWbp*, *ica*), alpha-Hemolysin precursor (*hly/hla*), aureolysin (*aur*), capsular polysaccharide (*cap8*), capsular polysaccharide (*chp*), glycerol ester hydrolase (*geh*, *scn*), hyaluronate lyase precursor (*hysA*), immune evasion (*adsA*, *esa*, *ess*, *esx*, *hld*, *hlg*, *lukF-PVL*, *lukD*, *lukS-PVL*, *sbi*), ion transporter (*isd*), serine protease (*ssp*), specific sortase (*srtB*), staphylocoagulase (*coa*), staphylocoagulase (*sak*), toxin (*eta*, *se*, *sel*, *tsst-1*), and triacylglycerol lipase precursor (*lip*)^{51,52}.

The *fnbA* and *fnbB* genes that facilitate adherence to the host epithelium⁵³ were found co-present only in CC1-*agrIII* (with one exception in ST22). Similarly, the *vWbp* gene, known to facilitate *S. aureus* dissemination as thromboembolic lesions and resistance to opsonophagocytic clearance by host immune cells⁵⁴, was found only in CC1-*agrIII*, CC630, and ST121, from humans, meat and vegetable origin. The *cap8H-K* genes, important in *S. aureus* infections, supporting the survival of *S. aureus* by modulating host immunity⁵⁵⁻⁵⁷, were almost entirely absent in CC398, and most of CC5, CC25 and CC9. The *map* gene was absent in CC398, CC5-*agrI* and CC25. Enterotoxins, *seb* and *selk/q* were found in 94% of the CC59 isolates, but absent in the other clonal groups ($p < 0.0001$). Similarly, CC1-t144

isolates possessed the *seh* and *sell* genes encoding toxins and toxic shock syndrome toxin 1 (*tsst-1*), rarely found in other clones ($p < 0.0001$). Several novel clones identified in this study, e.g. CC630-ST239 and CC5-ST6, possessed *selk/q* but *sea* instead of *seb*. Notably, CC398 and CC59 showed absence of *esaD/E/G*, genes responsible for staphylococci persistence in tissues and *esxB*, encoding a nuclease toxin involved in interspecies competition⁵⁸. CC9, and to a smaller extent CC59, harboured an increased number of antimicrobial resistance genes compared to other clonal complexes. Specifically, *acc(6')*-*aph(2')*, *aadD*, *ant(6)-Ia*, *fexA*, *inuB*, *isaE*, *ermC*, *tetL* and *dfrgG* major in CC9; *ant(6)-Ia*, *aph(3')-III*, *cat-p233*, *ermB* and *tetK* major in CC59, conferred resistance to chloramphenicols, lincomycins, macrolides, tetracyclines and trimethoprim.'

Overall, the descriptive sections of the results are quite well written but too much time and effort is spent reconfirming already well-known results.

We were delighted that the reviewer recognised the descriptive sections of the results as well-written. As correctly pointed out by the reviewer these sections are possibly too long. We have now shortened the main text (results) to address this comment. We hope that, through the text reductions introduced in the main text we have now made the navigation of the reported result easier. The results section has been edited as follows:

Results –the following text has been deleted:

Lines 149-151:

'The other four most prevalent CCs (CC5, CC1, CC398 and CC7), were mixed but predominantly MSSA (88%, 77%, 65% and 93% respectively)'

Lines 158-160:

'Among the PVL+ MRSA isolates, 94% (66/70) were CC59, showing a strong association with this clone. The other PVL+ MRSA were found in ST88 (n=2), ST398 (n=1) and ST1 (n=1) isolates'

Lines 177-180:

'Within this SCC*mec* type we observed a certain degree of variability, in particular subtype IVa and IVc were identified with SCC*mec* elements showing insertions not found with the already described subtypes (Figure2).'

Lines 190-193:

'According to the known SCC*mec* structures, the *mec* gene complex class A, is present in type II, III, VIII and XIII SCC*mec* cassettes, in combination with different *ccr* gene complex including *ccrA2B2*, *ccrA3B3*, *ccrA4B4*, and *ccrC2*, respectively. This is suggestive of differential evolution of MRSA strains with recombination event in China.'

'Oxacillin resistant CC9 isolates were sourced mainly from pork. Also, of note, amongst the oxacillin resistant isolates is a cluster of nine infectious human samples taken from a Shanghai hospital in 2015, all ST239 (CC630). Conversely, 124 susceptible human isolates, all MSSA, came from a wide range of *S. aureus* clones from both healthy and infectious individuals. These isolates were dated after 2015 with only three exceptions in line with recent research indicating an increase in the proportion of MSSA in China from 2015'

Lines 251-280:

Diversity of virulence and antimicrobial resistance traits among different clonal complexes. We searched the genomes for the presence of virulence genes (Figure 5 and Supplementary Table 1) and identified virulence genes including genes with adherence (*clf*, *cna*, *ebp*, *fnb*, *map*, *sdr*, *spa*, *vWbp*, *ica*), alpha-Hemolysin precursor (*hly/hla*), aureolysin

(*aur*), capsular polysaccharide (*cap8*), capsular polysaccharide (*chp*), glycerol ester hydrolase (*geh*, *scn*), hyaluronate lyase precursor (*hysA*), immune evasion (*adsA*, *esa*, *ess*, *esx*, *h1b*, *h1d*, *h1g*, *lukF-PVL*, *lukD*, *lukS-PVL*, *sbi*), ion transporter (*isdI*), serine protease (*ssp*), specific sortase (*srtB*), staphylocoagulase (*coa*), staphylocoagulase (*sak*), toxin (*eta*, *se*, *sel*, *tsst-1*), and triacylglycerol lipase precursor (*lip*)^{51,52}.

The *fnbA* and *fnbB* genes that facilitate adherence to the host epithelium⁵³ were found co-present only in CC1-*agrIII* (with one exception in ST22). Similarly, the *vWbp* gene, known to facilitate *S. aureus* dissemination as thromboembolic lesions and resistance to opsonophagocytic clearance by host immune cells⁵⁴, was found only in CC1-*agrIII*, CC630, and ST121, from humans, meat and vegetable origin. The *cap8H-K* genes, important in *S. aureus* infections, supporting the survival of *S. aureus* by modulating host immunity⁵⁵⁻⁵⁷, were almost entirely absent in CC398, and most of CC5, CC25 and CC9. The *map* gene was absent in CC398, CC5-*agrII* and CC25. Enterotoxins, *seb* and *selk/q* were found in 94% of the CC59 isolates, but absent in the other clonal groups ($p < 0.0001$). Similarly, CC1-t144 isolates possessed the *seh* and *sell* genes encoding toxins and toxic shock syndrome toxin 1 (*tsst-1*), rarely found in other clones ($p < 0.0001$). Several novel clones identified in this study, e.g. CC630-ST239 and CC5-ST6, possessed *selk/q* but *sea* instead of *seb*. Notably, CC398 and CC59 showed absence of *esaD/E/G*, genes responsible for staphylococci persistence in tissues and *esxB*, encoding a nuclease toxin involved in interspecies competition⁵⁸. CC9, and to a smaller extent CC59, harboured an increased number of antimicrobial resistance genes compared to other clonal complexes. Specifically, *acc(6')-aph(2')*, *aadD*, *ant(6)-Ia*, *fexA*, *inuB*, *isaE*, *ermC*, *tetL* and *dfgrG* major in CC9; *ant(6)-Ia*, *aph(3')-III*, *cat-p233*, *ermB* and *tetK* major in CC59, conferred resistance to chloramphenicols, lincomycins, macrolides, tetracyclines and trimethoprim.'

The final section, a machine-learning exploration of antibiotic resistance genetics is seriously flawed. The idea was to take an agnostic approach to finding genetic determinants of antibiotic resistance based on this excellent genomic dataset with linked traits. However, the genetic basis of many common antibiotic-resistant phenotypes in *S. aureus* is well known and several papers (not cited) have already shown genome-based prediction with > 95% accuracy. Comparing against these methods should have been the starting point for this section.

Thanks for the useful suggestion. We have now expanded and updated the discussion to acknowledge previously published works that identified the genetic determinants of antibiotic resistance by using genome-based predictions. And as correctly pointed out by the reviewer we have compared the findings reported in the newly cited works against our results. See Discussion line numbers 464-468 and 500-556 and reported below.

Discussion – Line 464-468

'Several approaches to analysing WGS against resistance phenotypes have been previously published in literature. These fall into two main branches, machine learning approaches, as we have implemented here and genome-wide association studies, a statistical association-based approach. For *S. aureus*, several groups have attempted to link resistance phenotypes to genes, with varying types of antibiotics and datasets (Hyun et al. PLoS Comput Biol, 2020, 16:3; Jaillard et al., PLoS Genet, 2018, 14:11; Wheeler et al., bioRxiv, 2019, 758144)'

Discussion – Line 500-556

'Other published papers have applied machine learning to identify AMR genes associated to resistance phenotypes, however there are many differences that make this work unique. Hyun et al. (Hyun et al. PLoS Comput Biol, 2020, 16:3) employ an SVM approach to identify genes from SNPs based on the pan genome of 288 *S. aureus* isolates. These isolates all

taken from human sources were isolated in Singapore, the US and Russia, and were primarily composed of ST239, ST22 and ST5. Hence, our study is very different, this is also highlighted by the fact that only 5 of the 13 gene candidates identified by Hyun (*hyfX* linked to erythromycin, *oppD* and *Acyl-CoA* linked to gentamicin, *P-type ATPase* and *rep* linked to tetracycline) were also found to be important in our primarily food-based work. In addition, in our study, as described above, novel determinants underlying resistance were found strongly associated with the AMR phenotypes, such as *tcaA*, tryptophan decarboxylase and IS1182, an original finding to respect to the literature. Both Jaillard *et al.* (Jaillard *et al.*, PLoS Genet, 2018) and Wheeler *et al.* (Wheeler *et al.*, bioRxiv, 2019, 758144) used genome-wide association studies to make predictions about *S. aureus* phenotypes from the whole genome sequences. Both studies used isolates from UK-based humans, giving a very different population structure to this study. Jaillard made predictions against four of the antibiotics we used in our study. Jaillard also tests methicillin which we do not, however we tested oxacillin which is chemically very similar and has replaced methicillin in clinical use. For these antibiotics, the predicted genes associated to the resistance phenotype for methicillin (*mecA*), erythromycin (*ermC*) and one of the 2 gentamicin genes (*aac*) were in common with our work. Whist, two genes predicted to correlate to trimethoprim resistance in Jaillard's work (*ybaK* and *mqa1*), were associated to different antibiotics in our study (*ybaK*: cefoxitin, erythromycin, gentamicin, oxacillin; *mqa1*: cefoxitin, clindamycin, erythromycin, gentamicin, oxacillin). This may be because in our study we tested trimethoprim in conjunction with sulfamethoxazole, as it is generally used clinically, rather than trimethoprim alone as in Jaillard. In the case of Wheeler, six antibiotics overlapped with our study (gentamicin, oxacillin/methicillin, erythromycin, tetracycline ciprofloxacin and clindamycin) and the genes found to be associated to resistance for five of these were also found by us (*aacA-aphD* linked to gentamicin, *mecA* linked to oxacillin/methicillin, *ermA* and *ermC* linked to erythromycin, *tetK* and *tetM* linked to tetracycline). However, differences were found in the case of ciprofloxacin where distinct genes were identified in the two studies (chromosomal genes *gyrA* and *parC* to be significant in Wheeler *et al.*). Such diversity may result from the differences in resistance mechanisms, with this study indicating a prevalence of plasmid-mediated ciprofloxacin resistance in China and Wheeler's study suggesting a prevalence of chromosomal resistance mechanisms in their UK-based study.

The strong overlaps between our results and previous works in identifying genes known to be correlated to the selected resistance phenotypes indicates the robustness of the methods we have employed and gives confidence to the novel predicted genes that have arisen from our analysis. Our more heterogenous dataset results in a slightly lower machine learning accuracy than other published works, 76.6-95.5%, compared to 91.9-98.6% (Hyun) and 94.7-100% (Wheeler). Further, we have also for the first time found genetic determinant of AMR of chloramphenicol and cefoxitin in *S. aureus* using machine learning. Our study has been able to assess the differentially evolved Chinese clones of *S. aureus* and show that while many resistance mechanisms align with those seen elsewhere globally, differences may have evolved. However, we would like to point out that several approaches for AMR prediction models that have already been developed from genome sequence collections of many of *Staphylococcus aureus*, and also for many other species are often designed to maximize accuracy in predicting AMR phenotypes, emphasizing their diagnostic capabilities over their capacity to uncover genetic mechanisms for resistance. Many such models are also based on the detection of genes from a curated set of known AMR determinants, rendering them difficult to generalize to different treatments or organisms and unsuitable for discovering novel genes or interactions that drive resistance. In our case we have not used known AMR determinants to search for but the whole genome information obtained by mapping back the k-mers on the pangenome.'

Furthermore, by not using GWAS methods that attempt to correct for population-structure, the machine-learning algorithms are at risk for finding spurious associations through genetic linkage.

The approach we have taken, GWAS and other methods in literature have their own strengths and weaknesses and no one approach has been shown to be wholly superior (see Hyun et al. *PLoS Comput Biol*, 2020, 16:3 and San et al., *Front. Microbiol*, 2020, 10:3119). However, we recognise the reviewer valid point that we had neglected to account for population structure within our models and thank the reviewer for this advice. We have now implemented this in our approach following the methodology of Aun et al., *PLoS Comput Biol*, 2018, 14:10. We have corrected for population structure at the feature selection stage of our pipeline, implementing a weighted chi square, based on mash distances between samples. This had a large effect on the k-mers selected with only a small proportion remaining the same. In addition, we have also corrected for overcounting of gene variants when performing the kmer-blast to find significant genes. This has resulted in some changes in significant genes (73% of the original genes are still present), removing as many spurious links as practicable. We feel this has significantly strengthened the methodology and conclusions drawn from the results and once again thank the reviewer for this valuable contribution.

The amended methods have been detailed in the manuscript on lines 668-676 and 688-703 and below:

Methods – Lines 668-676

'The python package Scikit-learn was used to reduce the number of features used. To correct for bias, the clonal the population structure by filtering the k-mers based on a weighted pairwise chi-squared tests between each feature and the phenotype class as suggested by Aun et al., *PLoS Comput Biol*, 2018, 14:10. The weights of each genome are calculated using the Gerstein, Sonnhammer and Cothia method. Subsequently, another k-mer filtering is performed and the top 2000 features (k-mers) are selected with highest chi-squared statistic (p value lower than 0.000000000000001, confidence value of more than 99.9999999%). The weighting was based on a distance matrix based on mash distances (Aun et al., *PLoS Comput Biol*, 2018, 14:10). A panel of machine learning algorithms were then run in the Scikit-learn package: logistic regression (LR); linear Support Vector Machine (SVM), radial basis function (RBF) SVM.'

Methods – Lines 688-703

'Where the machine learning was able to predict the antimicrobial class based on k-mers, these were then used to search the genome for genes that contained the k-mers. Using the pan genome of the study isolates, annotated using Prokka, the k-mers were mapped to genomes using a BLASTN query with the parameters: `evaluate=1000`, `word_size=13`, `gapopen=5`, `gapextend=2`, `outfmt=5`, `strand='plus'`. Genes with an identity of >70% and coverage of >70% were considered to be variants of the same gene and hence were discounted as duplicates, as done in previous literature (Meric *et al.*, *Nat Comms*. 2018, 9:5034), however a more stringent threshold was used to ensure all gene variants were accounted for. The k-mer hit count (how many k-mers mapped to each identified gene) of the genes identified was then assessed for statistical significance at a significance level of 0.05 using a binomial exact test, with the probability of a gene hit based on the length of the gene and number of k-mer combinations possible per gene. All genes found were checked in published literature to find previous associations with AMR, virulence, or HGT. A clustered heatmap was produced in using the Seaborn package in python showing the number of k-mers mapped to each gene per antibiotic. Gene names were colour coded according to their association with AMR, virulence, or HGT.

The results section has also been extensively altered in lines with the new results on lines 340-428.

These problems are exemplified in the results section, where a major "finding" is *mecA* being associated with oxacillin resistance (this has been known for decades) as well as the genes *maoC* and *ugpQ*, which are genetically linked to *mecA* on the SCCmec cassette and probably have no function in drug resistance. Most of the other results such as "virulence gene association with resistance" are due to spurious linkage. " does not reflect both the concept behind machine learning as previously explained as well as findings in literature.

Our intention in drawing attention to the *mecA* association with oxacillin resistance was not to claim novelty, but to demonstrate that our method is robust and that known determinants of resistance are found. This approach to testing the robustness of the methodology has been previously used for example in Jaillard et al., PLoS Genet, 2018. In this paper 26 of the identified genetic determinants has been previously described, whilst the other 8 had no previous AMR linkage. This work demonstrates that finding known genes gives confidence that the method is robust that that genes found are not caused by spurious linkage, particularly given the population correction that was applied.

However, we recognise, as correctly pointed out by the reviewer our depiction of the previous works was probably too simplistic, which led to the incorrect interpretation of our results and downplaying the contribution of different approaches. We also recognise that we did not emphasise enough the novel results from our work. We have now edited the Results and Discussion sections of the manuscript to better clarify the different methods, compare against them and the results with the other works. Regarding the concern of the association of the virulence genes to resistance due to spurious linkage we have made changes to the methodological approach to account for this issue by implementing a more robust approach that accounts for potentially biased subpopulations in the data (see previous comments).

Results – Line 348-411

‘For oxacillin and cefoxitin resistance, as expected the *mecA* gene previously recognised as conferring resistance (Broekema et al, J. Clin. Microbiol., 2009, 47:1; Hososaka, Journal of Infections and Chemotherapy, 2007, 13:2; Swenson et al., J. Clin. Microbiol., 2005, 43:8) was the primary gene found by machine learning, with 100 and 98 different k-mers, respectively, mapping back to this gene. The genes *maoC* and *ugpQ*, previously shown to be an SCCmec associated elements (Monecke et al., PLoS ONE, 2016, 11:9), were also found to be highly discriminant between resistant and susceptible cefoxitin and oxacillin phenotypes. Interestingly, *ugpQ* was also found to be located in the SCCmec cassette of 342 out of the 343 MRSA strains and *maoC* located in the SCCmec cassette of 332 isolates (Supplementary Table 7). The identification of genes known and expected to be correlated to the selected resistance phenotypes indicates the robustness of the methods employed as stressed by Jaillard (Jaillard et al., PLoS Genet, 2018). The insertion sequence IS1182 was found to be highly predictive for both antibiotics. This insertion sequence has been reported to be close to the SCCmec complex but has also been reported to inactivate the gene *lytH*, increasing resistance (Fujimura and Murakami, Antimicrob Agents Chemother., 2008, 52:2). Chloramphenicol resistance was highly associated with the known resistance gene *cat* (77 k-mers). The chloramphenicol resistance gene *cat* and the plasmid replication gene *rep* (also correlated with chloramphenicol resistance), were found to co-occur in 63 isolates from both animal and non-animal food sources but were not found in any human isolates. *rep* is an initiator protein in pT181 family plasmids, including pC221 known to typically carry chloramphenicol resistance (Kwong et al., Front Microbiol, 2017, 8:2279), indicating the likely presence of this plasmid in these food isolates. A further gene encoding tryptophan decarboxylase was also significant for chloramphenicol resistance. This gene is a promising

gene candidate as despite no previous known links to chloramphenicol resistance, the tryptophan biosynthesis pathway has been previously associated with vancomycin (Matsuo *et al.*, *Antimicrob Agents Chemother.*, 2013, 57:12). Interestingly, resistance to trimethoprim/sulfamethoxazole was highly associated with the gene *tcaA*. Inactivation of this gene has been previously shown to increase resistance to teicoplanin and vancomycin (Maki *et al.*, *Antimicrob Agents Chemother.*, 2004, 48:6), but no previous link has been found to trimethoprim/sulfamethoxazole. This would benefit from further experimental validation. Ciprofloxacin and tetracycline were primarily related to genes involved in horizontal gene transfer (IS256 and ISNCY, Figure 5c), suggesting these resistances may be plasmid-mediated. Resistance to ciprofloxacin is typically caused by point mutation in the chromosomal *gyrA* and *parC* genes, however there is growing evidence of plasmid mediated resistance (Kim, *Antimicrob Agents Chemother.*, 2009, 53:2). This likely indicates a prevalence of plasmid mediated quinolone resistance in food-related *S. aureus* isolates in China, as has previously been reported on a smaller scale study in *Escherichia coli* farmed fish (Jiang, *J Antimicrob Chemother.*, 2012, 67:10). Transposases, IS1182, IS6 and IS256 and the antimicrobial resistance gene *aacA-aphD* are present in many but not all isolates in our cohort from both human and animal sources suggesting the widespread prevalence of these genes correlated with multiple resistances in China.'

Discussion - Lines 500-547

'Other published papers have applied machine learning to identify AMR genes associated to resistance phenotypes, however there are many differences that make this work unique. Hyun *et al.* employ an SVM approach to identify genes from SNPs based on the pan genome of 288 *S. aureus* isolates. These isolates all taken from human sources were isolated in Singapore, the US and Russia, and were primarily composed of ST239, ST22 and ST5. Hence, our study is very different, this is also highlighted by the fact that only 5 of the 13 gene candidates identified by Hyun (*hylX* linked to erythromycin, *oppD* and *Acyl-CoA* linked to gentamicin, *P-type ATPase* and *rep* linked to tetracycline) were also found to be important in our primarily food-based work. In addition, in our study, as described above, novel determinants underlying resistance were found strongly associated with the AMR phenotypes, such as *tcaA*, tryptophan decarboxylase and IS1182, an original finding to respect to the literature. Both Jaillard *et al.* (Jaillard *et al.*, *PLoS Genet*, 2018) and Wheeler *et al.* (Wheeler *et al.*, *bioRxiv*, 2019, 758144) used genome-wide association studies to make predictions about *S. aureus* phenotypes from the whole genome sequences. Both studies used isolates from UK-based humans, giving a very different population structure to this study. Jaillard made predictions against four of the antibiotics we used in our study. Jaillard also tests methicillin which we do not, however we tested oxacillin which is chemically very similar and has replaced methicillin in clinical use. For these antibiotics, the predicted genes associated to the resistance phenotype for methicillin (*mecA*), erythromycin (*ermC*) and one of the 2 gentamicin genes (*aac*) were in common with our work. Whist, two genes predicted to correlate to trimethoprim resistance in Jaillard's work (*ybaK* and *mgo1*), were associated to different antibiotics in our study (*ybaK*: cefoxitin, erythromycin, gentamicin, oxacillin; *mgo1*: cefoxitin, clindamycin, erythromycin, gentamicin, oxacillin). This may be because in our study we tested trimethoprim in conjunction with sulfamethoxazole, as it is generally used clinically, rather than trimethoprim alone as in Jaillard. In the case of Wheeler, six antibiotics overlapped with our study (gentamicin, oxacillin/methicillin, erythromycin, tetracycline ciprofloxacin and clindamycin) and the genes found to be associated to resistance for five of these were also found by us (*aacA-aphD* linked to gentamicin, *mecA* linked to oxacillin/methicillin, *ermA* and *ermC* linked to erythromycin, *tetK* and *tetM* linked to tetracycline). However, differences were found in the case of ciprofloxacin where distinct genes were identified in the two studies (chromosomal genes *gyrA* and *parC* to be significant in Wheeler *et al.*). Such diversity may result from the differences in resistance mechanisms, with this study indicating a prevalence of plasmid-mediated ciprofloxacin resistance in China and Wheeler's study suggesting a prevalence of chromosomal resistance mechanisms in their UK-based study.

The strong overlaps between our results and previous works in identifying genes known to be correlated to the selected resistance phenotypes indicates the robustness of the methods we have employed and gives confidence to the novel predicted genes that have arisen from our analysis. Our more heterogenous dataset results in a slightly lower machine learning accuracy than other published works, 76.6-95.5%, compared to 91.9-98.6% (Hyun) and 94.7-100% (Wheeler). Further, we have also for the first time found genetic determinant of AMR of chloramphenicol and cefoxitin in *S. aureus* using machine learning. Our study has been able to assess the differentially evolved Chinese clones of *S. aureus* and show that while many resistance mechanisms align with those seen elsewhere globally, differences may have evolved.'

Not to belabor the problems with this section, as it is clear that considerable effort went into this work, but there is no evidence presented that the approach produces any results beyond what is currently known and reported in the literature.

We acknowledge that our manuscript failed to fully communicate the novelty of both the importance of the dataset we are contributing to the growing *S. aureus* sequencing library and the novelty of some of the genetic linkages we found to AMR resistance. We have updated our results and discussion sections to better communicate this. And importantly showed that our methodological approach and results are rather innovative with respect to those one previously published.

Discussion – Lines 485-556

'Thanks to ML, we were able to identify genes which, individually or in patterns, featured a strong correlation with resistance to methicillin or to panels of multiple antimicrobials, regardless of the source of the isolate. As an example of the robustness of the methodology, the *maoC*, *uggQ* and *mecA* genes were found strongly correlated to cefoxitin and oxacillin resistance. ML also revealed a correlation between IS1182 and cefoxitin and oxacillin; rep and tryptophan decarboxylase and chloramphenicol; and *tcaA* and trimethoprim-sulfamethoxazole, which could be investigated further. Additionally, ciprofloxacin resistance was found to be predominantly plasmid mediated, in contrast to the global prevalence of chromosomal point mutations. Notably, using machine learning significant correlations between virulence genes (*lpl2*, *essG*, *sp1F*, *sdrE*, *map*, and *ss7*, plus others less strongly associated) and antimicrobial resistance phenotypes were found.

Other published papers have applied machine learning to identify AMR genes associated to resistance phenotypes, however there are many differences that make this work unique. Hyun *et al.* (Hyun *et al.* PLoS Comput Biol, 2020, 16:3) employ an SVM approach to identify genes from SNPs based on the pan genome of 288 *S. aureus* isolates. These isolates all taken from human sources were isolated in Singapore, the US and Russia, and were primarily composed of ST239, ST22 and ST5. Hence, our study is very different, this is also highlighted by the fact that only 5 of the 13 gene candidates identified by Hyun (*hyx* linked to erythromycin, *oppD* and *Acyl-CoA* linked to gentamicin, *P-type ATPase* and *rep* linked to tetracycline) were also found to be important in our primarily food-based work. In addition, in our study, as described above, novel determinants underlying resistance were found strongly associated with the AMR phenotypes, such as *tcaA*, tryptophan decarboxylase and IS1182, an original finding to respect to the literature. Both Jaillard *et al.* (Jaillard *et al.*, PLoS Genet, 2018, 14:11) and Wheeler *et al.* (Wheeler *et al.*, bioRxiv, 2019, 758144) used genome-wide association studies to make predictions about *S. aureus* phenotypes from the whole genome sequences. Both studies used isolates from UK-based humans, giving a very different population structure to this study. Jaillard made predictions against four of the antibiotics we used in our study. Jaillard also tests methicillin which we do not, however we tested oxacillin which is chemically very similar and has replaced methicillin in clinical use. For these antibiotics, the predicted genes associated to the resistance phenotype for

methicillin (*mecA*), erythromycin (*ermC*) and one of the 2 gentamicin genes (*aac*) were in common with our work. Whist, two genes predicted to correlate to trimethoprim resistance in Jaillard's work (*ybaK* and *mqr1*), were associated to different antibiotics in our study (*ybaK*: cefoxitin, erythromycin, gentamicin, oxacillin; *mqr1*: cefoxitin, clindamycin, erythromycin, gentamicin, oxacillin). This may be because in our study we tested trimethoprim in conjunction with sulfamethoxazole, as it is generally used clinically, rather than trimethoprim alone as in Jaillard. In the case of Wheeler, six antibiotics overlapped with our study (gentamicin, oxacillin/methicillin, erythromycin, tetracycline ciprofloxacin and clindamycin) and the genes found to be associated to resistance for five of these were also found by us (*aacA-aphD* linked to gentamicin, *mecA* linked to oxacillin/methicillin, *ermA* and *ermC* linked to erythromycin, *ermA* linked to clindamycin, *tetK* and *tetM* linked to tetracycline). However, differences were found in the case of ciprofloxacin where distinct genes were identified in the two studies (chromosomal genes *gyrA* and *parC* to be significant in Wheeler *et al.*). Such differences may result from the differences in resistance mechanisms, with this study indicating a prevalence of plasmid-mediated ciprofloxacin resistance in China and Wheeler's study suggesting a prevalence of chromosomal resistance mechanisms in their UK-based study.

The strong overlaps between our results and previous works in identifying genes known to be correlated to the selected resistance phenotypes indicates the robustness of the methods we have employed and gives confidence to the novel predicted genes that have arisen from our analysis. Our more heterogenous dataset results in a slightly lower machine learning accuracy than other published works, 76.6-95.5%, compared to 91.9-98.6% (Hyun) and 94.7-100% (Wheeler). Further, we have also for the first time found genetic determinant of AMR of chloramphenicol and cefoxitin in *S. aureus* using machine learning. Our study has been able to assess the differentially evolved Chinese clones of *S. aureus* and show that while many resistance mechanisms align with those seen elsewhere globally, differences may have evolved. However, we would like to point out that several approaches for AMR prediction models that have already been developed from genome sequence collections of many of *Staphylococcus aureus*, and also for many other species are often designed to maximize accuracy in predicting AMR phenotypes, emphasizing their diagnostic capabilities over their capacity to uncover genetic mechanisms for resistance. Many such models are also based on the detection of genes from a curated set of known AMR determinants, rendering them difficult to generalize to different treatments or organisms and unsuitable for discovering novel genes or interactions that drive resistance. In our case we have not used known AMR determinants to search for but the whole genome information obtained by mapping back the k-mers on the pangenome.'

Results - Lines 345-411

'For every antibiotic, except penicillin we obtained a list of potential genetic determinants of antibiotic resistance, with importance measured as the number of unique k-mers mapped back to the gene. For penicillin the small number of sensitive samples impeded machine learning. For oxacillin and cefoxitin resistance, as expected the *mecA* gene previously recognised as conferring resistance (Broekema *et al.*, J. Clin. Microbiol., 2009, 47:1; Hososaka, Journal of Infections and Chemotherapy, 2007, 13:2; Swenson *et al.*, J. Clin. Microbiol., 2005, 43:8) was the primary gene found by machine learning, with 100 and 98 different k-mers, respectively, mapping back to this gene. The genes *maoC* and *ugpQ*, previously shown to be an SCC*mec* associated elements (Monecke *et al.*, PLoS ONE, 2016, 11:9), were also found to be highly discriminant between resistant and susceptible cefoxitin and oxacillin phenotypes. Interestingly, *ugpQ* was also found to be located in the SCC*mec* cassette of 342 out of the 343 MRSA strains and *maoC* located in the SCC*mec* cassette of 332 isolates (Supplementary Table 7). The identification of genes known and expected to be correlated to the selected resistance phenotypes indicates the robustness of the methods employed as stressed by Jaillard (Jaillard *et al.*, PLoS Genet, 2018). The insertion sequence IS1182 was found to be highly predictive for both antibiotics. This insertion sequence has

been reported to be close to the SCCmec complex but has also been reported to inactivate the gene *lytH*, increasing resistance (Fujimura and Murakami, Antimicrob Agents Chemother., 2008, 52:2). Chloramphenicol resistance was highly associated with the known resistance gene *cat* (77 k-mers). The chloramphenicol resistance gene *cat* and the plasmid replication gene *rep* (also correlated with chloramphenicol resistance), were found to co-occur in 63 isolates from both animal and non-animal food sources but were not found in any human isolates. *rep* is an initiator protein in pT181 family plasmids, including pC221 known to typically carry chloramphenicol resistance (Kwong *et al.*, Front Microbiol, 2017, 8:2279), indicating the likely presence of this plasmid in these food isolates. A further gene encoding tryptophan decarboxylase was also significant for chloramphenicol resistance. This gene is a promising gene candidate as despite no previous known links to chloramphenicol resistance, the tryptophan biosynthesis pathway has been previously associated with vancomycin (Matsuo *et al.*, Antimicrob Agents Chemother., 2013, 57:12). Interestingly, resistance to trimethoprim/sulfamethoxazole was highly associated with the gene *tcaA*. Inactivation of this gene has been previously shown to increase resistance to teicoplanin and vancomycin (Maki *et al.*, Antimicrob Agents Chemother., 2004, 48:6), but no previous link has been found to trimethoprim/sulfamethoxazole. This would benefit from further experimental validation. Ciprofloxacin and tetracycline were primarily related to genes involved in horizontal gene transfer (IS256 and ISNCY, Figure 5c), suggesting these resistances may be plasmid-mediated. Resistance to ciprofloxacin is typically caused by point mutation in the chromosomal *gyrA* and *parC* genes, however there is growing evidence of plasmid mediated resistance (Kim, Antimicrob Agents Chemother., 2009, 53:2). This likely indicates a prevalence of plasmid mediated quinolone resistance in food-related *S. aureus* isolates in China, as has previously been reported on a smaller scale study in *Escherichia coli* farmed fish (Jiang, J Antimicrob Chemother., 2012, 67:10). Transposases, IS1182, IS6 and IS256 and the antimicrobial resistance gene *aacA-aphD* are present in many but not all isolates in our cohort from both human and animal sources suggesting the widespread prevalence of these genes correlated with multiple resistances in China.'

Specific points

I find the title is an overstatement of the contents of the paper. There is actually actually little new information about the genetic basis of multidrug resistance.

We have edited the title, removing the word 'multidrug' to better reflect the core content of the manuscript.

L27 "is notorious for its extraordinary capacity .. ",

This is an overstatement that trivialises the nature of the treat of the bacterium :*S. aureus* is not as multidrug resistance as many Gram negative pathogens such as *E. coli*, *Pseudomonas*, *Acinetobacter* etc. Most of the isolates in the survey were actually MSSA.

We have edited this sentence to reflect this comment

Lines 27-28:

'Methicillin-resistant *S. aureus* (MRSA) has been found capable of acquiring resistance to most antimicrobials.'

Importance: part of this section are just cut and pasted from the abstract.

We have edited this section to better reflect the important findings in the manuscript and to further differentiate it from the abstract. see below

Importance - Lines 50-74

Little information is available on the epidemiology and characterization of *S. aureus* in China. The role of food is cause of major concern: Staphylococcal food-borne diseases affect thousands every year, and the presence of resistant Staphylococcus strains on raw retail meat products is well documented. We studied a large heterogeneous dataset of *S. aureus* isolates from many provinces of China isolated from food, as well as from individuals. Our large whole-genome collection represents a unique catalogue, that can be easily meta-analysed and integrated with further studies and adds to the library of *S. aureus* sequences in the public domain in a currently underrepresented geographical region. The new Bayesian dating of the split times of major drug-resistant enriched clones is relevant in showing that Chinese and European have evolved differently. Our machine learning approach, across a large number of antibiotics, shows novel determinants underlying resistance and reveal frequent resistant traits in specific clonal complexes posing the attention to the importance of specific resistant clonal complexes in China.

Our findings substantially expand what is known of the evolution and genetic determinants of resistance food-borne *S. aureus* in China and add crucial information for the WGS-based surveillance of *S. aureus*.

L95 - It would be good to describe how isolates were selected in the results section.

This is described in-depth in the Methods, however we edited the results section to include the description of how isolates were selected.

Results - Lines 118-128

'A total of 7937 food-associated *S. aureus* isolates were cultured from various foods from 27 provinces during the years 2010-2018 in China. In addition, 142 *S. aureus* human-associated isolates including 53 healthy and 89 infected people-associated isolates collected in Shanghai between 2015 and 2017 were employed in this study. In total, 343 food-associated isolates and 18 human-associated isolates were identified as MRSA. These MRSA isolates together with all human-associated MSSA isolates, and 250 food-associated MSSA isolates (randomly selected from the full collection), giving 735 isolates in total, were used for the study of the population structure and the molecular epidemiology (Supplementary Table 1) and sent for genome sequencing.'

L158 - "This is suggestive .." - This line is overly speculative for the results section and should be deleted.

Done

L244 - The title of this section is garbled.

The section title got mangled up, we have corrected the title.

Lines 281-282:

'Resistance to oxacillin and cefoxitin appears strongly associated to the clonal complexes CC59 and CC9.'

L251 and others. It is not clear why there are two values for sensitivity and specificity given.

As a proof of robustness each algorithm used for the classification was run 30 times and metrics (Accuracy, AUC, sensitivity and specificity) collected for each run. The mean of the

30 iteration was then used as the final result statistic. The values for sensitivity, specificity, Accuracy and AUC are the two averaged (30 runs) lower and upper limits of the range summarising the results of the 10 different antibiotic predictive models. We have now clarified this in the manuscript.

Results Lines 288-298:

'In this study, the best overall performance considering all antibiotic models was obtained using a linear RBF SVM classifier: accuracy = 76.6-95.5% (range of means across all antibiotic models); AUC (Area Under the Curve) = 66.5-96.2% (range of means across all antibiotic models), sensitivity = 5.79-99.9% (range of means across all antibiotic models), specificity = 32.9-99.8% (range of means across all antibiotic models), (Figure 5a and Supplementary Table 5).' The best predictions of resistance/susceptibility were obtained for oxacillin (accuracy 93.85%, sensitivity 89.42%, specificity 99.71% and AUC 96.24%) and cefoxitin (accuracy 92.83%, sensitivity 87.70%, specificity 99.79% and AUC 95.26%). Overall, 8 antibiotics achieved an AUC >0.8, except for chloramphenicol and penicillin.

In general, the quality of the figures is high but there is a tendency to put too many colors on the outside of the trees which ends up reducing the visual impact. This is a special problem for colorblind viewers. Suggest lumping the data into fewer categories and using fewer colors.

We thank the reviewer for this advice. We have edited all figures to simplify the figures and remove red/green combinations.

Figure 4 would be better if the strains were shaded by country of origin rather than whether they came from this study, which is less important

Done

Figure 5 would be better in the supplement.

In light of new results Figure 5 has been removed entirely (see previous comments regarding the prediction of the virulence factors)

Figure 6b too many colors make this figure impossible to decipher

This figure has been replaced with a more informative and clearer figure (now named Figure 5).

Figure 6c - did not understand what this figure is attempting to show. Better figure legend and/or simplification of the figure would help.

The figure legend for this heatmap has been updated to clarify the figure now named Figure 5c.

Reviewer #2 (Comments for the Author):

Staphylococcus aureus is a well-recognized pathogen that is encountered in nosocomial settings as well as being a food safety biological hazard that must be controlled to protect human health. In particular healthcare associated (HA) methicillin-resistant *Staphylococcus aureus* (MRSA) first identified in 1961, was mainly associated with those settings. More recently, other strains have been identified according to the locations from which these were cultured and these include livestock associated (LA)-MRSA and community associated (CA)-MRSA, which also represent threats to veterinary public health. Sequence type (ST)

398 is one example of a particularly pathogenic example of LA-MRSA, identified originally in Europe (and now being found in China). HA-; CA- and LA-MRSA are now being increasingly identified outside their original domains, suggesting that transmission is taking place. This epidemiological feature needs to be carefully described, to enable the development of strategies aimed at control. Whole genome sequencing (WGS) may provide a useful approach to dissecting the molecular epidemiology of these pathogens and uncovering their transmission routes.

In this paper, these authors describe the WGS and subsequent detailed analysis of a large collection of *S. aureus* originally cultured from various foods and hospitalized and healthy individuals. The bacterial collection spans 9-years from 2010-2018, and is taken from across 27 provinces in China. Following the determination of the WGS for each of 673 *Staphylococcus aureus* these data were then subjected to Bayesian divergence analysis and machine learning (ML) strategies to extend the understanding of the associated genomic data. Comparisons were made between data obtained using ML-based approaches and conventional antimicrobial susceptibility testing (AST) analysis to describe antimicrobial resistance geno-/phenotypes.

Thank you for your kindly comments and suggestions that will help us greatly improve the quality of our manuscript. Indeed, WGS and machine learning are powerful tools that can help obtain more comprehensive and accurate understanding of epidemiological features of these pathogens.

In reviewing this MS, the study presented is both extensive and comprehensive. The MS is well-written and is of interest to those working in the area of food-related bacterial infection. There are a few very minor issues summarized below that the authors should consider amending –

We were delighted that the reviewer recognized the research as well-designed, executed and written. We have revised the manuscript according to the reviewer's comments.

Minor items to be considered by the authors -
L85- food safety;

Done (see line 107).

L94- genomic;

Done (see line 118).

L125 (and elsewhere)- PVL+;

Done.

Figure-2; this is difficult to comprehend, as shown. Perhaps it could be drawn in Landscape view then the detail could be made clearer? For example, the delineation of the *mec* gene complex should be shown across all SCCmec types drawn, along with the *ccr* gene complex; then the insertion of plasmid pBORa53 should be delineated (ie- show the 5'- and 3'- boundaries in each case) and similarly for plasmid pUB110;

Figure 2 has been revised according to reviewer's suggestion.

L148- blaZ (NOT blaS as shown);

Done (see line 182).

L155- IS is NOT in italics as shown;

Done (see line 189).

L163- isolates;

Done (see line 198).

L190- evolution;

Done (see line 225).

L212- in reference to Figure 4d (ST239 Bayesian generated tree), I do not see any reference to Algeria? Should this be Argentina? Please correct?

Algeria is between China and Australia in the right of this ring. we acknowledge that in our previous figure this was not clear. Our revised Figure 4, with coloured country labels makes it much easier to identify the country of origin of each sample.

L236- do you mean possessed the enterotoxins *selk* and *selq*, and *sea* instead of *seb*? Please clarify? This is how I read it from the heatmap;

Yes, apologies for this error. We have removed this line and the figure in light of new results.

Figure-5; I note two CC1 clades, is this correct? Perhaps a note to indicate to the reader why this split has occurred. It appears to be related to some of the immune functioning genes, at least?

The two CC1 clades were CC1-ST1 and CC1-ST188. These two clades showed quite different virulence profiles, such as the *clfA*, *fnbA/B*, *vWbp*, *esaG1-8*, *esaD*, *esaE*, *esxA-D*, *lukF-PVL*, *lukS-PVL*, *coa*, *seh*, *selk/l/q*, and *tsst-1* mostly present in CC1-ST1.

However, this figure has now been removed in light of new results.

L248- explain the SVM classifier -a sentence is sufficient and also explain the abbreviation (see also L483);

We have added an explanation as requested.

Results - Lines 286-288.

'Support vector machines (SVMs) are powerful yet flexible supervised machine learning algorithms which aim to classify the data by finding an N-dimensional hyperplane to separate the datapoints.'

L303- initiator protein;

Done (see line 364).

L309- the numbers of all IS elements should be in italics;

Done.

L317- antibiotic names do not need to be capitalised (see also L367);

Done.

L322- cadmium;

Done (see line 440).

L377-378- delete this line as it is a repeat of an earlier one (see L334);

Done.

L338- shed (not showed);

Done (see line 446).

L399- *Staphylococcus aureus* ATCC®29231

Done (see line 586).

L406- explain the abbreviation STE?

Done (see line 593).

L705 & 774- references 87 and 99 are repeats;

This has been corrected.

L812- no data was presented for Taiwan?

Apologies for this mistake, this has been corrected as “The ST9 isolates from our cohort clustered together with isolates from China and away from those collected elsewhere.” (see line 1045).

December 21, 2020

Prof. Fengqin Li
China National Center for Food Safety Risk Assessment
Beijing
China

Re: mSystems01185-20 (Whole-genome sequencing and machine learning analysis of *Staphylococcus aureus* from multiple heterogeneous sources in China reveals common genetic traits of antimicrobial resistance)

Dear Prof. Fengqin Li:

I have taken the advice of the reviewers and recommend the manuscript be revised with mostly minor modifications. Both reviewers agree that the manuscript is important and deserves publication, but it still needs to be improved prior to publication. Please carefully address all of the reviewers comments below. In particular, please be certain to robustly address Reviewer #3 comments 1, 3, and 5 regarding literature review, MLSTs, and figure improvements.

Below you will find the comments of the reviewers.

To submit your modified manuscript, log onto the eJP submission site at <https://msystems.msubmit.net/cgi-bin/main.plex>. If you cannot remember your password, click the "Can't remember your password?" link and follow the instructions on the screen. Go to Author Tasks and click the appropriate manuscript title to begin the resubmission process. The information that you entered when you first submitted the paper will be displayed. Please update the information as necessary. Provide (1) point-by-point responses to the issues raised by the reviewers as file type "Response to Reviewers," not in your cover letter, and (2) a PDF file that indicates the changes from the original submission (by highlighting or underlining the changes) as file type "Marked Up Manuscript - For Review Only."

Due to the SARS-CoV-2 pandemic, our typical 60 day deadline for revisions will not be applied. I hope that you will be able to submit a revised manuscript soon, but want to reassure you that the journal will be flexible in terms of timing, particularly if experimental revisions are needed. When you are ready to resubmit, please know that our staff and Editors are working remotely and handling submissions without delay. If you do not wish to modify the manuscript and prefer to submit it to another journal, please notify me of your decision immediately so that the manuscript may be formally withdrawn from consideration by mSystems.

Sincerely,

Christopher Marshall

Editor, mSystems

Journals Department
Reviewer comments:

Reviewer #2 (Comments for the Author):

In performing this review of the originally revised paper, I am please to note that the MS has been substantially improved. Nonetheless, there remain a nuber of minor and in one case a major issue, as follows-

Most of the minor issues relate to gramatical errors and I will provide just a sample of these -

L40- delete the word;

L43- replace the word sensitive (when referring to antimicrobial compounds) throughout, with susceptible;

L43- how antimicrobial compounds were evaluated 11 (as stated here) or 10 (see L121; L280);

L60- posing?

L256- furthers;

L294- IS1168 was not found in Figure 5c? [see also L316; L411; L427];

L562- explain the abbreviation ESS?

The major issue arises on L293-

L315 through to L336 is a complete duplicate?

Reviewer #3 (Comments for the Author):

Wang and colleagues present a descriptive study on the diversity of *S. aureus* isolated from different sources in different Chinese regions over a time span of 8 years. They selected 673 isolates from both food and humans and performed WGS to characterize different lineages by MLST, SCCmec-, spa-, and PVL-typing. They moreover investigate the presence/absence profile of a selection of resistance and virulence genes and the phylogenetic divergence of Chinese branches of specific lineages. They additionally apply an ML approach to identify determinants of resistance for a subset of antibiotics.

Despite the overall work is definitely of interest for the field, there are major flaws that need to be addressed:

1. There is an overall lack of comparison with the available literature, even in the discussion section. The most striking example is the lack of comparison with a large WGS study focusing on European isolates (Aanensen et al., doi:10.1128/mBio.00444-16) and with a very similar study from Italy that performed exactly the same analysis, especially on the new variants of the SCCmecIV cassette (Manara et al., <https://doi.org/10.1186/s13073-018-0593-7>). This latter study also describes one of the variants here reported as novel (the kanamycin and bleomycin resistance one). I found a lot of overlap in a number of choices with this last work. For instance, the analysis of the SCCmecIV cassette's variants and the choice of genes of interest in Supplementary Table 1, which is very similar to those chosen by Manara et al. in their Supplementary Table 3 also in the way the genes' functions are described (e.g. adherence, immune evasion, ...). The authors definitely read the paper as they cite it in the Response to Reviewers (Reviewer #1, "Our choice was mandated by other works already published and in particular Manara et al 2018") and only forgot to cite it, but I also think that a comparison would be very relevant, at least for the type IV cassettes.

2. There is no discussion of a large part of the data, as the Discussion section focuses on the ML approach only, and other findings are only summarized without a proper discussion in light of available literature. I think a discussion of the findings reported in L120-126 is very easy to perform, as there is a considerable body of literature on antibiotic resistance in *S. aureus* in China and nearby countries, not to mention the rest of the world (e.g. Aanensen et al., that also used similar approaches, but also all the studies that performed antibiotic susceptibility testing in the lab). The same is true for the CC, ST, SCCmec, spa, and PVL prevalence analysis, as the only comparison with available literature is at the level of the single lineages (L152 and L156). As already mentioned above, also the novel cassettes are not discussed in light of available literature, with particular mention to the SCCmecIVc variant that is already described in Manara et al (L166-168), which make me doubt also of the fact that the other cassettes here described as novel variants might have been described in the literature before but it was not carefully checked.

3. The Authors report that they found 29 novel STs, but do not provide the allelic profile of the MLST genes, which could easily be added to Supplementary Table 2, and do not discuss for instance how much different these new STs are from those already available. It is known that single-nucleotide sequencing errors occur frequently in whole-genome sequencing, therefore it would be relevant to know whether these new STs are only one SNP away from "common STs" or their allelic profile is completely different. Also, it is not reported from which sources these new STs were isolated.

4. When there is some discussion of the results, this is pretty poor. For instance, in L192-193 the explanation for the presence of a single isolate from pork in Beijing clustering with US ones is that it could be an imported pork product (why????), while the closest isolates coming from pigs are not considered as an explanation (e.g. a lineage associated with live pigs that is then found in pork meat?). The same is true for L195-197 (and later in L387-388), where the explanation for the presence of ST398 is cross-contamination from food, when a large body of literature has shown that ST398 is becoming commonly present in humans (among others, Aanensen et al. and Manara et al. already reported above, <https://doi.org/10.1016/j.ijmm.2013.02.010>, DOI: 10.3201/eid1703.101036, and DOI: 10.1371/journal.pone.0041855) as we can see also from Figure 3C, where a number of isolates come from humans. Humans could be the most obvious source of contamination for food preparations.

5. Most Figures are not readable as there are many colors (too many unnecessary details!!!!) and the font used is extremely small. Please see below for more specific comments on the single Figures. This results in specific sections of the manuscript pointing to Figures where the reader cannot really observe what the Authors would like them to observe, e.g. L248-249 that point to Figure 5b (unreadable) and even worse L253-256 that point to Supplementary Figure 2a (even more unreadable and therefore useless).

6. There is a large section of the manuscript that is repeated (L320-342 are exactly the same as L298-L320).

7. There is no data reported for the Bayesian analysis. Which clock models, coalescent prior, and substitution models did the Authors test? Likelihood and Bayes factor should be reported for all tested combinations (including the chosen one) for the reader to understand how reliable these analyses are. 60 million steps are not a lot, but I can understand that the Authors needed to reduce this number to speed up the analysis.

Additionally, I also have some minor points that need to be addressed:

L27: food poisoning also results in infections, I would suggest rephrasing the sentence with something like "from food poisoning to lethal infections".

L39-42: these sentences are very convoluted and the meaning is not clear. I suggest the Authors rephrase at least the second sentence with something like "This pattern was not observed for ST398 clones". Moreover, I am not sure why the Authors did not mention ST9 together with ST59 and ST239 in the first sentence, as it shows a very similar pattern to ST59 according to Figure 4.

L108-116: At the beginning of the Results section there is a very convoluted description of the isolates collected over the years, which are then not analyzed nor used for comparison in the study. I do not see the point of adding these numbers, which are not relevant for the study itself and result really confusing for the reader. If the Authors want to give an overview of the total number of isolates collected over the years, I suggest they condense this information in one sentence like the following one: "Food-associated *S. aureus* isolates used in this study were selected from a collection of 7937 isolates collected between 2010 and 2018 from different foods in 27 provinces in China. For this study, we selected 593 food-associated *S. aureus* isolates (343 MRSA and 250 MSSA) and sequenced them together with 142 isolates (18 MRSA and 124 MSSA) obtained from 53 healthy and 89 infected people in Shanghai between 2015 and 2017 (Supplementary Table 1). Of these, 673 resulted.... {continue with L119}"

L117-118: these lines are a repetition of L109. Please remove.

L145: I think the Authors wanted to refer to Figure 1D instead of 1A, B, C?

L172: there is an inconsistency with Figure 2. Is the *mec* gene complex of this cassette of type A as reported in text or type B as reported in the Figure?

L185: I don't think we can see this from Figure 3a (definitely not for SCCmecV, whose isolates are intermixed with those obtained in other countries). Are there any ST59-SCCmecIV isolates that were not collected in China or only the ones from Japan and Romania (I think it's Japan, too many

colors)? Are the not-annotated genomes in between SCCmecIV and V carrying other cassette types? Please clarify these points, and see extra comments on the Figure below.

L247: AUC>80% to be consistent with the previous lines where AUC is always reported in percentage.

L413-414: the sentence in its present form suggests that the ML approach allowed the Authors to identify new determinants of resistance to methicillin and multiple antimicrobials, which is not the case. Please rephrase this sentence to clarify.

L416-423: this is not a discussion, it is a summary of the results. Please compare with available literature.

L536: in WGS we cannot be sure that *S. aureus* genomes are completely free from contamination even after quality checks. For instance, if the isolate was a mixture of two strains, or if contamination occurred during DNA extraction and library preparation, different strains would be assembled together. Please remove "without contamination".

L615-616: please remove the "color coding" sentence.

Figures:

Figure 1a: The map is huge and the font is very small, it would be better to make the map smaller and the font bigger to make it readable. This panel in my opinion could be added to Supplementary Figure 1.

Figure 1b: It is not easy to read black on blue, please either use white for the font on the blue bars or change the blue to a lighter color. This panel in my opinion is not very relevant and could be added to Supplementary Figure 1.

Figure 1c: Same comment as for 1b for font color on blue bars. Use lighter colors for the bars or for the font.

Figure 1d: The spa type is not readable at all, report this information in a table instead. The SCCmec type N/A should be substituted by "MSSA" or something similar, and colored in white to make the figure more readable. The MRSA/MSSA tip points are impossible to read. Overall, this should be a separate single figure to make it readable. If panels a-c are moved to Supplementary Figure 1, the problem might be solved.

Figure 2: inconsistency between legend (and text) and figure: is the mec gene complex class A or B? Compare the novel variants with those called in Manara et al., because the Type IVc is the same, and I have doubts also for the others.

Figure 3: unreadable. I would suggest to group countries by continent or subcontinent, and sources by major source groups (e.g. all noodles + rice together, all environments together, bean-product + fruit and veg, and so on) also because they are not discussed in the paper, so there's no point in keeping them separate. I can understand leaving pig / cattle / sheep or pork / beef separate. It is moreover impossible to read the tip points to understand whether it is an MRSA/MSSA. Please add a ring to the tree reporting this info.

Figure 4: same comments as for Figure 3. In addition, the order in the panels and in the caption is swapped and panel c is reported in capital letters in disagreement with other panels. Please remove country names from the ring, it is not readable and only adds confusion.

Figure 5b: unreadable, especially the annotations. Please either remove them or change them to make them readable. There is no name and unit for the scale.

Figure 5c: gene names and the number of k-mers are written in a very small font and are not readable. Please reduce the number of reported genes to make the heatmap understandable.

Supplementary Figure 2a is not readable at all. How is the reader supposed to look at these??? Either split the panels into different figures or put these data into a Supplementary Table, there is no point in keeping this.

Typos:

L40: showed THAT Chinese and European ST59 and ST239

L91: a comma is lacking between "ST59-SCCmecIV/V" and "ST22-SCCmecIV"

Dear Dr. Christopher Marshall,

We thank both the Editor and the Reviewers for their useful comments, which we have addressed below and in the revised manuscript.

In the following, a point-by-point response to all the questions and comments is provided. The original questions are in black, our replies in blue.

Items included in submission:

- i. Response to reviewers
- ii. Revised manuscript marked-up copy
- iii. Revised manuscript
- iv. Figure 1 – revised as per Reviewer 3’s suggestion
- v. Figure 2 – corrected as per Reviewer 3’s suggestion
- vi. Figure 3 – revised as per Reviewer 3’s suggestion
- vii. Figure 4 – revised as per Reviewer 3’s suggestion
- viii. Figure 5 – revised as per Reviewer 3’s suggestion
- ix. Supplementary Table 1 - revised as per Reviewer 3’s suggestion
- x. Supplementary Table 2 - revised as per Reviewer 3’s suggestion
- xi. Supplementary Table 3 – minor changes to table legend
- xii. Supplementary Table 4 – minor changes to table legend
- xiii. Supplementary Table 5 – minor changes to table legend
- xiv. Supplementary Table 6 – minor changes to table legend
- xv. Supplementary Table 7 – minor changes to table legend
- xvi. Supplementary Figure 1 – revised as per Reviewer 3’s suggestion
- xvii. Supplementary Figure 2 – (previously Supplementary Figure 3) revised as per Reviewer 3’s suggestion
- xviii. Supplementary Dataset 1 – new, replaces Supplementary Figure 2 as per Reviewer 3’s suggestion

Reviewer comments:

Reviewer #2 (Comments for the Author)

In performing this review of the originally revised paper, I am pleased to note that the MS has been substantially improved. Nonetheless, there remain a number of minor and in one case a major issue, as follows-

We were delighted that the reviewer recognized our effort in revising this paper. We have checked and revised throughout the whole manuscript according to the reviewer's comments. Please see the responses below.

1. Most of the minor issues relate to grammatical errors and I will provide just a sample of these

Many thanks for your kind comments. We have carefully checked throughout the manuscript and corrected those errors.

2. L40- delete the word;

Done. This sentence has also been edited in line with corrections from reviewer 3 (see lines 39-42)

3. L43- replace the word sensitive (when referring to antimicrobial compounds) throughout, with susceptible;

Done throughout the manuscript.

4. L43- how antimicrobial compounds were evaluated 11 (as stated here) or 10 (see L121; L280);

Apologies for this error. All isolates underwent laboratory testing for resistance to a panel of 13 antimicrobials. However, when performing the machine learning analysis three antibiotics (Daptomycin, Linezolid, Vancomycin) did not have enough samples in one of the two classes (resistant/susceptible) to allow cross validation and SMOTE

so were not taken further. This is because the number of splits in the nested cross validation and the number of k-nearest neighbours for SMOTE necessitated at least 12 samples in each class. Hence, we ended up with 10 antibiotics that had sufficient observations in both classes to use in machine learning. We have now updated the Abstract (to reflect the correct number of antimicrobials) and Methods (to better explain the selection criteria used by the machine learning, so that it is now clearer why we started with 13 antibiotics but we ended up analysing 10). See Abstract 44-46, Methods lines 851-859 and below:

Abstract - Lines 44 - 46:

“Finally, we identified genetic determinants of resistance to 10 antimicrobials, discriminating drug-resistant bacteria from susceptible strains in the cohort.”

Methods - Lines 851-859:

“As the classes were unbalanced Synthetic Minority Oversampling Technique (SMOTE) was applied to oversample data of minority class, compensating for unbalanced classes (Chawla et al 2002). The number of splits in the nested cross validation and the number of k-nearest neighbours (default values of 5 neighbours) for SMOTE necessitated a minimum number of 12 samples in the minority class. From the panel of 13 antimicrobials, three (Daptomycin, Linezolid, Vancomycin) had an insufficient number of resistant isolates (minority class) to train and cross validate the machine learning model, leaving 10 trained models.”

Chawla, N.V., Bowyer, K.W., Hall, L.O. and Kegelmeyer, W.P. (2002) SMOTE: synthetic minority over-sampling technique. *Journal of artificial intelligence research*, 16, pp.321-357.

5. L60- posing?

Yes, apologies for this error. This has been now changed as follows:

Importance (Lines 61-64):

“Our machine learning approach, across a large number of antibiotics, shows novel determinants underlying resistance and reveals frequent resistant traits in specific clonal complexes highlighting the importance of particular clonal complexes in China.”

6. L256- furthers;

Apologies, this has been corrected as “further indicates”. (see line 324)

7. L294- IS1168 was not found in Figure 5c? [see also L316; L411; L427];

We could not find IS1168 in our manuscript but assume this to mean IS1182. Thank you for bringing our attention to this matter. There was some inconsistency in the manuscript with using either the specific name (*ISSau3*) or the family name (IS1182) of the transposase. We have now updated the manuscript to specify for each transposase the match between the specific and family name and edited throughout the text for a more coherent use of the two terms. Finally, we have also edited Figure5c to show both names for each transposase. Specifically, changes were made to lines:

Results - Line 364-365:

“The insertion sequence *ISSau3* (IS1182 family) was found to be highly predictive for both antibiotics.”

Results - Lines 380-382:

“Ciprofloxacin and tetracycline were primarily related to genes involved in horizontal gene transfer (IS256 and *ISBli29* (ISNCY family), Figure 5c)”

Results - Lines 409-413:

“Transposases, *ISSau3* (IS1182 family), *ISBli29* (ISNCY family) and IS256 and the antimicrobial resistance gene *aacA-aphD* are present in many but not all isolates in

our cohort from both human and animal-food sources suggesting the widespread prevalence of these genes correlated with multiple resistances in China.”

Discussion - Lines 589-590:

“ML also revealed a correlation between ISSau3 (IS 1182 family) and cefoxitin and oxacillin.”

Discussion - Lines 631-634:

“In addition, in our study, as described above, novel determinants underlying resistance were found strongly associated with the AMR phenotypes, such as *tcaA*, tryptophan decarboxylase and ISSau3 (IS 1182 family), an original finding to respect to the literature.”

8. L562- explain the abbreviation ESS?

Thank you. The abbreviation has been explained (see Methods lines 839-842 and below).

Methods - Lines 839-842:

“The analysis was run for 2 independent chains until the Effective Sample Size (ESS), that is the effective number of independent draws from the posterior distribution, for all parameters was greater than 200 per chain.”

9. The major issue arises on L293-L315 through to L336 is a complete duplicate?

Yes, apologies for this error, we have erroneously duplicated the same text. We have now deleted the duplicated lines 387-409.

Reviewer #3 (Comments for the Author):

Wang and colleagues present a descriptive study on the diversity of *S. aureus* isolated from different sources in different Chinese regions over a time span of 8

years. They selected 673 isolates from both food and humans and performed WGS to characterize different lineages by MLST, SCCmec-, spa-, and PVL-typing. They moreover investigate the presence/absence profile of a selection of resistance and virulence genes and the phylogenetic divergence of Chinese branches of specific lineages. They additionally apply an ML approach to identify determinants of resistance for a subset of antibiotics.

Despite the overall work is definitely of interest for the field, there are major flaws that need to be addressed:

1. There is an overall lack of comparison with the available literature, even in the discussion section. The most striking example is the lack of comparison with a large WGS study focusing on European isolates (Aanensen et al., doi:10.1128/mBio.00444-16) and with a very similar study from Italy that performed exactly the same analysis, especially on the new variants of the SCCmecIV cassette (Manara et al., <https://doi.org/10.1186/s13073-018-0593-7>). This latter study also describes one of the variants here reported as novel (the kanamycin and bleomycin resistance one). I found a lot of overlap in a number of choices with this last work. For instance, the analysis of the SCCmecIV cassette's variants and the choice of genes of interest in Supplementary Table 1, which is very similar to those chosen by Manara et al. in their Supplementary Table 3 also in the way the genes' functions are described (e.g. adherence, immune evasion, ...). The authors definitely read the paper as they cite it in the Response to Reviewers (Reviewer #1, "Our choice was mandated by other works already published and in particular Manara et al 2018") and only forgot to cite it, but I also think that a comparison would be very relevant, at least for the type IV cassettes.

Thanks for the useful suggestion. As correctly pointed out by the reviewer there was an overall lack of comparison with the available literature. We have now expanded and updated the results, discussion and methods to acknowledge previously published works and make appropriate comparisons as suggested by the reviewer. In particular, we added further discussion to make comparison with the European

studies (Aanensen *et al.*, 2016; Manara *et al.*, 2018) and other available literature (see Results lines 175-182, 184-211, 255-257, 261-263; Discussion lines 482-541, 557-565, and Methods lines 783-795 and below). Additional analysis of the presence of the sasX virulence factor was conducted to allow comparison with Aanensen *et al.* (Supplementary Table 1). Further, as the reviewer correctly pointed out, we were mistaken to describe the SCC_{med}Vc variant as novel, as it was previously found in the Manara *et al.*, 2018 paper. Additionally, whilst checking the SCC_{mec} IVa variant we described in our study we have found additional new matches to an unpublished Chinese SCC_{mec} cassette and a newly sequenced plasmid. We have now corrected this in the manuscript and made certain by providing evidence and proper comparison with the literature that the other novel variants we described were correct (See Results Lines 184-211, Methods Lines 783-795 and below)

Regarding the virulence factors in Supplementary Table 1, as the Reviewer correctly pointed out, for this analysis we followed entirely the pipeline of Manara *et al.*, 2018. In our original manuscript we had a section discussing these results, on 1st review we improved our analysis at the suggestion of our original Reviewer 1 who, although we referred to the published Manara protocol criticized that the analysis was based on BLAST search of public virulence factor databases against the genomes using a single cutoff threshold. Upon reflection, this section was taken out due to the lack of novelty. However, Reviewer 3 is correct that a comparison of the virulence factors in our cohort in comparison with European isolates would be a valuable addition to the manuscript. Hence, we have added back a very brief description of our results into the manuscript and some comparison with the Manara *et al.*, 2018 manuscript and updated back the method again in the discussion (see Results lines 175-182, Discussion lines 482-493, Methods lines 802-814 and below).

Results - Lines 175-182:

“We searched the genomes for the presence of virulence factors and identified a wide range of genes (Supplementary Table 1) including adherence (*clf*, *cna*, *ebp*, *fnb*, *map*, *sdr*, *spa*, *vWbp*, *ica*), alpha-Hemolysin precursor (*hly* and *hla*), aureolysin (*aur*),

capsular polysaccharide (*cap8*), capsular polysaccharide (*chp*), glycerol ester hydrolase (*geh*, *scn*), hyaluronate lyase precursor (*hysA*), immune evasion (*adsA*, *esa*, *ess*, *esx*, *hlb*, *hld*, *hlg*, *lukD*, *lukF-PVL*, *lukS-PVL*, *sbi*), ion transporter (*isd*), serine protease (*ssp*), specific sortase (*srtB*), staphylocoagulase (*coa*), staphylocoagulase (*sak*), toxin (*eta*, *se*, *sel*, *tsst-1*), and triacylglycerol lipase precursor (*lip*).”

Results - Lines 184-211:

“Variants of type IV SCC*mec* cassette carrying extra antibiotic resistance genes and novel SCC*mec* cassette type. Among the identified 343 MRSA isolates, nearly half of the isolates (169/343) carried a type IV SCC*mec* cassette. However, in this study, by aligning reconstructed SCC*mec* with the reference cassettes (Ref IVa: AB063172.2 and Ref IVc: B096217.1), we observed several variabilities inside the type IV cassette. Two subtype IVc isolates were found to have the insertion of kanamycin and bleomycin resistance genes as previously found in Italy by Manara et al (Manara et al. 2018). Our isolates (11A1151 and 18A25) were almost identical to the cassette (MF062) found by Manara et al. (Manara et al., 2018), see Figure 2. Isolate 11A1151 carried a cassette identical to MF062, whilst isolate 18A25 had a deletion of the gene *uqpQ*. We confirmed our variant to be an integrated plasmid, pUB110 (McKenzie et al., 1986) via a BLAST search against N315 which is a confirmed SCC*mec*II isolate with integrated pUB110 (assession no D81934.2, identity 99.93%). The insertion of plasmid pUB110, carrying these resistance genes, is frequently associated with type II SCC*mec* elements (Yamaguchi et al., 2020, Ito et al., 1999; Oliveira et al., 2001, Dubin et al. 1991).

An SCC*mec*IVa cassette variant was found, in two isolates from our cohort, to have genes conferring resistance to beta-lactams; *blaI*, *blaR1* and *blaZ* were found between IS431 and the *mec* gene complex. When blasting this SCC*mec* cassette variant against the NCBI nonredundant database we found as subject sequence this SCC*mec* cassette variant (99% identity and query cover) has recently also been found in an SCC*mec*IVa clinical isolate from Wuhan, China (GenBank CP033086.1).

The genome sequence, located in the J3 region (Yamaguchi *et al.*, 2020) of the cassette indicated an integrated plasmid as its structure *Tn552-CadX-CadD* is associated with plasmid pBORa53 (Massida *et al.*, 2006). The plasmid-like region also shows high similarity (100% identity and 94% query cover) to *S. aureus* plasmid pPM1 (GenBank AB699881.1).

According to the International Working Group on the Classification of Staphylococcal Cassette Chromosome Elements and recent reports, there are 14 types of SCC*mec* cassette (IWG-SCC, 2009; Wu *et al.*, 2015; Baig *et al.*, 2018; Urushibara *et al.*, 2020). Three of our MRSA isolates could not be attributed to any of the 14 known cassette types. These novel SCC*mec* elements carried the *ccr* gene complex 7 (A1B6) and the *mec* gene complex class A. Therefore, our study highlights the presence of further resistances and diversity within the same cassette type and the ongoing rearrangements in MRSA.”

Results - Lines 255-257:

“Several studies have shown that ST398 is becoming commonly present in humans (Cuny *et al.*, 2013; Aanensen *et al.*, 2016; Manara *et al.*, 2018), suggesting that humans could be the most obvious source of contamination for food preparations.”

Results - Lines 261-263:

The nine human isolates all carried the virulence gene *sasX*, widely associated with human-associated ST239 in China (Li *et al.*, 2012; Aanensen *et al.*, 2016). None of the other isolates in this study carried this gene, including the two food associated ST239 isolates.

Discussion - Lines 482-493:

“Comparing the virulence factors present in our Chinese cohort with those of the Italian study by Manara *et al.* (Manara *et al.*, 2018) shows many similarities, despite the differences in isolate hosts. Iron uptake, conserved antigen and ACME virulence factors, found in Manara were not present in our cohort. However, immune evasion

genes, toxins, ion transporter, adhesins, capsular polysaccharides and serine proteases were found in similar proportions of isolates in both cohorts, though specific gene presence varied. Several immune evasion genes in our cohort are present in almost all isolates including *esxA/B/C/D*, *hld* and *sbi* as in Manara et al (Manara et al., 2018). Additionally, the toxic shock syndrome toxin *tsst-1* associated with ST22 in Manara et al. (Manara et al., 2018), was present in 28 of our isolates, however in our cohort this toxin was associated with ST1 (n=17, p<0.0001) and ST30 (n=4, p>0.0001). The presence of these virulence factors increases the risk of isolates carrying these factors harming human health so the widespread existence of these in food-related isolates needs to be monitored.”

Discussion - Lines 494-541:

“The CC59-t437-SCC*mecIV/V*, CC9-t899-SCC*mecXII*, and CC398-t011-SCC*mecV* clones were the most frequent food-associated MRSA clones in our study. It has been well documented that CC59-t437-SCC*mecIV/V* clones are major CA-MRSA clones in China and other Asian countries, threatening a vast population due to their epidemiological potential (Chen and Huang, 2014). However, they remain geographically confined and are seen in only low numbers in Europe (Aanensen *et al*, 2016; Manara *et al.*, 2018), suggesting importation to Europe from Asia as has been noted previously (Glasner *et al.*, 2015). The considerable numbers of this CA-MRSA lineage from food in this study could indicate some human contamination, possibly a result of inadequate hygiene measures and improper handling of food (Kadariya *et al*, 2014). Additionally, most of the PVL positive isolates (86.8%) belonged to CC59, demonstrating the pathogenic potential of these MRSA strains. CC9, the predominant LA-MRSA clone in Asia, was mostly identified from retail meat samples in this study. Elsewhere, CC398 is the prevalent LA-MRSA strain found in animals and humans across European countries and North America (Chuang and Huang, 2013). Furthermore, in addition to meat, CC398-t011-SCC*mecV* MRSA isolates were detected from more various food samples including non-animal food such as cake, noodle, fruit, and vegetable, suggesting a different epidemiological characteristic of

these CC398 MRSA strains in China. Recently, this CC398-t011-SCCmecV MRSA clone has been detected from humans with non-animal contact in China and denominated as CA-MRSA (He et al, 2018 Genome Med 10:5). Thus, continued monitoring of this strain's epidemiology and preventing its widespread transmission are essential.

MRSA belonging to CC5, CC22, and CC30 clones are major HA-MRSA clones in Europe, according to a previous pan European study (Aanensen *et al.* 2016). In our study, whilst all the three clones are present in human samples they are in low numbers (CC5, n=19; CC22, n=4; CC30, n=1) and only one of these isolates was MRSA (CC5), highlighting the importance of geographical isolation in MRSA dissemination. Moreover, an Italian study on human-associated *S. aureus* in a paediatric hospital showed that CC5, CC22, CC8, CC1 and CC121 were the prevalent clones and that all CC1 and most CC5 (>96%) isolates were MRSA (Manara *et al.*, 2018). In contrast in our study more than 95% of the CC1 and CC5 isolates were MSSA. In addition, for our human MRSA isolates the major SCCmec type was SCCmecIII (9/14, 64%) while SCCmecIV (54/83, 67%) was frequently detected from Italian study and SCCmecIII was not detected in any samples (Manara *et al.*, 2018). Similarly, Aanensen et al. (Aanensen *et al.*, 2016) only detected SCCmecIII in clones from ST239 and the authors suggested these were likely to have been largely imported from Asia, with SCCmecIV again the most prevalent type in the pan-European study. However, the presence of PVL positive isolates were strongly associated with MSSA in our study, which agreed with the Italian results (Manara et al., 2018).

CC630 has been circulating in Asia since the 1970s and is the prevalent HA-MRSA clone (Chen and Huang, 2014). Correspondingly, in our study the CC630 (ST239-t037-SCCmecIII) clone was the predominant MRSA clone among human samples (9/14), with the other isolates being ST59 (3), ST1(1) and ST5 (1). Whilst these are not prevalent clones in Europe both have been found in clinical isolates. ST239 clones were present in 8 isolates in the pan-European collection (Aanensen et al., 2016), and ST59 were present in 3 of 86 MRSA isolates in Manara *et al.* (Manara

et al., 2018). As in Aanensen *et al.* (Aanensen et al., 2016), the human ST239 isolates in our study carried the virulence gene *sasX*, reflecting the widespread distribution of this gene in ST239 clones in China. However, the two food associated ST239 isolates did not carry this gene and clustered away from the other ST239 isolates in our study in the phylogenetic tree, possibly indicating lineages not carrying this gene may be circulating in food. Additionally, other MRSA lineages, for instance, ST88-SCCmecIV/V, which has been already known as the “African clone” (Schaumburg et al., 2014), were also frequently detected in this study. This finding indicates that the MRSA clones have spread cross-regionally.”

Discussion- Lines 557-565:

“For *S. aureus*, several groups have attempted to link resistance phenotypes to genes (Aanensen et al., 2016; Jaillard et al. 2018; Wheeler et al., 2019, Hyun et al., 2020). For example, Aanensen *et al.* (Aanensen *et al.* 2016) used published resistance genes, with manual curation, to predict antibiotic resistance phenotypes from genotypes with high accuracy. Although their findings provided an *in silico* typing method comparable with phenotypic accuracy, our machine-learning based approach allows impartial identification of genetic determinants of resistance, not limited to those already known. This is particularly important for China, where the differential evolution may have allowed resistance determinants to arise that are not as well represented in the public databases.”

Methods – Lines 783-795:

“SCCmecFinder (Kaya *et al.*, 2018) was used for typing SCCmec and the contigs that aligned to the SCCmec region were further annotated with Prokav1.14.5 (Seeman, 2014). The names of predicted proteins from Prokka were further confirmed with BLASTP (Altschul *et al.*, 1990). The variant SCCmec regions were further aligned using BLAST to identify SCCmec variants and plasmid integration into the SCCmec cassette (Rolo *et al.*, 2017; Côrtes *et al.*, 2018). SCCmec variants in isolates 11A1151 and 18A25 (region encompassing *OrfX* (*rlmH*) to the direct repeat (DR) sequence

GAAGCTTATCATAAGTAA), were aligned to MF062 (GenBank GCA_003240235) and N315 (GenBank D81934.2) using default BLASTn settings (Altschul *et al.*, 1990). Pairwise comparison using BLASTn with default settings was performed for the SCC*meclVa* variant from isolates 11A832 and 18A245 (region encompassing *OrfX(rlmH)* to the DR sequence GAAGCGTATCATAAATGA) against the NCBI nonredundant database using both the SCC*mec* cassette region and only the J3 region of the cassette, using default BLASTn settings.”

Methods – Lines 802-811:

“**Virulence factors and ARG analysis.** Virulence factors and ARGs were searched through Abricate software (<https://github.com/tseemann/abricate>) using the VFDB (Liu *et al.*, 2019) database set B and the CARD database (Alcock *et al.*, 2020) The BLAST search within Abricate was conducted with the following parameters: identity > 90% and coverage (proportion of gene covered) > 75%, in agreement with previous studies (Manara *et al.*, 2018). Virulence gene functional categories were assigned manually based on the gene entry in the VFDB database (Liu *et al.*, 2019). All results were manually curated by careful search of the literature. In addition, to specifically target the *fnbA/B* genes known to generate false negative due to isoforms not present in the public databases (Loughman *et al.*, 2008), we performed an additional blast search, using as queries the different isoforms as described in Loughman *et al.*, and selecting the best hit entries.”

Methods – Lines 811-814:

“To identify the presence of the virulence factor *sasX* within each of our isolates, a BLASTn search was used with *sasX* (GenBank MH143577.1) as the query. Thresholds of 90% identity and 90% coverage were used.”

References

Aanensen, D. M., E. J. Feil, M. T. G. Holden, J. Dordel, C. A. Yeats, A. Fedosejev, R. Goater, S. Castillo-Ramírez, J. Corander, C. Colijn, M. A. Chlebowicz, L. Schouls, M.

Heck, G. Pluister, R. Ruimy, G. Kahlmeter, J. Åhman, E. Matuschek, A. W. Friedrich, J. Parkhill, S. D. Bentley, B. G. Spratt and H. Grundmann (2016). "Whole-Genome Sequencing for Routine Pathogen Surveillance in Public Health: a Population Snapshot of Invasive *Staphylococcus aureus* in Europe." *mBio* 7(3): e00444-00416.

Alcock BP, Raphenya AR, Lau TT, Tsang KK, Bouchard M, Edalatmand A, Huynh W, Nguyen ALV, Cheng AA, Liu S. 2020. CARD 2020: antibiotic resistance surveillance with the comprehensive antibiotic resistance database. *Nucleic Acids Res* 48:D517-D525.

Altschul SF, Gish W, Miller W, Myers EW, Lipman DJ. 1990. Basic local alignment search tool. *J Mol Biol* 215:403-410.

Baig, S., T. B. Johannesen, S. Overballe-Petersen, J. Larsen, A. R. Larsen and M. Stegger (2018). "Novel SCCmec type XIII (9A) identified in an ST152 methicillin-resistant *Staphylococcus aureus*." *Infect Genet Evol* 61: 74-76.

Chen, C.-J. and Y.-C. Huang (2014). "New epidemiology of *Staphylococcus aureus* infection in Asia." *Clinical Microbiology and Infection* 20(7): 605-623.

Chuang, Y.-Y. and Y.-C. Huang (2013). "Molecular epidemiology of community-associated methicillin-resistant *Staphylococcus aureus* in Asia." *The Lancet Infectious Diseases* 13(8): 698-708.

Côrtés MF, Botelho AM, Almeida LG, Souza RC, de Lima Cunha O, Nicolás MF, Vasconcelos AT, Figueiredo AM. 2018. Community-acquired methicillin-resistant *Staphylococcus aureus* from ST1 lineage harboring a new SCCmec IV subtype (SCCmec IVm) containing the tetK gene. *Infection and drug resistance* 11:2583-2592.

Cuny, C., R. Köck and W. Witte (2013). "Livestock associated MRSA (LA-MRSA) and its relevance for humans in Germany." *Int J Med Microbiol* 303(6-7): 331-337.

Drummond, A. J. and A. Rambaut (2007). "BEAST: Bayesian evolutionary analysis by sampling trees." *BMC Evolutionary Biology* 7(1): 214.

Dubin D, Matthews P, Chikramane S, Stewart P. 1991. Physical mapping of the mec region of an American methicillin-resistant *Staphylococcus aureus* strain. *Antimicrob Agents Chemother* 35:1661-1665.

Glasner, C., G. Pluister, H. Westh, J. P. Arends, J. Empel, E. Giles, F. Laurent, F. Layer, L. Marstein, A. Matussek, A. Mellmann, M. Pérez-Vásquez, E. Ungvári, X. Yan, H. Žemličková, H. Grundmann and J. M. van Dijk (2015). "Staphylococcus aureus spa type t437: identification of the most dominant community-associated clone from Asia across Europe." *Clinical Microbiology and Infection* 21(2): 163.e161-163.e168.

He L, Zheng HX, Wang Y, Le KY, Liu Q, Shang J, Dai YX, Meng HW, Wang X, Li T, Gao QQ, Qin JX, Lu HY, Otto M, Li M. (2018). Detection and analysis of methicillin-resistant human-adapted sequence type 398 allows insight into community-associated methicillin-resistant Staphylococcus aureus evolution. *Genome Med* 10:5.

Hyun, J. C., E. S. Kavvas, J. M. Monk and B. O. Palsson (2020). "Machine learning with random subspace ensembles identifies antimicrobial resistance determinants from pan-genomes of three pathogens." *PLOS Computational Biology* 16(3): e1007608.

Ito, T., Y. Katayama and K. Hiramatsu (1999). "Cloning and nucleotide sequence determination of the entire mec DNA of pre-methicillin-resistant Staphylococcus aureus N315." *Antimicrob Agents Chemother* 43(6): 1449-1458.

IWG-SCC (2009). "Classification of staphylococcal cassette chromosome mec (SCCmec): guidelines for reporting novel SCCmec elements." *Antimicrob Agents Chemother* 53(12): 4961-4967.

Jaillard, M., L. Lima, M. Tournoud, P. Mahé, A. van Belkum, V. Lacroix and L. Jacob (2018). "A fast and agnostic method for bacterial genome-wide association studies: Bridging the gap between k-mers and genetic events." *PLoS genetics* 14(11): e1007758-e1007758.

Kadariya, J., T. C. Smith and D. Thapaliya (2014). "Staphylococcus aureus and staphylococcal food-borne disease: an ongoing challenge in public health." *BioMed research international* 2014.

Kaya H, Hasman H, Larsen J, Stegger M, Johannesen TB, Allesoe RL, Lemvig CK, Aarestrup FM, Lund O, Larsen AR. 2018. SCCmecFinder, a web-based tool for typing

of staphylococcal cassette chromosome mec in *Staphylococcus aureus* using whole-genome sequence data. *mSphere* 3:e00612-17.

Li, M., X. Du, A. E. Villaruz, B. A. Diep, D. Wang, Y. Song, Y. Tian, J. Hu, F. Yu, Y. Lu and M. Otto (2012). "MRSA epidemic linked to a quickly spreading colonization and virulence determinant." *Nature Medicine* 18(5): 816-819.

Li, S., S. Sun, C. Yang, H. Chen, Y. Yin, H. Li, C. Zhao and H. Wang (2018). "The Changing Pattern of Population Structure of *Staphylococcus aureus* from Bacteremia in China from 2013 to 2016: ST239-030-MRSA Replaced by ST59-t437." *Front Microbiol* 9: 332.

Liu B, Zheng D, Jin Q, Chen L, Yang J. 2019. VFDB 2019: a comparative pathogenomic platform with an interactive web interface. *Nucleic Acids Res* 47:D687-D692.

Loughman, A., T. Sweeney, F. M. Keane, G. Pietrocola, P. Speziale and T. J. Foster (2008). "Sequence diversity in the A domain of *Staphylococcus aureus* fibronectin-binding protein A." *BMC microbiology* 8: 74-74.

Liu, B., D. Zheng, Q. Jin, L. Chen and J. Yang (2019). "VFDB 2019: a comparative pathogenomic platform with an interactive web interface." *Nucleic acids research* 47(D1): D687-D692.

Manara, S., E. Pasolli, D. Dolce, N. Ravenni, S. Campana, F. Armanini, F. Asnicar, A. Mengoni, L. Galli, C. Montagnani, E. Venturini, O. Rota-Stabelli, G. Grandi, G. Taccetti and N. Segata (2018). "Whole-genome epidemiology, characterisation, and phylogenetic reconstruction of *Staphylococcus aureus* strains in a paediatric hospital." *Genome Medicine* 10(1): 82.

Massidda, O., M. Mingoia, D. Fadda, M. B. Whalen, M. P. Montanari and P. E. Varaldo (2006). "Analysis of the β -lactamase plasmid of borderline methicillin-susceptible *Staphylococcus aureus*: Focus on bla complex genes and cadmium resistance determinants cadD and cadX." *Plasmid* 55(2): 114-127.

McKenzie T, Hoshino T, Tanaka T, Sueoka N. 1986. The nucleotide sequence of pUB110: some salient features in relation to replication and its regulation. *Plasmid* 15:93-103.

Oliveira, D. C., A. Tomasz and H. de Lencastre (2001). "The evolution of pandemic clones of methicillin-resistant *Staphylococcus aureus*: identification of two ancestral genetic backgrounds and the associated mec elements." *Microb Drug Resist* 7(4): 349-361.

Rolo J, Worning P, Nielsen JB, Bowden R, Bouchami O, Damborg P, Guardabassi L, Perreten V, Tomasz A, Westh H, de Lencastre H, Miragaia M. 2017. Evolutionary Origin of the Staphylococcal Cassette Chromosome mec (SCCmec). *Antimicrob Agents Chemother* 61:e02302-16.

Schaumburg, F., A. S. Alabi, G. Peters and K. Becker (2014). "New epidemiology of *Staphylococcus aureus* infection in Africa." *Clinical Microbiology and Infection* 20(7): 589-596.

Song, Z., F. F. Gu, X. K. Guo, Y. X. Ni, P. He and L. Z. Han (2017). "Antimicrobial Resistance and Molecular Characterization of *Staphylococcus aureus* Causing Childhood Pneumonia in Shanghai." *Front Microbiol* 8: 455.

Urushibara, N., M. S. Aung, M. Kawaguchiya and N. Kobayashi (2020). "Novel staphylococcal cassette chromosome mec (SCCmec) type XIV (5A) and a truncated SCCmec element in SCC composite islands carrying speG in ST5 MRSA in Japan." *J Antimicrob Chemother* 75(1): 46-50.

Wheeler, N. E., S. Reuter, C. Chewapreecha, J. A. Lees, B. Blane, C. Horner, D. Enoch, N. M. Brown, M. Estée Török, D. M. Aanensen, J. Parkhill and S. J. Peacock (2019). "Contrasting approaches to genome-wide association studies impact the detection of resistance mechanisms in *Staphylococcus aureus*." *bioRxiv*: 758144.

Wu, Z., F. Li, D. Liu, H. Xue and X. Zhao (2015). "Novel Type XII Staphylococcal Cassette Chromosome mec Harboring a New Cassette Chromosome Recombinase, CcrC2." *Antimicrob Agents Chemother* 59(12): 7597-7601.

Yamaguchi, T., D. Ono and A. Sato (2020). *Staphylococcal Cassette Chromosome mec (SCCmec) Analysis of MRSA. Methicillin-Resistant Staphylococcus Aureus (MRSA) Protocols*, Springer: 59-78.

Yan, X., B. Wang, X. Tao, Q. Hu, Z. Cui, J. Zhang, Y. Lin, Y. You, X. Shi and H.

Grundmann (2012). "Characterization of *Staphylococcus aureus* strains associated with food poisoning in Shenzhen, China." *Appl Environ Microbiol* 78(18): 6637-6642.

2. There is no discussion of a large part of the data, as the Discussion section focuses on the ML approach only, and other findings are only summarized without a proper discussion in light of available literature. I think a discussion of the findings reported in L120-126 is very easy to perform, as there is a considerable body of literature on antibiotic resistance in *S. aureus* in China and nearby countries, not to mention the rest of the world (e.g. Aanensen et al., that also used similar approaches, but also all the studies that performed antibiotic susceptibility testing in the lab). The same is true for the CC, ST, SCCmec, spa, and PVL prevalence analysis, as the only comparison with available literature is at the level of the single lineages (L152 and L156).

We thank the reviewer's very useful suggestions. The antibiotic phenotype results in L126-134 have now been contextualized with existing literature works done in China, Korea, Brazil and South Africa, as well as the USA and Europe, (see Discussion lines 440-451 and below). Analogously, we have contextualised our CC, ST, SCCmec, spa, and PVL prevalence results with existing literature (see Discussion lines 449-481 and below)

Discussion - Lines 449-481:

"Using a comparative genomics pipeline inspired by Manara *et al.* (Manara et al., 2018) we considered our isolates in the context of other similar studies. Across all isolates in our study, 97% of the *S. aureus* isolates showed resistance to at least one antimicrobial and 72% showed MDR, which was consistent with previous reports in China (>94% showed resistance to at least one antibiotic and >58% were MDR in food based isolates) (Liao et al., 2018). The number of isolates carrying at least one resistance found in this study was higher than in other food based reports from Brazil (83%) (Kroning et al., 2016), South Africa (71%) (Pekana and Green, 2018), and Korea (51%) (Kim et al. 2018). Analogously, the MDR in our study was higher than in

India (53%) (Zehra et al., 2019), Korea (35%) (Kim et al. 2018), the USA (10%) (Ge et al. 2017), and Brazil (8%) (Kroning et al. 2016). The recent surveillance study in Europe (Aanensen et al., 2016) revealed a high level of resistance among human-associated *S. aureus*, with 90% of isolates showing resistance to at least one antibiotic and 45% MDR, which is in accordance with our human data (84% showed resistance to at least one antibiotic and 31% were MDR).

CC59 and CC5 were the most predominant clones in this study, which agrees with those previously reported in Asia (Chuang and Huang 2013; Chen and Huang, 2014). CC5 was also the most abundant CC type in a pan-European study of invasive *S. aureus* infections (Aanensen et al., 2016). Data from food-borne disease outbreaks and investigations showed that both CC59 and CC5 are common epidemic clones that cause staphylococcal food poisoning and infectious diseases (Yan et al., 2012; Song et al., 2017; Li et al., 2018). Our food isolates also showed a prevalence of SCC*mec* type IV (49.1%) and V (25.4%) similar to previous findings in both China (Liao et al. 2018) and Germany (Vossenkuhl et al., 2014). Additionally, SCC*mec*XII was also found in 63 of our food isolates. SCC*mec*XII mainly spreads as LA-MRSA in China (Wu et al., 2015) and several Asian countries, such as Japan, Malaysia, and Thailand (Neela et al., 2009; Baba et al., 2010; Larsen, et al., 2012). Recently, MRSA with SCC*mec*XII has been reported to lead to clinical infections, suggesting a potential pathogenic risk for humans (Chen et al. 2018). In our human isolates SCC*mec*III was prevalent (9/14), which is similar to other hospital-based studies in China (Liu et al., 2010; Peng et al., 2010), whereas SCC*mec*IV was prevalent in studies from Europe (Wannet et al., 2004; Manara et al, 2018). The *spa* types in our cohort showed a different distribution to another Chinese food-based study (Liao et al, 2018), with t437 most prevalent in ours compared to t071 and t091 in Liao *et al.*, although all of the most prevalent isolates in that study were also present in our cohort. In the Italian study (Manara et al., 2018) *spa* types t001 (4.9%), t002 (4.3%), t008 (3.3%) and t127 (2.7%) were prevalent. Whilst t002 (5.8%) and t127 (5.8%) were present in our human isolates t001 and t008 were absent and *spa* types t164 (11.6%), t189 (7.2%), t037 (6.5%) and t085 (6.5%) were the most prevalent. Our cohort has a

low level of PVL+ human isolates (2.17%) compared to other studies in China (Yu et al., 2008) (12.8%), Europe (Manara et al., 2018) (27.4%) and Africa (Schaumburg et al., 2014) (17-80%)”

References

- Aanensen, D. M., E. J. Feil, M. T. G. Holden, J. Dordel, C. A. Yeats, A. Fedosejev, R. Goater, S. Castillo-Ramírez, J. Corander, C. Colijn, M. A. Chlebowicz, L. Schouls, M. Heck, G. Pluister, R. Ruimy, G. Kahlmeter, J. Åhman, E. Matuschek, A. W. Friedrich, J. Parkhill, S. D. Bentley, B. G. Spratt and H. Grundmann (2016). "Whole-Genome Sequencing for Routine Pathogen Surveillance in Public Health: a Population Snapshot of Invasive *Staphylococcus aureus* in Europe." *mBio* 7(3): e00444-00416.
- Baba, K., K. Ishihara, M. Ozawa, Y. Tamura and T. Asai (2010). "Isolation of methicillin-resistant *Staphylococcus aureus* (MRSA) from swine in Japan." *Int J Antimicrob Agents* 36(4): 352-354.
- Chen, C.-J. and Y.-C. Huang (2014). "New epidemiology of *Staphylococcus aureus* infection in Asia." *Clinical Microbiology and Infection* 20(7): 605-623.
- Chuang, Y.-Y. and Y.-C. Huang (2013). "Molecular epidemiology of community-associated methicillin-resistant *Staphylococcus aureus* in Asia." *The Lancet Infectious Diseases* 13(8): 698-708.
- Chen, C. J., T. Y. Lauderdale, C. T. Lu, Y. Y. Chuang, C. C. Yang, T. S. Wu, C. Y. Lee, M. C. Lu, W. C. Ko and Y. C. Huang (2018). "Clinical and molecular features of MDR livestock-associated MRSA ST9 with staphylococcal cassette chromosome mecXII in humans." *J Antimicrob Chemother* 73(1): 33-40.
- Ge B, Mukherjee S, Hsu CH, Davis JA, Tran TTT, Yang Q, Abbott JW, Ayers SL, Young SR, Crarey ET, Womack NA, Zhao S, McDermott PF. 2017. MRSA and multidrug-resistant *Staphylococcus aureus* in U.S. retail meats, 2010-2011. *Food Microbiol* 62:289-297.
- Kim YB, Seo KW, Jeon HY, Lim SK, Lee YJ. 2018. Characteristics of the antimicrobial resistance of *Staphylococcus aureus* isolated from chicken meat produced by different integrated broiler operations in Korea. *Poult Sci* 97:962-969.

Kroning IS, Iglesias MA, Sehn CP, Valente Gandra TK, Mata MM, da Silva WP. 2016. *Staphylococcus aureus* isolated from handmade sweets: Biofilm formation, enterotoxigenicity and antimicrobial resistance. *Food Microbiol* 58:105-11.

Larsen, J., M. Imanishi, S. Hinjoy, P. Tharavichitkul, K. Duangsong, M. F. Davis, K. E. Nelson, A. R. Larsen and R. L. Skov (2012). "Methicillin-resistant *Staphylococcus aureus* ST9 in pigs in Thailand." *PLoS One* 7(2): e31245.

Li, S., S. Sun, C. Yang, H. Chen, Y. Yin, H. Li, C. Zhao and H. Wang (2018). "The Changing Pattern of Population Structure of *Staphylococcus aureus* from Bacteremia in China from 2013 to 2016: ST239-030-MRSA Replaced by ST59-t437." *Front Microbiol* 9: 332.

Liao, F., W. Gu, Z. Yang, Z. Mo, L. Fan, Y. Guo, X. Fu, W. Xu, C. Li and J. Dai (2018). "Molecular characteristics of *Staphylococcus aureus* isolates from food surveillance in southwest China." *BMC Microbiology* 18(1): 91.

Liu, Q.-Z., Q. Wu, Y.-B. Zhang, M.-N. Liu, F.-P. Hu, X.-G. Xu, D.-M. Zhu and Y.-X. Ni (2010). "Prevalence of clinical methicillin-resistant *Staphylococcus aureus* (MRSA) with high-level mupirocin resistance in Shanghai and Wenzhou, China." *International journal of antimicrobial agents* 35(2): 114-118.

Manara, S., E. Pasolli, D. Dolce, N. Ravenni, S. Campana, F. Armanini, F. Asnicar, A. Mengoni, L. Galli, C. Montagnani, E. Venturini, O. Rota-Stabelli, G. Grandi, G. Taccetti and N. Segata (2018). "Whole-genome epidemiology, characterisation, and phylogenetic reconstruction of *Staphylococcus aureus* strains in a paediatric hospital." *Genome Medicine* 10(1): 82.

Neela V, Mohd Zafrul A, Mariana NS, van Belkum A, Liew YK, Rad EG. 2009. Prevalence of ST9 methicillin-resistant *Staphylococcus aureus* among pigs and pig handlers in Malaysia. *J Clin Microbiol* 47:4138-40.

Pekana A, Green E. 2018. Antimicrobial Resistance Profiles of *Staphylococcus aureus* Isolated from Meat Carcasses and Bovine Milk in Abattoirs and Dairy Farms of the Eastern Cape, South Africa. *Int J Environ Res Public Health* 15.

Peng, Q., B. Hou, S. Zhou, Y. Huang, D. Hua, F. Yao and Y. shu Qian (2010). "Staphylococcal cassette chromosome mec (SCCmec) analysis and antimicrobial

susceptibility profiles of methicillin-resistant *Staphylococcus aureus* (MRSA) isolates in a teaching hospital, Shantou, China." *African Journal of Microbiology Research* 4(9): 844-848.

Schaumburg, F., A. S. Alabi, G. Peters and K. Becker (2014). "New epidemiology of *Staphylococcus aureus* infection in Africa." *Clin Microbiol Infect* 20(7): 589-596.

Song, Z., F. F. Gu, X. K. Guo, Y. X. Ni, P. He and L. Z. Han (2017). "Antimicrobial Resistance and Molecular Characterization of *Staphylococcus aureus* Causing Childhood Pneumonia in Shanghai." *Front Microbiol* 8: 455.

Vossenkuhl, B., J. Brandt, A. Fetsch, A. Käsbohrer, B. Kraushaar, K. Alt and B.-A. Tenhagen (2014). "Comparison of spa Types, SCCmec Types and Antimicrobial Resistance Profiles of MRSA Isolated from Turkeys at Farm, Slaughter and from Retail Meat Indicates Transmission along the Production Chain." *PLOS ONE* 9(5): e96308.

Wannet, W., M. Heck, G. Pluister, E. Spalburg, M. Van Santen, X. Huijsdans, E. Tiemersma and A. De Neeling (2004). "Panton-Valentine leukocidin positive MRSA in 2003: the Dutch situation." *Eurosurveillance* 9(11): 3-4.

Wu Z, Li F, Liu D, Xue H, Zhao X. 2015. Novel Type XII Staphylococcal Cassette Chromosome mec Harboring a New Cassette Chromosome Recombinase, CcrC2. *Antimicrob Agents Chemother* 59:7597-601.

Yan, X., B. Wang, X. Tao, Q. Hu, Z. Cui, J. Zhang, Y. Lin, Y. You, X. Shi and H. Grundmann (2012). "Characterization of *Staphylococcus aureus* strains associated with food poisoning in Shenzhen, China." *Appl Environ Microbiol* 78(18): 6637-6642.

Zehra A, Gulzar M, Singh R, Kaur S, Gill JPS. 2019. Prevalence, multidrug resistance and molecular typing of methicillin-resistant *Staphylococcus aureus* (MRSA) in retail meat from Punjab, India. *J Glob Antimicrob Resist* 16:152-158.

As already mentioned above, also the novel cassettes are not discussed in light of available literature, with particular mention to the SCCmecIVc variant that is already described in Manara et al (L166-168), which make me doubt also of the fact that the other cassettes here described as novel variants might have been described in the

literature before but it was not carefully checked.

Following the Reviewer's comment all the variants described in the manuscript have been re-checked. As the reviewer correctly pointed out, we were mistaken to describe the SCC*mecIVc* variant as novel, as it was also found in the Manara paper (Manara et al., 2018). Additionally, whilst checking the SCC*mecIVa* variant we described we have found additional new matches to an unpublished Chinese SCC*mec* cassette and a newly sequenced plasmid. We have now corrected this in the manuscript (see Results lines 184-211, Methods Lines 783-795 and below) and added the additional data into the text.

Results - Lines 184-211:

“Variants of type IV SCC*mec* cassette carrying extra antibiotic resistance genes and novel SCC*mec* cassette type. Among the identified 343 MRSA isolates, nearly half of the isolates (169/343) carried a type IV SCC*mec* cassette. However, in this study, by aligning reconstructed SCC*mec* with the reference cassettes (Ref IVa: AB063172.2 and Ref IVc: B096217.1), we observed several variabilities inside the type IV cassette. Two subtype IVc isolates were found to have the insertion of kanamycin and bleomycin resistance genes as previously found in Italy by Manara et al (Manara et al. 2018). Our isolates (11A1151 and 18A25) were almost identical to the cassette (MF062) found by Manara *et al.* (Manara et al., 2018), see Figure 2. Isolate 11A1151 carried a cassette identical to MF062, whilst isolate 18A25 had a deletion of the gene *uqpQ*. We confirmed our variant to be an integrated plasmid, pUB110 (McKenzie *et al.*, 1986) via a BLAST search against N315 which is a confirmed SCC*mecII* isolate with integrated pUB110 (assession no D81934.2, identity 99.93%). The insertion of plasmid pUB110, carrying these resistance genes, is frequently associated with type II SCC*mec* elements (Yamaguchi et al., 2020, Ito et al., 1999; Oliveira et al., 2001, Dubin *et al.* 1991).

An SCC*mecIVa* cassette variant was found, in two isolates from our cohort, to have genes conferring resistance to beta-lactams; *blaI*, *blaR1* and *blaZ* were found between IS431 and the *mec* gene complex. When blasting this SCC*mec* cassette

variant against the NCBI nonredundant database we found as subject sequence this *SCCmec* cassette variant (99% identity and query cover) has recently also been found in an *SCCmecIVa* clinical isolate from Wuhan, China (GenBank CP033086.1). The genome sequence, located in the J3 region (Yamaguchi *et al.*, 2020) of the cassette indicated an integrated plasmid as its structure *Tn552-CadX-CadD* is associated with plasmid pBORa53 (Massida *et al.*, 2006). The plasmid-like region also shows high similarity (100% identity and 94% query cover) to *S. aureus* plasmid pPM1 (GenBank AB699881.1).

According to the International Working Group on the Classification of Staphylococcal Cassette Chromosome Elements and recent reports, there are 14 types of *SCCmec* cassette (IWG-SCC, 2009; Wu *et al.*, 2015; Baig *et al.*, 2018; Urushibara *et al.*, 2020). Three of our MRSA isolates could not be attributed to any of the 14 known cassette types. These novel *SCCmec* elements carried the *ccr* gene complex 7 (A1B6) and the *mec* gene complex class A. Therefore, our study highlights the presence of further resistances and diversity within the same cassette type and the ongoing rearrangements in MRSA.”

Methods – Lines 783-795:

“*SCCmecFinder* (Kaya *et al.*, 2018) was used for typing *SCCmec* and the contigs that aligned to the *SCCmec* region were further annotated with Prokav1.14.5 (Seeman, 2014). The names of predicted proteins from Prokka were further confirmed with BLASTP (Altschul *et al.*, 1990). The variant *SCCmec* regions were further aligned using BLAST to identify *SCCmec* variants and plasmid integration into the *SCCmec* cassette (Rolo *et al.*, 2017; Côrtes *et al.*, 2018). *SCCmec* variants in isolates 11A1151 and 18A25 (region encompassing *OrfX (rlmH)* to the direct repeat (DR) sequence GAAGCTTATCATAAGTAA), were aligned to MF062 (GenBank GCA_003240235) and N315 (GenBank D81934.2) using default BLASTn settings (Altschul *et al.*, 1990). Pairwise comparison using BLASTn with default settings was performed for the *SCCmecIVa* variant from isolates 11A832 and 18A245 (region encompassing *OrfX(rlmH)* to the DR sequence GAAGCGTATCATAAATGA) against the NCBI

nonredundant database using both the SCC*mec* cassette region and only the J3 region of the cassette, using default BLASTn settings.”

References

- Aanensen, D. M., E. J. Feil, M. T. G. Holden, J. Dordel, C. A. Yeats, A. Fedosejev, R. Goater, S. Castillo-Ramírez, J. Corander, C. Colijn, M. A. Chlebowicz, L. Schouls, M. Heck, G. Pluister, R. Ruimy, G. Kahlmeter, J. Åhman, E. Matuschek, A. W. Friedrich, J. Parkhill, S. D. Bentley, B. G. Spratt and H. Grundmann (2016). "Whole-Genome Sequencing for Routine Pathogen Surveillance in Public Health: a Population Snapshot of Invasive *Staphylococcus aureus* in Europe." *mBio* 7(3): e00444-00416.
- Baig, S., T. B. Johannesen, S. Overballe-Petersen, J. Larsen, A. R. Larsen and M. Stegger (2018). "Novel SCC*mec* type XIII (9A) identified in an ST152 methicillin-resistant *Staphylococcus aureus*." *Infect Genet Evol* 61: 74-76.
- Chen, C.-J. and Y.-C. Huang (2014). "New epidemiology of *Staphylococcus aureus* infection in Asia." *Clinical Microbiology and Infection* 20(7): 605-623.
- Chuang, Y.-Y. and Y.-C. Huang (2013). "Molecular epidemiology of community-associated methicillin-resistant *Staphylococcus aureus* in Asia." *The Lancet Infectious Diseases* 13(8): 698-708.
- Cuny, C., R. Köck and W. Witte (2013). "Livestock associated MRSA (LA-MRSA) and its relevance for humans in Germany." *Int J Med Microbiol* 303(6-7): 331-337.
- Drummond, A. J. and A. Rambaut (2007). "BEAST: Bayesian evolutionary analysis by sampling trees." *BMC Evolutionary Biology* 7(1): 214.
- Hyun, J. C., E. S. Kavvas, J. M. Monk and B. O. Palsson (2020). "Machine learning with random subspace ensembles identifies antimicrobial resistance determinants from pan-genomes of three pathogens." *PLOS Computational Biology* 16(3): e1007608.
- Ito, T., Y. Katayama and K. Hiramatsu (1999). "Cloning and nucleotide sequence determination of the entire *mec* DNA of pre-methicillin-resistant *Staphylococcus aureus* N315." *Antimicrob Agents Chemother* 43(6): 1449-1458.

IWG-SCC (2009). "Classification of staphylococcal cassette chromosome mec (SCCmec): guidelines for reporting novel SCCmec elements." *Antimicrob Agents Chemother* 53(12): 4961-4967.

Manara, S., E. Pasolli, D. Dolce, N. Ravenni, S. Campana, F. Armanini, F. Asnicar, A. Mengoni, L. Galli, C. Montagnani, E. Venturini, O. Rota-Stabelli, G. Grandi, G. Taccetti and N. Segata (2018). "Whole-genome epidemiology, characterisation, and phylogenetic reconstruction of *Staphylococcus aureus* strains in a paediatric hospital." *Genome Medicine* 10(1): 82.

Massidda, O., M. Mingoia, D. Fadda, M. B. Whalen, M. P. Montanari and P. E. Varaldo (2006). "Analysis of the β -lactamase plasmid of borderline methicillin-susceptible *Staphylococcus aureus*: Focus on bla complex genes and cadmium resistance determinants cadD and cadX." *Plasmid* 55(2): 114-127.

Oliveira, D. C., A. Tomasz and H. de Lencastre (2001). "The evolution of pandemic clones of methicillin-resistant *Staphylococcus aureus*: identification of two ancestral genetic backgrounds and the associated mec elements." *Microb Drug Resist* 7(4): 349-361.

Urushibara, N., M. S. Aung, M. Kawaguchiya and N. Kobayashi (2020). "Novel staphylococcal cassette chromosome mec (SCCmec) type XIV (5A) and a truncated SCCmec element in SCC composite islands carrying speG in ST5 MRSA in Japan." *J Antimicrob Chemother* 75(1): 46-50.

Wu, Z., F. Li, D. Liu, H. Xue and X. Zhao (2015). "Novel Type XII Staphylococcal Cassette Chromosome mec Harboring a New Cassette Chromosome Recombinase, CcrC2." *Antimicrob Agents Chemother* 59(12): 7597-7601.

Yamaguchi, T., D. Ono and A. Sato (2020). *Staphylococcal Cassette Chromosome mec (SCCmec) Analysis of MRSA. Methicillin-Resistant Staphylococcus Aureus (MRSA) Protocols*, Springer: 59-78.

3. The Authors report that they found 29 novel STs, but do not provide the allelic profile of the MLST genes, which could easily be added to Supplementary Table 2, and do not discuss for instance how much different these new STs are from those

already available. It is known that single-nucleotide sequencing errors occur frequently in whole-genome sequencing, therefore it would be relevant to know whether these new STs are only one SNP away from "common STs" or their allelic profile is completely different. Also, it is not reported from which sources these new STs were isolated.

Following the reviewer suggestion, we have added the allelic profiles of the 29 novel STs in Supplementary Table 2. The isolates that were suspected to possess novel STs were confirmed by PCR and DNA sequencing before being submitted to PubMLST. We have provided all this information in the methods (See Methods lines 774-780 and below) and added a description of the source of the novel ST types to the results (see Results lines 137-143).

Methods - Lines 774-780:

Sequence types were identified through MLST which mapped the sequences to PubMLST *Staphylococcus aureus* MLST database (Jolley et al., 2018). Novel STs all showed one or more single base mutations from known STs and were further confirmed by PCR, following the same protocol given on PubMLST website (<https://pubmlst.org/organisms/staphylococcus-aureus/primers>), and DNA sequencing before being submitted to PubMLST for verification (Jolley et al., 2018).

Results - Lines 137-143:

"A total of 72 STs were identified, of these 29 STs (from 47 isolates) were found to be novel (Supplementary Tables 1 and 2), showing a single-base mutation in one or more alleles. Two STs (ST3656 and ST5713) out of these 29 were identified from human-associated isolates. The other 27 STs were from food-associated isolates, most of which were animal food-related (24 STs from 36 isolates) (Supplementary Table 2). The resistance and virulence profiles of these novel STs were similar to their clonal complexes."

References

Jolley, K., J. Bray and M. Maiden (2018). "Open-access bacterial population genomics: BIGSdb software, the PubMLST.org website and their applications [version 1; peer review: 2 approved]." Wellcome Open Research 3(124).

4. When there is some discussion of the results, this is pretty poor. For instance, in L192-193 the explanation for the presence of a single isolate from pork in Beijing clustering with US ones is that it could be an imported pork product (why?????Find evidence in the literature or import is important), while the closest isolates coming from pigs are not considered as an explanation (e.g. a lineage associated with live pigs that is then found in pork meat?). The same is true for L195-197 (and later in L387-388), where the explanation for the presence of ST398 is cross-contamination from food, when a large body of literature has shown that ST398 is becoming commonly present in humans (among others, Aanensen et al. and Manara et al. already reported above, <https://doi.org/10.1016/j.ijmm.2013.02.010>, DOI: 10.3201/eid1703.101036, and DOI: 10.1371/journal.pone.0041855) as we can see also from Figure 3C, where a number of isolates come from humans. Humans could be the most obvious source of contamination for food preparations.

As correctly suggested by the reviewer the reasons underlying the clustering of ST9 needed further explanation the same is true for the presence of ST398 as possibly related to human contamination. The results have been edited (see Results lines 240-257 and below) to reflect these concepts.

Results -Lines 240-257

"The only exception was an isolate from pork in Beijing which clustered with pig isolates from the US. One possible explanation for this US/China clustering is that China is one of the top three markets of the US pork production (USDA, 2019), and the MRSA isolates might be transported during pork trade. As previously observed MRSA clones can transmit cross-regionally via food trading (Agersø et al, 2012; Kim et al, 2015; Rodríguez-Lázaro et al, 2017), although evidence of a risk to humans

from this is lacking. ST398 (Figure 3c) is commonly found in livestock-associated isolates from pig farming in Europe (Witte, Strommenger et al. 2007, Pan et al. 2009) and also in China (Li et al. 2015), though at very low prevalence. It is possible that this isolate derives from ST398 sourced from swine in China. However, here, ST398 was found in higher numbers (5%, n=34), from human and more diverse food sources, including pork and non-meat products. Several studies have shown that ST398 is becoming commonly present in humans (Cuny et al., 2013; Aanensen et al., 2016; Manara et al., 2018), suggesting that humans could be an obvious source of contamination for food preparations.”

References

- Aanensen, D. M., E. J. Feil, M. T. G. Holden, J. Dordel, C. A. Yeats, A. Fedosejev, R. Goater, S. Castillo-Ramírez, J. Corander, C. Colijn, M. A. Chlebowicz, L. Schouls, M. Heck, G. Pluister, R. Ruimy, G. Kahlmeter, J. Åhman, E. Matuschek, A. W. Friedrich, J. Parkhill, S. D. Bentley, B. G. Spratt and H. Grundmann (2016). "Whole-Genome Sequencing for Routine Pathogen Surveillance in Public Health: a Population Snapshot of *Staphylococcus aureus* in Europe." *mBio* 7(3): e00444-00416.
- Agersø Y, Hasman H, Cavaco LM, Pedersen K, Aarestrup FM. (2012). Study of methicillin resistant *Staphylococcus aureus* (MRSA) in Danish pigs at slaughter and in imported retail meat reveals a novel MRSA type in slaughter pigs. *Vet Microbiol* 157: 246-250.
- Cuny, C., R. Köck and W. Witte (2013). "Livestock associated MRSA (LA-MRSA) and its relevance for humans in Germany." *Int J Med Microbiol* 303(6-7): 331-337.
- Kim YJ, Oh DH, Song BR, Heo EJ, Lim JS, Moon JS, Park HJ, Wee SH, Sung K. (2015). Molecular characterization, antibiotic resistance, and virulence factors of methicillin-resistant *Staphylococcus aureus* strains isolated from imported and domestic meat in Korea. *Foodborne Pathog Dis* 12: 390-398.
- Li, G., C. Wu, X. Wang and J. Meng (2015). "Prevalence and characterization of methicillin susceptible *Staphylococcus aureus* ST398 isolates from retail foods." *International Journal of Food Microbiology* 196: 94-97.

Luo R, Zhao L, Du P, Luo H, Ren X, Pang L, Cui S, Luo Y. (2020). Characterization of an oxacillin-susceptible meca-positive *Staphylococcus aureus* isolate from an imported meat product. *Microb Drug Resist* 26: 89-93.

Manara, S., E. Pasolli, D. Dolce, N. Ravenni, S. Campana, F. Armanini, F. Asnicar, A. Mengoni, L. Galli, C. Montagnani, E. Venturini, O. Rota-Stabelli, G. Grandi, G. Taccetti and N. Segata (2018). "Whole-genome epidemiology, characterisation, and phylogenetic reconstruction of *Staphylococcus aureus* strains in a paediatric hospital." *Genome Medicine* 10(1): 82.

Pan, A., A. Battisti, A. Zoncada, F. Bernieri, M. Boldini, A. Franco, M. Giorgi, M. Iurescia, S. Lorenzotti, M. Martinotti, M. Monaci and A. Pantosti (2009).

"Community-acquired methicillin-resistant *Staphylococcus aureus* ST398 infection, Italy." *Emerging infectious diseases* 15(5): 845-847.

Rodríguez-Lázaro D, Oniciuc EA, García PG, Gallego D, Fernández-Nata I, Dominguez-Gil M, Eiros-Bouza JM, Wagner M, Nicolau AI, Hernández M (2017). Detection and characterization of *Staphylococcus aureus* and methicillin-resistant *S. aureus* in foods confiscated in EU borders. *Front Microbiol* 8:1344.

USDA (2019). 2019 U.S. Agricultural Export Yearbook. F. A. Service.

Witte, W., B. Strommenger, C. Stanek and C. Cuny (2007). "Methicillin-resistant *Staphylococcus aureus* ST398 in humans and animals, Central Europe." *Emerging infectious diseases* 13(2): 255-258.

5. Most Figures are not readable as there are many colors (too many unnecessary details!!!!) and the font used is extremely small. Please see below for more specific comments on the single Figures. This results in specific sections of the manuscript pointing to Figures where the reader cannot really observe what the Authors would like them to observe, e.g. L248-249 that point to Figure 5b (unreadable) and even worse L253-256 that point to Supplementary Figure 2a (even more unreadable and therefore useless).

As correctly pointed out by the reviewer the figures could not be properly appreciated by the reader, we have now edited the colours, font and content of figures (1, 3, 4, 5, Supplementary Figure 1 and Supplementary Figure 2). The edits have been done and described in relation to the specific comments provided by the Reviewer (see below).

6. There is a large section of the manuscript that is repeated (L320-342 are exactly the same as L298-L320).

Apologies for this, we have erroneously duplicated the same text. We have now deleted the duplicated lines 387-409.

7. There is no data reported for the Bayesian analysis. Which clock models, coalescent prior, and substitution models did the Authors test? Likelihood and Bayes factor should be reported for all tested combinations (including the chosen one) for the reader to understand how reliable these analyses are. 60 million steps are not a lot, but I can understand that the Authors needed to reduce this number to speed up the analysis.

We undertook tests of several models and used path sampling to select the best model, unfortunately in the original submission we were unable to get the model to converge. We have now been able to run longer chains for all models and have used the model objectively selected as the best (GTR gamma substitution model, random uncorrelated exponential clock model and Bayesian skyline growth model). The results of the models are largely the same as previously. The methods section has been updated to include the additional details requested by the reviewer and the results and discussion modified as appropriate, please see lines Methods 824-845, Results Lines 273-294 and below:

Methods (Lines 824-845):

“Bayesian divergence estimates. A subset of sequences from this study and

PATRIC from ST9-t899-SCC*mecXII*, ST59-t437-SCC*mecIV/V*, ST239-t037-SCC*mecIII* and ST398-t011-SCC*mecV* in our cohort, alongside reference genomes publicly available (Wattam et al., 2017) for the same lineages, for which collection dates were available were selected for Bayesian evolutionary analysis using BEAST v 1.10.4 (Suchard et al., 2018). As only two reference genomes belonging to ST59-t437-SCC*mecIV/V* were available, the other eleven ST59-SCC*mecIV/V* reference genomes were also recruited to the analysis dataset. Analysis was conducted on a core genome alignment of each lineage, using Roary v3.11.2 (Page et al., 2015). All combinations of three clock models (strict, uncorrelated log normal and uncorrelated exponential) and four tree priors (constant coalescent, logistic growth, Bayesian skyline and birth-death model) were tested using stepping stone sampling on a subset of the isolates (ST239 isolates) to identify the best model. Log marginal likelihood values were in the range -2459423 to -2459156. The best model was a random uncorrelated exponential clock model, with Bayesian skyline growth model. The GTR-gamma nucleotide substitution model was used, as selected for the maximum likelihood tree. For ST59, the model did not converge after 250 million steps so instead a simpler model was used (strict clock and constant coalescent growth). The analysis was run for 2 independent chains until the Effective Sample Size (ESS), that is the effective number independent draws from the posterior distribution, for all parameters was greater than 200 per chain. This entailed each chain running for approximately 150 million steps. Convergence was assessed in Tracer v1.7.1 (Rambaut et al. 2018) and chains were subsequently combined using LogCombiner v1.10.4 (Drummond and Rambaut, 2007). The maximum clade credibility tree was selected using TreeAnnotator v1.10.4 (Drummond and Rambaut, 2007) and then visualised in iTOL v5 (Letunic and Bork, 2019)."

Results - Lines 273-294):

"For ST59-SCC*mecIV/V*, the non-Chinese isolates showed a pattern of divergence starting in the 1960s, with divergence from Chinese clones occurring in the 1940s, earlier than another study of East Asian clones (Ward et al., 2016), however small

samples sizes in both studies may explain the differences. On the contrary, Chinese isolates diversified from ~1980 onwards, with the majority of divergence occurring after ~1990, in agreement with another study of food-related *S. aureus* isolates in China (Pang et al. 2020). Our data included a small number of human samples from China within this clone (two from our cohort and three reference genomes). These isolates showed no differential evolution to the food-related *S. aureus* isolates and were most closely related to isolates from both meat and vegetable products, suggestive of CA- rather than HA- transmission. ST9-t899-SCC*mecXII* isolates showed much more recent diversification with our isolates diversifying between 2004 and 2010, the most recent common ancestor dating to the year 2000. Diversification of the Chinese ST398-t011-SCC*mecV* isolates dated back to ~2008, later than human ST398 isolates in a recent study in Taiwan (Huang and Chen, 2020), with their most recent common ancestor to the European strains dating back to approximately 1996. ST239-t037-SCC*mecIII*, showed a diverse pattern of evolution, with diversification starting in the 1950s. Our Chinese isolates diversified from 1995 onwards and were found most closely related to other samples from China. Two of our samples were more closely related to samples from Algeria with earlier diversification. The evolution of our Chinese cohort appears to differ from geographically local countries, such as Singapore (Hsu et al. 2015), possibly due to different political and economic histories. Overall, the molecular evolution of MRSA in China appears to be more recent than in other countries.”

References

- Drummond, A. J. and A. Rambaut (2007). "BEAST: Bayesian evolutionary analysis by sampling trees." *BMC Evolutionary Biology* 7(1): 214.
- Hsu, L.-Y., S. R. Harris, M. A. Chlebowicz, J. A. Lindsay, T.-H. Koh, P. Krishnan, T.-Y. Tan, P.-Y. Hon, W. B. Grubb, S. D. Bentley, J. Parkhill, S. J. Peacock and M. T. G. Holden (2015). "Evolutionary dynamics of methicillin-resistant *Staphylococcus aureus* within a healthcare system." *Genome biology* 16(1): 81-81.

Huang, Y.-C. and C.-J. Chen (2020). "Detection and phylogeny of *Staphylococcus aureus* sequence type 398 in Taiwan." *Journal of Biomedical Science* 27(1): 15.

Letunic, I. and P. Bork (2019). "Interactive Tree Of Life (iTOL) v4: recent updates and new developments." *Nucleic acids research* 47(W1): W256-W259.

Page, A. J., C. A. Cummins, M. Hunt, V. K. Wong, S. Reuter, M. T. G. Holden, M. Fookes, D. Falush, J. A. Keane and J. Parkhill (2015). "Roary: rapid large-scale prokaryote pan genome analysis." *Bioinformatics (Oxford, England)* 31(22): 3691-3693.

Pang, R., S. Wu, F. Zhang, J. Huang, H. Wu, J. Zhang, Y. Li, Y. Ding, J. Zhang, M. Chen, X. Wei, Y. Zhang, Q. Gu, Z. Zhou, B. Liang, W. Li and Q. Wu (2020). "The Genomic Context for the Evolution and Transmission of Community-Associated *Staphylococcus aureus* ST59 Through the Food Chain." *Frontiers in Microbiology* 11(422).

Rambaut, A., A. J. Drummond, D. Xie, G. Baele and M. A. Suchard (2018). "Posterior Summarization in Bayesian Phylogenetics Using Tracer 1.7." *Syst Biol* 67(5): 901-904.

Suchard, M. A., P. Lemey, G. Baele, D. L. Ayres, A. J. Drummond and A. Rambaut (2018). "Bayesian phylogenetic and phylodynamic data integration using BEAST 1.10." *Virus Evol* 4(1): vey016.

Ward, M. J., M. Goncheva, E. Richardson, P. R. McAdam, E. Raftis, A. Kearns, R. S. Daum, M. Z. David, T. L. Lauderdale, G. F. Edwards, G. R. Nimmo, G. W. Coombs, X. Huijsdens, M. E. J. Woolhouse and J. R. Fitzgerald (2016). "Identification of source and sink populations for the emergence and global spread of the East-Asia clone of community-associated MRSA." *Genome biology* 17(1): 160-160.

Wattam AR, Davis JJ, Assaf R, Boisvert S, Brettin T, Bun C, Conrad N, Dietrich EM, Disz T, Gabbard JL, Gerdes S, Henry CS, Kenyon RW, Machi D, Mao C, Nordberg EK, Olsen GJ, Murphy-Olson DE, Olson R, Overbeek R, Parrello B, Pusch GD, Shukla M, Vonstein V, Warren A, Xia F, Yoo H, Stevens RL. 2017. Improvements to PATRIC, the all-bacterial Bioinformatics Database and Analysis Resource Center. *Nucleic Acids Res* 45:D535-d542.

Additionally, I also have some minor points that need to be addressed:

L27: food poisoning also results in infections, I would suggest rephrasing the sentence with something like "from food poisoning to lethal infections".

We have rephrased the Abstract as suggested by the Reviewer see Abstract lines 26-27 and below

Abstract – Lines 26-27:

“*Staphylococcus aureus* is a worldwide leading cause of numerous diseases ranging from food-poisoning to lethal infections.”

L39-42: these sentences are very convoluted and the meaning is not clear. I suggest the Authors rephrase at least the second sentence with something like "This pattern was not observed for ST398 clones". Moreover, I am not sure why the Authors did not mention ST9 together with ST59 and ST239 in the first sentence, as it shows a very similar pattern to ST59 according to Figure 4.

Again many thanks for the useful suggestion, we have now edited the text (Abstract - Lines 39-42 and below).

Abstract - Lines 39-42:

“New Bayesian dating of the split times of major clades, showed that ST9, ST59 and ST239 in China and European countries fell in different branches, whereas this pattern was not observed for ST398 clones.”

L108-116: At the beginning of the Results section there is a very convoluted description of the isolates collected over the years, which are then not analyzed nor used for comparison in the study. I do not see the point of adding these numbers, which are not relevant for the study itself and result really confusing for the reader. If the Authors want to give an overview of the total number of isolates collected over the

years, I suggest they condense this information in one sentence like the following one:
"Food-associated *S. aureus* isolates used in this study were selected from a collection of 7937 isolates collected between 2010 and 2018 from different foods in 27 provinces in China. For this study, we selected 593 food-associated *S. aureus* isolates (343 MRSA and 250 MSSA) and sequenced them together with 142 isolates (18 MRSA and 124 MSSA) obtained from 53 healthy and 89 infected people in Shanghai between 2015 and 2017 (Supplementary Table 1). Of these, 673 resulted....
{continue with L119}"

We agree with the reviewer's comments and edited the text accordingly (Results – Lines 111-116 and below).

Results - Lines 111-116:

"Food-associated *S. aureus* isolates used in this study were selected from a collection of 7,937 isolates collected between 2010 and 2018 from different foods in 27 provinces in China. For this study, we selected 593 food-associated *S. aureus* isolates (343 MRSA and 250 MSSA) and sequenced them together with 142 isolates (18 MRSA and 124 MSSA) obtained from 53 healthy and 89 infected people in Shanghai between 2015 and 2017 (Supplementary Table 1)."

L117-118: these lines are a repetition of L109. Please remove.

Done.

L145: I think the Authors wanted to refer to Figure 1D instead of 1A, B, C?

Figure 1 has been revised, with 1A, B and C moved to the supplementary material, leaving only 1D in Figure 1 (as suggested by the reviewer to improve clarity) and the manuscript has been changed as follows.

Results - Lines 158-160:

"Analysis of the Pantone-Valentine leukocidin (PVL) genes indicated that 76 isolates

(11%, 76/673) were PVL positive and 70 of these (92%, 70/76) were MRSA (Figure 1 and Supplementary Table 1).”

L172: there is an inconsistency with Figure 2. Is the *mec* gene complex of this cassette of type A as reported in text or type B as reported in the Figure?

The *mec* gene complex is class A. Figure 2 was incorrect and this now has been corrected. Additionally, the *SCCmecIVa/c* cassette was incorrectly labelled as this is Class B, this has also been corrected.

L185: I don't think we can see this from Figure 3a (definitely not for *SCCmecV*, whose isolates are intermixed with those obtained in other countries). Are there any ST59-*SCCmecIV* isolates that were not collected in China or only the ones from Japan and Romania (I think it's Japan, too many colors)? Are the not-annotated genomes in between *SCCmecIV* and *V* carrying other cassette types? Please clarify these points, and see extra comments on the Figure below.

Figure 3 has been revised to include only continents and reduce the number of isolate sources to improve the clarity of the figure. The non-annotated isolates were also *SCCmecIV*, the figure has been updated to clarify this. This has resulted in the figure now clearly showing that *SCCmecIV* isolates were found in many countries, a large Asia only clade (with the exception of one European isolate) and a European/North American clade and a smaller Asian clade more closely linked to *SCCmecV*.

L247: AUC>80% to be consistent with the previous lines where AUC is always reported in percentage.

Done. (see Results - line 311)

L413-414: the sentence in its present form suggests that the ML approach allowed the

Authors to identify new determinants of resistance to methicillin and multiple antimicrobials, which is not the case. Please rephrase this sentence to clarify.

This sentence has been amended:

Discussion – Lines 582-584

“Thanks to ML, we were able to identify genes which, individually or in patterns, featured a strong correlation with resistance to multiple antimicrobials, regardless of the source of the isolate. “

L416-423: this is not a discussion, it is a summary of the results. Please compare with available literature.

We have edited the manuscript contextualizing the results as suggested by the reviewer (See Discussion lines 582-611 and below)

Discussion - Lines 582-611:

“Thanks to ML, we were able to identify genes which, individually or in patterns, featured a strong correlation with resistance to multiple antimicrobials, regardless of the source of the isolate. As an example of the robustness of the methodology, the *maoC*, and *ugpQ* genes previously found to be *SCCmec* associated elements (Monecke et al., 2016), in addition to *mecA*, were found strongly correlated to ceftiofur and oxacillin resistance. The identification of genes known and expected to be correlated to the selected resistant phenotypes indicates the robustness of the methods employed as stressed by Jaillard (Jaillard et al., 2018). ML also revealed a correlation between *ISSau3* (IS1182 family) and ceftiofur and oxacillin. This insertion sequence has been reported to be close to the *SCCmec* cassette and has also been reported to inactivate the gene *lytH*, increasing resistance (Fujimura and Murakami, 2008). The *rep* gene is an initiator protein in pT181 family plasmids, including pC221 known to typically carry chloramphenicol resistance (Kwong et al., 2017), indicating the likely presence of this plasmid in these food isolates. Another gene encoding tryptophan decarboxylase was also found to be significant for chloramphenicol

resistance (Kwong et al., 2017). Although not previously linked to chloramphenicol resistance, this gene is a promising gene candidate as the tryptophan biosynthesis pathway has been previously associated with vancomycin resistance (Matsuo et al., 2013). The *tcaA* gene was found to be associated with resistance to trimethoprim/sulfamethoxazole. Inactivation of this gene has been previously shown to increase resistance to teicoplanin and vancomycin (Maki et al., 2004), but no previous link has been found to trimethoprim/sulfamethoxazole. This would benefit from further experimental validation. Additionally, resistance to ciprofloxacin is typically caused by point mutation in the chromosomal *gyrA* and *parC* genes, however there is growing evidence of plasmid mediated resistance (Kim et al. 2009). In this study, the ML result showed that ciprofloxacin resistance linked to several insertion sequences suggesting a potential rapid spread of those resistance among food and humans. Notably, using ML significant correlations between virulence genes (*lpl2*, *essG*, *spIF*, *sdrE*, *map*, and *ssl7*, plus others less strongly associated) and antimicrobial resistance phenotypes were found. The correlation between the presence of ARGs and virulence factors has been observed previously (Schroeder et al., 2017) and it has been proposed that an increase in virulence allows the bacteria to overcome the fitness costs associated with the carriage of AMR genes (Guillard et al. 2016, Roux et al., 2015)."

References

- Fujimura, T. and K. Murakami (2008). "Staphylococcus aureus Clinical Isolate with High-Level Methicillin Resistance with an lytH Mutation Caused by IS1182 Insertion." Antimicrobial Agents and Chemotherapy 52(2): 643-647.
- Guillard T, Pons S, Roux D, Pier GB, Skurnik D. 2016. Antibiotic resistance and virulence: Understanding the link and its consequences for prophylaxis and therapy. Bioessays 38:682-693.
- Jaillard, M., L. Lima, M. Tournoud, P. Mahé, A. van Belkum, V. Lacroix and L. Jacob (2018). "A fast and agnostic method for bacterial genome-wide association studies:

Bridging the gap between k-mers and genetic events." *PLoS genetics* 14(11): e1007758-e1007758.

Kim, H. B., C. H. Park, C. J. Kim, E.-C. Kim, G. A. Jacoby and D. C. Hooper (2009). "Prevalence of plasmid-mediated quinolone resistance determinants over a 9-year period." *Antimicrobial agents and chemotherapy* 53(2): 639-645.

Kwong, S. M., J. P. Ramsay, S. O. Jensen and N. Firth (2017). "Replication of Staphylococcal Resistance Plasmids." *Frontiers in microbiology* 8: 2279-2279.

Maki, H., N. McCallum, M. Bischoff, A. Wada and B. Berger-Bächi (2004). "tcaA inactivation increases glycopeptide resistance in *Staphylococcus aureus*." *Antimicrobial agents and chemotherapy* 48(6): 1953-1959.

Matsuo, M., L. Cui, J. Kim and K. Hiramatsu (2013). "Comprehensive Identification of Mutations Responsible for Heterogeneous Vancomycin-Intermediate *Staphylococcus aureus* (hVISA)-to-VISA Conversion in Laboratory-Generated VISA Strains Derived from hVISA Clinical Strain Mu3." *Antimicrobial Agents and Chemotherapy* 57(12): 5843-5853.

Monecke, S., L. Jatzwauk, E. Müller, H. Nitschke, K. Pfohl, P. Slickers, A. Reissig, A. Ruppelt-Lorz and R. Ehricht (2016). "Diversity of SCCmec Elements in *Staphylococcus aureus* as Observed in South-Eastern Germany." *PLOS ONE* 11(9): e0162654.

Roux D, Danilchanka O, Guillard T, Cattoir V, Aschard H, Fu Y, Angoulvant F, Messika J, Ricard J-D, Mekalanos JJ, Lory S, Pier GB, Skurnik D. 2015. Fitness cost of antibiotic susceptibility during bacterial infection. *Sci Transl Med* 7:297ra114.

Schroeder M, Brooks BD, Brooks AE. 2017. The Complex Relationship between Virulence and Antibiotic Resistance. *Genes* 8:39.

L536: in WGS we cannot be sure that *S. aureus* genomes are completely free from contamination even after quality checks. For instance, if the isolate was a mixture of two strains, or if contamination occurred during DNA extraction and library preparation, different strains would be assembled together. Please remove "without

contamination".

Done. (see line 770-771)

L615-616: please remove the "color coding" sentence.

Done. (see lines 892-893)

Figures:

Figure 1a: The map is huge and the font is very small, it would be better to make the map smaller and the font bigger to make it readable. This panel in my opinion could be added to Supplementary Figure 1.

We have edited the figure according to the reviewer's suggestion. Panel A has been moved to Supplementary Figure 1; the map size has been reduced allowing the font to be enlarged. (see Supplementary Figure 1a)

Figure 1b: It is not easy to read black on blue, please either use white for the font on the blue bars or change the blue to a lighter color. This panel in my opinion is not very relevant and could be added to Supplementary Figure 1.

Panel 1b has been moved to Supplementary Figure 1b. The font colour on numbers overlying the coloured bars has been made white to improve clarity.

Figure 1c: Same comment as for 1b for font color on blue bars. Use lighter colors for the bars or for the font.

Panel 1c has been moved to Supplementary Figure 1c. The font colour on numbers overlying the coloured bars has been made white to improve clarity.

Figure 1d: The spa type is not readable at all, report this information in a table instead.

The SCCmec type N/A should be substituted by "MSSA" or something similar, and colored in white to make the figure more readable. The MRSA/MSSA tip points are impossible to read. Overall, this should be a separate single figure to make it readable. If panels a-c are moved to Supplementary Figure 1, the problem might be solved.

We have edited this figure following the reviewer's suggestion. Specifically, we have moved the MSSA/MRSA annotation to an outer ring to improve readability; annotated SCCmec N/A as MSSA and coloured it white; and moved panels a-c to Supplementary Figure 1. Panel 1d is now the only panel of Figure 1 and as a result is larger and more readable (see Figure 1). We have also made these same annotation changes to the related Supplementary Figure 2 (previously Supplementary Figure 3).

The legends have been edited, Lines 1373-1385, 1494-1502, 1511-1518, and below:

Figure 1. Sample information and phylogenetic tree of the whole cohort. Maximum likelihood phylogenetic tree based on 1,585 core genes of the 673 *S. aureus* isolates. Clonal complex clusters are distinguished by means of numbers and background colours. Methicillin-resistant and -susceptible phenotypes are indicated by the inner coloured ring; *agr* subgroup, SCCmec type, PVL type, and the sample sources are colour-coded in the following rings. One branch consisting of 5 isolates was removed from the tree as the length of the branch made visualisation of the tree difficult. These isolates were all found to be *S. aureus* (subgroup *S. argenteus*). The full tree is shown in Supplementary Figure 2."

Supplementary Figure 1. Sample information and antimicrobial susceptible testing of *S. aureus* isolates in this study. (a) Map of China showing number of isolates from each province. (b) Distribution of MRSA and MSSA across different sample source (food, humans) and (c) collection years. (d) Antimicrobial resistance profiles of the 673 *S. aureus* isolates against a panel of 13 antimicrobials: PEN = Penicillin; ERY = Erythromycin; CFX = Cefoxitin; OXA = Oxacillin; CLI = Clindamycin; CIP = Ciprofloxacin; TET = Tetracycline; SXT = Trimethoprim-sulfamethoxazole; CHL = Chloramphenicol; GEN = Gentamicin; DAP = Daptomycin; LZD = Linezolid and VAN

= Vancomycin.”

“Supplementary Figure 2. Maximum likelihood phylogenetic tree based on 1,585 core genes of the 673 *S. aureus* isolates, including those identified as *S. argenteus* and removed from Figure 1. Clonal complex clusters are distinguished by means of numbers and background colours. Methicillin-resistant and -susceptible phenotypes are indicated on the inner coloured ring; *agr* subgroup, *SCCmec* type, PVL type, and the sample sources are colour-coded in the subsequent rings.”

Figure 2: inconsistency between legend (and text) and figure: is the *mec* gene complex class A or B? Compare the novel variants with those called in Manara et al., because the Type IVc is the same, and I have doubts also for the others.

The *mec* gene complex is class A. Figure 2 was incorrect and this now has been corrected. Additionally, the *SCCmecIVa/c* cassette was incorrectly labelled as this is Class B, this has also been corrected. Additionally, the *SCCmecIVc* variant identified in Manara *et al.* has been added to the figure, highlighting the similarity of the variants.

The legend has been edited, lines 1387-1399, as below:

“Figure 2. Schematic representation of the *SCCmecIV* cassette variability carrying extra antimicrobial resistance genes and of a novel *SCCmec* type found in our cohort. Genes are shown with the direction of transcription and colour-coded according to their gene name. Integrated transposons or plasmids (pBORa53, Tn552, Tn554 and pUB110) upstream or downstream of the *mec* complex and carrying resistance genes are shaded in pink. The *mec* gene complex (class A and B) and the *ccr* gene complex 2 (A2B2) and 7 (A1B6) are shaded in light orange and blue, respectively. *SCCmecIV* cassettes are compared with the available references in the Genebank database (Ref IVa: AB063172.2 and Ref IVc: B096217.1) for the recovered subtypes IVa and IVc. A recently reported variant of *SCCmecIVc* element (MF062, GenBank GCA_003240235) closely related to our study is also shown. A novel *SCCmec* cassette carrying the *ccr* gene complex 7

(*ccrA1* and *ccrB6*) and the *mec* gene complex class A (*mecI-mecR1-mecA-IS431*) is aligned to the closest reference cassette *SCCmec* type II (Ref II: D86934.1). ”

Figure 3: unreadable. I would suggest to group countries by continent or subcontinent, and sources by major source groups (e.g. all noodles + rice together, all environments together, bean-product + fruit and veg, and so on) also because they are not discussed in the paper, so there's no point in keeping them separate. I can understand leaving pig / cattle / sheep or pork / beef separate. It is moreover impossible to read the tip points to understand whether it is an MRSA/MSSA. Please add a ring to the tree reporting this info.

We have edited Figure 3 following the reviewer suggestions. Specifically, we have grouped countries by continent and grouped sources by major source classes including human, animal and animal product, environment, cereal fruit and vegetable, rice and flour product. The MRSA and MSSA have been added as a ring to improve readability (see Figure 3). The legend has been edited, lines 1401-1420, as below:

“Figure3. Whole-genome maximum likelihood phylogenetic trees of the four major ST subsets. Plots show all the isolates from this study together with all publicly available reference genomes for ST59, ST9, ST398 and ST239. Methicillin-resistant and susceptible genotypes are indicated by the inner ring. Origin (sample sources) and region are colour-coded in the following rings. Isolates from this study and publicly available reference genomes are colour-coded in salmon and cyan respectively. (a) ST59 isolates. The Chinese ST59 *SCCmec* V and ST59 *SCCmec* IV isolates clustered separately when compared to the reference genomes collected elsewhere. ST59 isolates from our Chinese cohort were almost exclusively MRSA and spa-type t437 and t441 in contrast to what found elsewhere. (b) ST9 isolates. The ST9 isolates from our cohort clustered together with isolates from China and away from those collected elsewhere. ST9 isolates from our Chinese cohort were almost exclusively MRSA and spa-type t899 in contrast to what found elsewhere. (c) ST398

isolates. The Chinese ST398 showed to cluster separately when compared to the reference genomes collected elsewhere. The Chinese isolates are mainly MSSA and show a wide diversity of origin unlike in Europe where they are linked to pig farming. (d) ST239 isolates. The clustering indicates a close relationship between UK samples and samples from this study possibly indicating international routes of transmission. Two food-associated MRSA (t030) were found in this study and showed to cluster far away from the other nine HA-MRSA in this study (t037).“

Figure 4: same comments as for Figure 3. In addition, the order in the panels and in the caption is swapped and panel c is reported in capital letters in disagreement with other panels. Please remove country names from the ring, it is not readable and only adds confusion.

We have now updated Figure 4 to reflect the reviewer's comments. Specifically, we have ensured the panels are ordered as in Figure 3 and edited the lettering to lower case; we have grouped countries by continent and grouped sources by major source classes including human, animal and animal product, environment, cereal fruit and vegetable, rice and flour product (see new Figure 4). The legend has been edited, lines 1422-1429, as below:

“Figure 4. Evolution of four major MRSA lineages. Bayesian evolutionary analysis of (a) ST59-SCC*mecIV*, (b) ST9-t899-SCC*mecXII*, (c) ST239-t037-SCC*mecIII* and (d) ST398-t011-SCC*mecV*. The lineages are evolutionarily distinct. In all CC subsets, East Asian isolates cluster separately and have generally evolved later than European and American branches. Isolates from this study and publicly available reference genomes are indicated by the inner ring. Origin (sample sources) and region are colour-coded in the following rings. The size of the red circles on trees

represent the posterior probability of each node.”

Figure 5b: unreadable, especially the annotations. Please either remove them or change them to make them readable. There is no name and unit for the scale.

We have edited Figure 5b by removing the annotation for location, which was unnecessary, leaving more space to enlarge the font of the other annotations. The legend has been altered accordingly. The scale bar has been edited to indicate that it reflects the normalised k-mer count.

Legend to Figure 5b (Lines 1440-1443):

“(b) Hierarchical clustering of 673 isolates based on the 2,000 Oxacillin resistant genomic signatures (k-mers) recognised as the most significant by the trained classifier. Results have been data-mined with respect to year of collection, ST, CC, type of sample and source.”

Figure 5c: gene names and the number of k-mers are written in a very small font and are not readable. Please reduce the number of reported genes to make the heatmap understandable.

As suggested we have reduced the number of reported genes to 50 and increased the font size.

Legend to Figure 5c (Lines 1446-1452):

“(c) Hierarchically clustered heatmap of the 10 antimicrobials based on the top 50 genes corresponding to the genomic features recognised as the most significant by the trained classifiers. Genes are colour-coded according to their function: resistance gene (red), virulence gene (blue), (HGT) genes with function in horizontal gene

transfer (purple), genes with other functions (black). For each gene, the number of different k-mers present per gene per antibiotic model is shown on the plot, out of 2,000 recognised as the most significant by the trained classifier.”

Supplementary Figure 2a is not readable at all. How is the reader supposed to look at these??? Either split the panels into different figures or put these data into a Supplementary Table, there is no point in keeping this.

Apologies for this, as we were unable to achieve figure of acceptable quality within the journal's file size limits, we have instead presented this data in Supplementary Dataset 1. Please see the legend at lines 1520-1523 and below:

“Supplementary Dataset 1. Normalised k-mer counts of 673 isolates based on the 2,000 resistant genomic signatures (k-mers) recognised as the most significant by the trained classifier for each of the 9 antimicrobials. Results have been data-mined with respect to geographical location, year of collection, ST, CC, type of sample and source.”

Typos:

L40: showed THAT Chinese and European ST59 and ST239

Done. (see line 40)

L91: a comma is lacking between "ST59-SCCmecIV/V" and "ST22-SCCmecIV"

Done. (see line 94)

May 10, 2021

Prof. Fengqin Li
China National Center for Food Safety Risk Assessment
Beijing
China

Re: mSystems01185-20R1 (Whole-genome sequencing and machine learning analysis of *Staphylococcus aureus* from multiple heterogeneous sources in China reveals common genetic traits of antimicrobial resistance)

Dear Prof. Fengqin Li:

Your manuscript has been accepted, and I am forwarding it to the ASM Journals Department for publication. For your reference, ASM Journals' address is given below. Before it can be scheduled for publication, your manuscript will be checked by the mSystems senior production editor, Ellie Ghatineh, to make sure that all elements meet the technical requirements for publication. She will contact you if anything needs to be revised before copyediting and production can begin. Otherwise, you will be notified when your proofs are ready to be viewed.

- Minimum resolution of 1280 x 720
- .mov or .mp4. video format
- Provide video in the highest quality possible, but do not exceed 1080p
- Provide a still/profile picture that is 640 (w) x 720 (h) max

We recognize that the video files can become quite large, and so to avoid quality loss ASM suggests sending the video file via <https://www.wetransfer.com/>. When you have a final version of

the video and the still ready to share, please send it to Ellie Ghatineh at eghatineh@asmusa.org.

Sincerely,

Christopher Marshall
Editor, mSystems

Journals Department
Supplementary Table S5: Accept
Supplementary Table S7: Accept
Supplementary Table S2: Accept
Supplementary Table S6: Accept
Supplementary Table S1: Accept
Supplementary Table S3: Accept
Supplementary Figure 2: Accept
Supplementary Table S4: Accept
Supplementary Dataset 1: Accept
Supplementary Figure 1: Accept